# IntervalGP-VAE: Uncertainty-Aware Individual Treatment Effect Estimation via Identifiable Proxy-Based Latent Confounder Recovery

## Abstract

Estimating individual treatment effects (ITEs) in the presence of unobserved confounding remains a central challenge in causal inference. Existing proxy-based methods aim to recover latent confounders from observational proxies, but typically produce only point estimates without uncertainty quantification. This lack of uncertainty modeling provides incomplete and potentially insufficient information for downstream decision-making, especially when uncertainty is inherent in the data. We propose IntervalGP-VAE, a novel framework that combines variational autoencoders with Interval Gaussian Processes (GPs) to model both the latent confounders and their associated uncertainty. This approach accounts for uncertainty arising from noisy and imperfect proxy variables and yields calibrated ITE intervals to support more robust causal decisions. We provide theoretical guarantees for identifiability of the latent confounder up to a smooth invertible reparameterisation under weak assumptions. Experiments on identifiable synthetic datasets show that IntervalGP-VAE achieves accurate ITE estimation, reliable latent recovery, and well-calibrated ITE intervals. On the semi-synthetic IHDP benchmark, IntervalGP-VAE provides competitive PEHE and ATE estimation, sharper intervals, and lower computational cost.

## 1 Introduction

Estimating Individual Treatment Effects (ITEs) from observational data is a core challenge in causal inference, particularly when important confounding variables are not directly observed (Pearl, 2009; Peters et al., 2017). In many real-world applications, such as healthcare, economics, and social sciences, treatment assignment is influenced by latent factors that affect both the treatment and the outcome. Ignoring such hidden confounders can lead to biased causal estimates and unreliable treatment recommendations.

A promising line of recent research addresses this challenge by recovering latent confounders from observed *proxy variables* that carry indirect information about the hidden confounding structure (Louizos et al., 2017; Zhang et al., 2021; Wu et al., 2024; Harada & Kashima, 2024). Proxy-based approaches have demonstrated that under suitable structural assumptions, the latent confounder and causal effects can be recovered even when the confounder itself is unobserved. However, in practice proxy measurements are often noisy, incomplete, and weakly informative. As a result, recovering the latent confounder becomes an inverse problem under measurement error, where a single proxy observation corresponds not to a unique latent state but to a distribution of plausible latent values.

Consequently, two fundamental challenges arise in proxy-based causal inference. First, one must determine *when* recovery of latent confounders and individual causal effects is theoretically possible from proxy observations. Second, even when identifiability conditions hold, one must quantify *how much uncertainty remains* in the recovered latent structure and the resulting causal estimates due to measurement noise.

Our theoretical results (Theorem 1) address the first question by establishing structural conditions, most notably the number of proxies, conditional independence, variability and bounded completeness, under which

the latent confounder and ITE become identifiable in principle. Under these conditions, the latent confounder is identifiable up to a smooth invertible transformation, which does not affect causal identifiability.

However, identifiability alone does not imply accurate pointwise recovery: even when the model is identifiable in principle, noisy proxy measurements induce unavoidable uncertainty in the inferred latent confounder and the resulting causal estimates. This observation highlights a fundamental gap between *identifiability* and *estimation*. While identifiability theory clarifies the structural conditions under which causal recovery is possible, practical inference must still account for the epistemic uncertainty arising from noisy proxy measurements and stochastic outcome generation mechanisms. Reliable decision support therefore requires explicit modeling and propagation of this uncertainty when estimating counterfactual outcomes and ITEs.

Our interval formulation is designed to address this second challenge by capturing the uncertainty induced by measurement noise. Because proxies are noisy measurements of the latent confounder, the posterior distribution of the confounding variable induced by proxy observations typically spans a range of plausible latent values rather than collapsing to a single point. Moreover, the structural outcome equations governing the data-generating process introduce additional exogenous noise in the outcome mechanism, implying that even conditional on the latent confounder, counterfactual outcomes remain uncertain. Especially, in high-dimensional and noisy settings, point estimates of the latent confounder can become unstable and overly sensitive to measurement noise. Instead, a more appropriate representation is to infer a *set* or *distribution* of plausible latent values that reflects the uncertainty induced by proxy noise while preserving similarity relationships in the proxy space. Interval representations therefore provide a concise and interpretable summary of the combined uncertainty induced by proxy measurement noise and outcome variability, avoiding overconfident point estimates while remaining faithful to the underlying structural model. To support principled inference under this uncertainty, we model the relationship between proxies and latent confounders using Gaussian Processes (GPs) (Rasmussen & Williams, 2006). GPs provide a flexible nonparametric Bayesian framework that captures uncertainty in function estimation. The GP posterior therefore provides calibrated uncertainty estimates that can be propagated through downstream causal predictions.

Building on these insights, we propose *IntervalGP-VAE*, a novel framework that integrates interval-valued Gaussian Processes with variational autoencoders (VAEs) to recover latent confounders with uncertainty for indivdiual causal effect estimation. The VAE encoder learns a structured latent representation from noisy proxy measurements, enabling flexible modeling of complex proxy–confounder relationships. An interval-valued GP prior is imposed over this latent representation to model the confounder while explicitly accounting for measurement-induced uncertainty. Outcome prediction is further modeled using interval-valued GP regression that maps inferred confounder intervals to calibrated outcome bounds. This formulation enables coherent uncertainty propagation from noisy proxies through latent confounder recovery to interval-valued ITE estimates.

The resulting framework links theory and method in a coherent manner. Identifiability results determine *when* causal recovery from proxies is possible by specifying minimal structural conditions on the proxy generation mechanism. Once these conditions hold, the proposed IntervalGP-VAE provides a principled approach for *how* uncertainty-aware inference can be performed in practice. The interval representation captures unavoidable uncertainty arising from noisy proxy and outcome measurements, while the Gaussian Process provides a Bayesian mechanism for smooth, data-adaptive inference and calibrated uncertainty propagation.

To the best of our knowledge, IntervalGP-VAE is the first method that combines proxy-based latent variable modeling with interval-valued Gaussian Processes for causal effect estimation. While Gaussian Processes have been widely used for regression and uncertainty quantification, existing methods rarely address interval-valued observations, and none incorporate GP priors for recovering latent confounders from proxy variables. Existing approaches such as CEVAE (Louizos et al., 2017) typically rely on simple latent priors (e.g., isotropic Gaussian distributions) and do not explicitly model the structured uncertainty induced by proxy measurements. In contrast, our framework introduces structured Bayesian priors that enable principled uncertainty propagation.

Our main contributions are summarised as follows:

- We present a theoretical analysis of the identifiability conditions under which ITE can be recovered from proxies, offering formal guarantees for the proposed method.

- We propose IntervalGP-VAE, a novel framework that combines VAEs with an interval-valued GP prior over the latent space and a GP-based interval likelihood head to enable uncertainty-aware estimation of counterfactual outcomes and treatment effects.

- We evaluate IntervalGP-VAE on 24 identifiable synthetic settings and on the semi-synthetic IHDP benchmark across 100 replications. On the synthetic datasets, IntervalGP-VAE achieves accurate ITE estimation, reliable latent recovery, and well-calibrated ITE intervals. On IHDP, it provides competitive PEHE and ATE estimation, sharper intervals, and lower computational cost, highlighting its calibration–efficiency trade-off under possible proxy-assumption violations.

## 2 Problem Setting

Key notations used in this paper are summarised in Table 1. We consider the following structural causal model, illustrated in Figure 1.

**Outcome model.** The observed outcome is generated according to the structural equation

$$Y = f(T, U, Z_Y) + \epsilon_Y, \qquad (1)$$

where $T \in \{0, 1\}$ is the treatment, $U \in \mathbb{R}^d$ denotes latent confounders, $Z_Y \subseteq Z$ is a subset of proxies that directly affect the outcome, and the exogenous noise satisfies $\epsilon_Y \perp (T, U, Z_Y)$.

We allow a subset of proxy variables, denoted by $Z_Y \subseteq Z$, to serve a dual role in our framework. Like the remaining proxy variables, $Z_Y$ is generated by the latent confounder $U$ through the proxy-generation mechanism in Eq. 4 and therefore contributes information for latent confounder recovery. However, unlike the remaining proxies, $Z_Y$ is also assumed to contain outcome-relevant information and is permitted to enter the structural outcome equation directly.

Table 1: Summary of key notations.

| Symbol | Description |
|---|---|
| $Z \in \mathbb{R}^k$ | Proxy variables |
| $Z_Y \in \mathbb{R}^{k'}$ | Outcome-related proxies |
| $T \in \{0, 1\}$ | Binary treatment variable |
| $Y \in \mathbb{R}$ | Outcome variable |
| $\hat{Y}(t)$ | Estimated outcome under $t$ |
| $U \in \mathbb{R}^d$ | Latent confounder(s) |
| $\epsilon_Y, \epsilon_Z$ | Noise for $Y$ and $Z$ |
| $f(T, U, Z_Y)$ | Outcome function |
| $f_\psi(t, u, Z_Y)$ | Outcome prediction network |
| $g(U)$ | Mapping from $U$ to proxies |
| $q_\phi(u \mid z)$ | Posterior over $U$ (encoder) |
| $p_\theta(z \mid u, \epsilon)$ | Proxy likelihood (decoder) |
| $\phi, \theta, \psi$ | Encoder, Decoder, Outcome-head parameters |
| $\tau^j$ | Individual treatment effect for individual $j$ |
| $[\hat{\tau}^j_{\text{lower}}, \hat{\tau}^j_{\text{upper}}]$ | Estimated ITE interval for individual $j$ |

$Z_Y$ is treated as a pre-specified subset, chosen based on domain knowledge or the experimental design. Its dimension is denoted by $k'$ and satisfies $0 \le k' \le k$ by definition, where $k$ is the total number of proxies. When no proxy is assumed to directly affect the outcome, we set $k' = 0$, and the outcome model reduces to $Y = f(T, U) + \epsilon_Y$. We note that $Z_Y$ is not learned or selected automatically from data in the current framework. Instead, it is treated as a pre-specified subset determined before model fitting, based on domain knowledge or the experimental design. Learning $Z_Y$ in a data-adaptive manner is an interesting direction for future work, but is beyond the scope of the present method.

**ITE and counterfactual.** Following Pearl's abduction–action–prediction semantics for counterfactuals, let $\epsilon_{Y,j}$ denote the realised exogenous noise in Eq. 1 for individual $j$. The potential outcome under the intervention $T := t$ is obtained by substituting $t$ into the same structural equation:

$$Y_j(t) := f(t, U_j, Z_{Y,j}) + \epsilon_{Y,j}. \qquad (2)$$

The *individual treatment effect (ITE)* is then defined as the difference between the two counterfactual outcomes for the *same* realisation of the exogenous variables:

$$\tau_j := Y_j(1) - Y_j(0) = f(1, U_j, Z_{Y,j}) - f(0, U_j, Z_{Y,j}), \qquad (3)$$

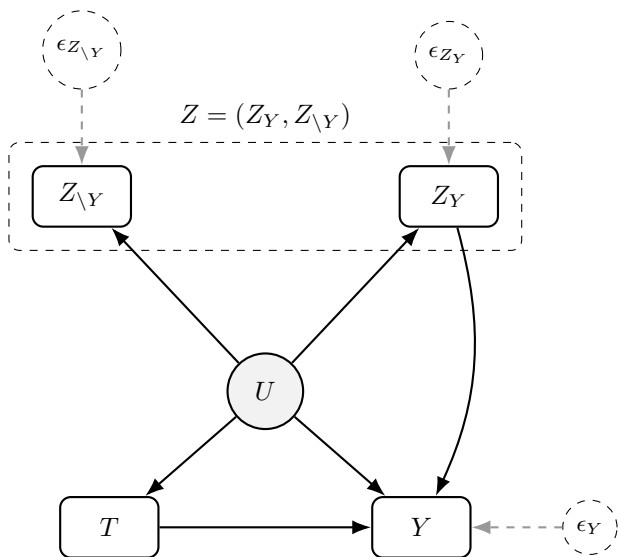

Figure 1: Structural causal model (SCM) for proxy-based ITE estimation. The latent confounder $U$ jointly influences treatment assignment $T$ and outcome $Y$. The observed proxy vector $Z = (Z_Y, Z_{\backslash Y})$ is generated from $U$ through noisy proxy mechanisms. The subset $Z_Y \subseteq Z$ denotes outcome-related proxies that may directly affect the outcome. Dashed arrows denote mutually independent exogenous noise terms.

where the exogenous noise cancels because it is held fixed across the two counterfactual worlds.

We clarify that the cross-world dependence in our framework is induced through the shared latent confounders and exogenous noise in the Structural Causal Model (SCM). In particular, holding $U$ and $\epsilon_{Y,j}$ fixed across interventions is the standard SCM mechanism for defining counterfactual outcomes. In the framework of SCMs, a counterfactual query asks: "What would have happened to this specific unit had the treatment been different, while all other conditions were held constant?" This condition must encompass not only observable background variables but also the latent stochastic factors or measurement noise $\epsilon_{Y,j}$ associated with the factual observation. As Pearl (2009) formalizes in the three-step process (*Abduction*, *Action*, *Prediction*), the *Abduction* step specifically serves to "freeze" the state of the individual (represented by $U$ and $\epsilon$) before the *Action* (intervention) is applied. Without this invariance, the concept of an "individual" effect would lose its structural grounding, as outcome variations could be conflated with fluctuating noise rather than the treatment itself.

**Proxy model.** We do not observe $U$ directly but instead observe proxies $Z = (Z_1, \ldots, Z_k)$ generated via

$$Z_i = g_i(U) + \epsilon_{Z_i}, \qquad \epsilon_{Z_i} \perp U, \quad i = 1, \ldots, k, \tag{4}$$

where the noise terms are mutually independent and the structural functions $g_i$ are unknown but sufficiently non-redundant to allow information about $U$ to be recovered from $Z$.

**Observed data** There are $n$ samples of $(Z, T, Y)$ in the observed data, and our aim is twofold:

- **Identifiability:** To determine the structural conditions under which the latent confounder $U$ can be recovered (up to an equivalence class) from the proxy variables, so that $\tau_j$ becomes identifiable from the observed distribution $p(Z, T, Y)$.

- **Uncertainty quantification:** To develop principled interval-valued estimates of treatment outcomes and ITEs with uncertainty.

To support identifiability of the latent confounder $U$ and ITE $\tau_j$, we adopt the following assumptions:

- **Positivity.** Each individual has a non-zero probability of receiving either treatment level:

$$0 < P(T = 1 \mid U) < 1.$$

- **Latent Ignorability.** Conditional on the latent confounder $U$, the treatment assignment is independent of potential outcomes:

$$(Y(0), Y(1)) \perp\!\!\!\perp T \mid U.$$

Thus $U$ fully accounts for confounding between $T$ and $Y$.

- **Consistency and well-defined outcomes.** For each $t \in \{0, 1\}$, the potential outcome $Y(t)$ is defined by substituting the intervention $T := t$ into the structural outcome equation (Eq. 1).

- **SCM-based counterfactual consistency.** Counterfactual outcomes are defined under a structural causal model. For each individual, the latent confounder $U$ and exogenous noise $\epsilon_Y$ remain unchanged across interventions. Consequently, the dependence between $Y(1)$ and $Y(0)$ is induced through the common latent variables and exogenous noise.

In addition to the standard causal assumptions above, identifiability from proxies requires two structural conditions on the proxy-generating process:

- **Conditional independence of proxies.** For all $i \neq j$,

$$Z_i \perp\!\!\!\perp Z_j \mid U. \tag{5}$$

This is consistent with the structural model $Z_i = g_i(U) + \epsilon_{Z_i}$ with mutually independent noise terms.

- **Variability of the proxy mechanism.** The proxy mechanisms satisfy the following conditions:

  1) **Joint injectivity.** The joint proxy map

  $$g = (g_1, \ldots, g_k) : \mathbb{R}^d \to \mathbb{R}^k$$

  is injective on the support of $U$, is $C^1$, and has full column-rank Jacobian almost everywhere.

  2) **Partitionable proxy variability.** There exists a fixed partition of the proxy coordinates into three disjoint blocks,

  $$Z = (Z_{L_1}, Z_{L_2}, Z_{L_3}),$$

  such that

  $$|L_1| = d + 1, \qquad |L_2| = d + 1, \qquad |L_3| \geq 2.$$

  For any finite anchor set

  $$\mathcal{U}_r = \{u^0, \ldots, u^d\}, \qquad r = d + 1,$$

  the block conditional distributions satisfy

  $\left\{ p(Z_{L_1} \mid U = u^0), \ldots, p(Z_{L_1} \mid U = u^d) \right\}$ is linearly independent as functions of $Z_{L_1}$,

  $\left\{ p(Z_{L_2} \mid U = u^0), \ldots, p(Z_{L_2} \mid U = u^d) \right\}$ is linearly independent as functions of $Z_{L_2}$,

  $\left\{ p(Z_{L_3} \mid U = u^0), \ldots, p(Z_{L_3} \mid U = u^d) \right\}$ contains at least two linearly independent functions of $Z_{L_3}$.

$$\tag{6}$$

  3) **Noise non-degeneracy.** The proxy noise density $p_\epsilon$ is non-degenerate in the sense that distinct shifts of $p_\epsilon$ are linearly independent.

Under the proxy model (Eq. 4), joint injectivity and noise non-degeneracy imply that, for any $d+1$ distinct latent values $u^0, \ldots, u^d$ in the support of $U$, the corresponding conditional proxy distributions are linearly independent:

$$p(Z \mid U = u^0), \ldots, p(Z \mid U = u^d) \quad \text{are linearly independent as functions of } Z. \tag{7}$$

- **Bounded completeness.**
  Define the linear operator

$$T(q)(z) := \int q(u) \, p(z \mid u) \, du.$$

We assume that $T$ is injective on the relevant bounded function class: for any bounded measurable function $q$,

$$T(q)(z) = 0 \quad \text{for almost every } z \quad \implies \quad q(u) = 0 \quad \text{for almost every } u. \tag{8}$$

This is a bounded-completeness condition in the sense of Hu and Schennach Hu & Schennach (2008), adapted here to the proxy-latent representation setting. See also related latent-variable identifiability discussions in Kivva et al. (2022).

The assumptions of positivity, latent ignorability, and consistency are standard in the causal inference literature and foundational for potential outcomes frameworks. In our setting, identifiability further relies on structural assumptions for the proxy and outcome models. We assume that the proxies are generated via an additive noise model (Eq. 4) with mutually independent noise terms independent of $U$. This formulation is widely used in nonlinear latent variable models and additive noise models (ANMs). The conditional independence of proxies given $U$, as well as the *variability* condition, requiring that different values of $U$ induce distinguishable distributions over the proxies, are consistent with established identifiability results in proxy or auxiliary variable-based latent models (e.g., nonlinear ICA (Hyvärinen et al., 2019; Khemakhem et al., 2020)). Further, the bounded completeness assumption intuitively ensures that distinct latent mixing distributions cannot generate the same observed proxy distribution through the same conditional proxy family.

## 3 Related Work

We categorize related work into three areas: latent confounder modeling, Gaussian Processes and VAE, and interval-valued Gaussian Processes.

### 3.1 From Proxy-Based Identification to Personalized Latent Recovery

Leveraging proxy variables to address unobserved confounding is a classical strategy in causal inference. Early measurement-error and proxy-based approaches, such as Kuroki and Pearl (Kuroki & Pearl, 2014) and Miao et al. (Miao et al., 2018), established identification results using proxy variables under rank, completeness, or invertibility conditions. Building on this line of work, proximal causal inference (PCI) (Miao et al., 2018; Tchetgen Tchetgen et al., 2020; Cui et al., 2024) provides a semiparametric framework that identifies causal effects using treatment-inducing and outcome-inducing proxy variables together with bridge functions, without explicitly recovering the latent confounder itself.

In contrast, our work does not follow the proximal bridge-function framework. Rather than identifying causal effects directly from observable proxy relations, our objective is to recover an individualized latent confounder representation from multiple noisy proxy measurements. This setting is more closely related to recent identifiable representation learning approaches. Several VAE-based models, including CEVAE (Louizos et al., 2017), TEDVAE (Zhang et al., 2021), CEMVAE (Wu et al., 2024), InfoVAE (Zhao et al., 2019), and InfoCEVAE (Harada & Kashima, 2024), aim to learn latent confounders from proxies, while GP-based alternatives such as the Sequential Deconfounder and Structured GP Confounder (Kuzmanovic et al., 2021; Hatt

& Feuerriegel, 2024; Witty et al., 2020) provide additional modelling flexibility. However, these approaches generally lack structural identifiability guarantees linking the recovered latent representation to the underlying confounding mechanism. While InfoVAE (Zhao et al., 2019) and InfoCEVAE (Harada & Kashima, 2024) encourage informative latent representations via mutual information objectives, they do not establish identifiability of the latent confounder from proxy observations.

Our method establishes structural identifiability. Under nonlinear and injective proxy mappings, we show that the latent confounder $U \in \mathbb{R}^d$ is identifiable up to a smooth invertible transformation from $k \geq 2d + 4$ proxy variables. Unlike proximal causal inference, which avoids explicit latent recovery and primarily targets identification of causal functionals, our framework explicitly infers individualized latent confounders for previously unseen individuals from proxy observations. This recovered latent representation then supports individualized treatment effect (ITE) estimation.

Another related line of work in identifiable representation learning, including nonlinear ICA (Hyvärinen et al., 2019), identifiable VAEs (Khemakhem et al., 2020), and causal-effect VAEs such as (Wu & Fukumizu, 2021), exploits the principle that sufficiently rich variability in conditional distributions can render latent representations identifiable up to an invertible transformation. These approaches build on the idea that when an auxiliary or modulation variable induces sufficiently diverse changes in conditional distributions (e.g. $p(Z \mid U)$), the latent representation can be uniquely determined up to an invertible mapping. This forms a necessary ingredient within our broader identifiability result for proxy-based latent confounder recovery. However, despite this conceptual similarity, the identification problems addressed in these works differ fundamentally from ours. In nonlinear ICA and related identifiable generative models, the auxiliary or modulation variable driving the variability is *observed*. Identifiability arises because the entire conditional distribution of the latent variables changes in sufficiently rich ways as the observed auxiliary variable varies. For each fixed value of the auxiliary variable, the model generates a full distribution over the latent components, and the observed modulation enables recovery of the latent sources.

In contrast, our setting concerns proxy-based latent confounder recovery for causal inference. Here the latent confounder $U$ is *unobserved* and must be inferred from multiple noisy proxy variables $\{Z_i\}_{i=1}^k$ generated from $U$. The goal is not source separation but the recovery of latent confounders for personalised treatment effect estimation. In this setting the variability condition is necessary but not sufficient: because $U$ itself is hidden, one must additionally determine how many conditionally independent proxy variables are required to recover $U$ from its measurements. Our Theorem 1 provides precisely such a result, establishing a proxy-count sufficient condition together with structural conditions under which the latent confounder becomes identifiable from noisy proxies. Thus, while our work is conceptually related to identifiable generative modelling through the shared variability principle, it addresses a distinct identification problem: recovering latent confounders from proxy variables to enable individualised causal effect estimation under hidden confounding. Existing identifiable representation learning approaches do not provide such proxy-based identifiability guarantees.

### 3.2 GP-VAE and Uncertainty-Aware Estimation

Gaussian Processes (GPs) offer a flexible, nonparametric framework for uncertainty modeling (Rasmussen & Williams, 2006). The foundational work by (Casale et al., 2018) introduced GP-VAEs, replacing the isotropic Gaussian prior in VAEs with a GP prior to induce structured latent representations that vary smoothly with the input. Subsequent work extended this idea to sequential data (Fortuin et al., 2020) and latent confounder trajectories (Hatt & Feuerriegel, 2024). In the Structured GP Confounder model (Witty et al., 2020), separate GPs are used to model treatment and outcome given latent confounders. Our proposed IntervalGP-VAE introduces an interval-valued GP prior over the latent space to capture both structured dependencies and epistemic uncertainty in confounder recovery from noisy proxies. For outcome prediction, we incorporate a GP head trained on interval-valued targets to produces calibrated predictive uncertainty for counterfactual queries. To the best of our knowledge, this is the first model to integrate GP-based latent inference with interval-valued uncertainty propagation for ITE estimation.

### 3.3 Interval-Valued Gaussian Processes

Modeling interval-valued outputs with GPs has received limited attention in the literature. To the best of our knowledge, the only existing work that explicitly supports interval observations in a GP setting is the Generalized Multi-Output Censored GP model by (Gammelli et al., 2020; 2022). Their framework introduces a likelihood formulation capable of handling output intervals across multiple outputs. However, their method is designed for multi-output regression tasks and does not address causal inference, latent confounders, or proxy variables. Our approach supports interval supervision of the latent space and, through a GP, interval prediction of individual treatment effects. This allows us to model both uncertainty in confounder inference and outcome prediction in a principled and calibrated manner.

### 3.4 Cross-world dependence and SCM

Cross-world dependence is explicitly modeled through a structural causal model. There are multiple equivalent ways to formalise cross-world dependence in causal inference. For example, the work of Oganisian et al. (Oganisian, 2025) adopts a *direct modeling* approach, in which the joint distribution of $(Y(1), Y(0))$ is specified explicitly, e.g., via a parametric family such as a bivariate normal distribution with a dependence parameter. In contrast, our work adopts the SCM framework (Pearl, 2009), which provides a principled and widely used alternative to direct joint modeling. Under this formulation, the joint distribution of $(Y(1), Y(0))$ is not specified directly, but is instead *induced* through shared latent variables. Thus, cross-world dependence arises naturally from the fact that both potential outcomes are generated using the same realization of $(U, \varepsilon)$. We emphasise that both direct joint modeling approaches (e.g., Oganisian et al. (Oganisian, 2025)) and SCM-based approaches recognise that individual treatment effects cannot be recovered from marginal distributions alone, but depend critically on cross-world dependence between potential outcomes. The difference lies in how this dependence is modeled: directly via parametric assumptions on $(Y(1), Y(0))$, or indirectly via structural equations and latent variables. In practice, both approaches introduce inductive biases, for example, Gaussianity and linearity in direct modeling, Oganisian et al. (Oganisian, 2025), versus structural assumptions such as ANM in SCM. Our work adopts the latter approach and complements it with identifiability analysis for latent confounders.

Our modeling choice is aligned with a large body of existing work on individual-level causal inference. In particular, methods such as CEVAE (Louizos et al., 2017) and TEDVAE (Zhang et al., 2021) adopt latent-variable formulations in which unobserved confounders are inferred from proxy variables and subsequently used for treatment effect estimation. Further, our use of an additive noise model is a standard structural assumption in causal inference. ANM simplifies the functional form of the structural equation and has been widely used in both causal discovery and causal effect estimation due to its favorable identifiability properties.

## 4 Identifiability Analysis

### 4.1 Identifiability of the latent confounder from noisy proxies

**Definition 1** (Identifiability of the latent confounder for ITE estimation)**.** Under the structural equations (Eqs. 1–4), the latent confounder $U$ is said to be *identifiable for ITE estimation* from the observational distribution $p(Z, T, Y)$ if there exists a measurable mapping $\phi : \mathbb{R}^k \to \mathbb{R}^d$ such that the recovered representation $\hat{U} = \phi(Z) = h(U)$ for some smooth and invertible transformation $h : \mathbb{R}^d \to \mathbb{R}^d$, and the individual treatment effect (Eq. 3) is unchanged under this transformation:

$$\tau^{(j)}(h(U), Z_Y^{(j)}) = \tau^{(j)}(U, Z_Y^{(j)}) \qquad \text{for all individuals } j.$$

Thus identifiability requires recovering $U$ up to an invertible reparameterisation that preserves the ITE.

The purpose of Definition 1 is to formalise the notion of identifiability that is necessary for ITE estimation. The primary objective of this work is the recovery of ITEs, for which the latent variable $U$ serves as an intermediate representation enabling personalised causal effect estimation. Definition 1 is formulated to

capture precisely the level of identifiability needed for personalised causal effect estimation: the latent state is identifiable up to an invertible transformation, and this degree of identifiability is fully sufficient for ITE estimation.

Thus, $U$ need not be identifiable in an absolute sense; it is sufficient that the aspects of $U$ required for computing the ITE in Eq. 3 are uniquely recoverable from the proxy variables up to an invertible change of coordinates. In our framework, the latent confounder $U$ is an intermediate representation. Because Definition 1 implies that $p(Z)$ identifies $U$ up to a smooth invertible transformation under which the ITE is invariant, recovering any element of this equivalence class is sufficient. Our focus is thus on causal identifiability rather than exact reconstruction of $U$.

In the ideal case where $U$ is directly observable, the structural outcome equation (Eq. 1) yields identifiable causal effects under standard assumptions. However, in realistic settings $U$ is unobserved and must be inferred from proxy variables. Thus, the central question becomes: under what conditions can $U$ be recovered from proxies (up to a smooth and invertible transformation), thereby ensuring identifiability of causal effects? Theorem 1 provides the formal conditions required to achieve this.

---

**Theorem 1** (Identifiability of the latent confounder from noisy proxies)

Consider the proxy–latent relationship defined in Eq. 4. Suppose the proxy variables satisfy conditional independence (Eq. 5), the variability condition (Eq. 6, 7), and the bounded completeness condition (Eq. 8). A sufficient condition for identifiability is

$$k \geq 2d + 4.$$

Under these conditions, the latent confounder $U \in \mathbb{R}^d$ is identifiable from the observed proxy distribution $p(Z)$ up to a smooth, invertible reparameterisation.

---

*Proof.* See Appendix A for the proof of Theorem 1. □

The requirement on the number of proxies reflects the amount of independent information needed to resolve the latent confounder. Because the confounder lives in a $d$-dimensional space, sufficiently many independent proxy channels are required to distinguish between different latent states. Once the number of proxies exceeds this threshold, the joint proxy distribution contains enough independent variation to uniquely determine the latent confounder structure (up to a smooth invertible transformation). This is precisely the regime captured by the condition $k \geq 2d + 4$ in Theorem 1.

The key reason the latent confounder becomes identifiable from the proxy distribution relies on the combination of two structural properties: conditional independence of the proxies given the latent variable (Eq. 5), and sufficient variability of the proxy responses across latent states (Eq. 7), and the bounded completeness condition (Eq. 8).

First, conditional independence means that once the latent confounder is fixed, each proxy behaves as an independent observation channel of the same hidden state. In other words, the proxies provide multiple independent views of the latent confounder. This structure allows information about the latent state to accumulate across proxies: each proxy contributes partial information, and the joint behaviour of several proxies becomes progressively more informative about the underlying latent variable.

Second, the variability assumption ensures that different latent states induce genuinely different proxy distributions. Intuitively, this condition guarantees that changes in the latent confounder produce observable changes in the proxy behaviour. If the proxy distributions were identical across latent states, the proxies would carry no information about the latent variable and identifiability would be impossible. The variability condition therefore ensures that the mapping from the latent confounder to the proxy distribution is injective at the level of conditional distributions.

Third, the bounded completeness assumption extends this finite-anchor identifiability result to the continuous latent setting. Specifically, bounded completeness guarantees that no nonzero bounded function of the latent

confounder is annihilated by the proxy kernel ($p(Z \mid U)$). Equivalently, the conditional expectation operator induced by the proxy mechanism is injective. This property ensures that distinct mixing distributions over the latent confounder cannot generate the same observed proxy distribution through the same proxy kernel.

When these three properties are combined, the joint proxy distribution becomes highly structured. Conditional independence provides multiple independent views of the latent state, the variability condition ensures that these views change sufficiently across latent values to satisfy the finite-anchor Kruskal conditions, and bounded completeness guarantees that the resulting finite-anchor identifiability extends to the continuous latent setting. Consequently, the latent confounder responsible for generating the observed proxy distribution becomes uniquely determined up to a smooth reparameterisation.

The finite-anchor construction fixes the number of anchor latent states to $r = d + 1$. This choice is motivated by the intrinsic dimensionality of the latent space $U \in \mathbb{R}^d$. In particular, $d+1$ distinct points are the minimal number required to form a non-degenerate affine configuration in a $d$-dimensional space. The proof therefore uses arbitrary anchor sets of size $d + 1$ as local finite representations of the continuous latent geometry.

Notably, the proof does not claim that this condition is necessary or tight. In particular, weaker conditions may still permit identifiability under stronger structural assumptions on the proxy mechanisms, noise model, or latent geometry. The present theorem instead establishes a conservative sufficient condition guaranteeing identifiability of the continuous latent confounder up to smooth invertible reparameterisation.

Note the theorem is stated relative to the proxy-based causal structure already specified in the manuscript through the structural equations and the corresponding SCM (Equation 1 and 4). Our work studies the proxy-based latent-confounding setting, where a causal structure linking $U$, $T$, $Y$, and $Z$ is assumed - see Figure 1, and then derives additional conditions under which recovery of $U$ and hence ITE estimation become possible. In our setting, the observed variables $Z$ are not arbitrary covariates, but noisy proxy measurements of an underlying latent confounder $U$. Without such a structural relationship, the notion of a "proxy" is not well defined, and there is no basis for recovering a latent confounder from observed variables. This is standard in proxy-based methods, including latent-variable approaches such as CEVAE (Louizos et al., 2017) and TEDVAE (Zhang et al., 2021), as well as more broadly in SCM-based causal inference.

In general, conditioning on $Z$ alone does not remove confounding. In fact, this is precisely the reason additional conditions are needed. The role of Theorem 1 is not to justify adjustment on $Z$, but to establish when the latent confounder $U$ can be recovered from the proxy variables $Z$ under further assumptions, namely conditional independence (Equation 5), variability (Equation 6 and 7), bounded completeness (Equation 8) and a sufficient number of proxies. Once $U$ is recovered (up to invertible reparameterisation), individual-level counterfactual inference becomes possible. This is different from sensitivity-analysis approaches based on confounding functions (e.g., (Brumback et al., 2004)) are not directly applicable in our setting. More broadly, they reflect a preference for avoiding explicit assumptions about latent causal structure. While this may be appropriate for population-level sensitivity analysis, it is insufficient for proxy-based ITE estimation: one cannot simply take an arbitrary set of observed variables $Z$ and infer a latent confounder from them without specifying $Z$ are structurally related to (hence the proxies of) $U$.

More details can be found in Appendix A

## 4.2 ITE invariance under latent reparameterisation

The central goal is individual treatment effect (ITE) estimation, for which recovering the latent confounder $U$ is a necessary but not final step. From Theorem 1, the latent confounder is identifiable from proxy variables only up to an invertible transformation. This naturally raises the question of whether such latent ambiguity affects the personalised causal quantity of interest. Let $h : \mathcal{U} \to \mathcal{U}'$ be a reparameterisation mapping and define

$$\hat{U} = h(U). \tag{9}$$

We assume that:

(a) **Invertibility** The mapping $h$ is a smooth bijection (diffeomorphism), so that the original latent variable can be uniquely recovered through

$$U = h^{-1}(\hat{U}).$$

This condition guarantees that the transformed latent representation preserves the underlying structural information required for counterfactual reasoning.

(b) **Bi-Lipschitz regularity of $h$.** The mapping $h$ and its inverse $h^{-1}$ are bi-Lipschitz continuous. That is, there exist constants $L_h, L_{h^{-1}} > 0$ such that

$$\|h(u_1) - h(u_2)\|_2 \leq L_h \|u_1 - u_2\|_2,$$

and

$$\|h^{-1}(u_1') - h^{-1}(u_2')\|_2 \leq L_{h^{-1}} \|u_1' - u_2'\|_2.$$

This regularity condition is used later for stability and error propagation analysis of the learned outcome model.

Evaluating the structural equation (Eq. 1) at the same exogenous noise realisation $\epsilon_Y^j$ yields

$$f(t, U^j, Z_Y^j) = f\left(t, h^{-1}(\hat{U}^j), Z_Y^j\right), \qquad t \in \{0, 1\}.$$

Thus, by the definition of the potential outcomes in Eq. 3 and the invertibility of $h$ above,

$$
\begin{aligned}
\tau^j(\hat{U}^j, Z_Y^j) &= f\left(1, h^{-1}(\hat{U}^j), Z_Y^j\right) - f\left(0, h^{-1}(\hat{U}^j), Z_Y^j\right) \\
&= f(1, U^j, Z_Y^j) - f(0, U^j, Z_Y^j) \\
&= \tau^j(U^j, Z_Y^j).
\end{aligned}
$$

Hence the individual treatment effect satisfies

$$\tau^j(\hat{U}^j, Z_Y^j) = \tau^j(U^j, Z_Y^j), \qquad \text{for all individuals } j.$$

Namely, the ITE is invariant under smooth invertible reparameterisations of the latent space. Consequently, although the latent confounder is only identifiable up to an equivalence class of invertible transformations, the induced personalised treatment effects remain uniquely defined. Thus all admissible latent representations yield identical causal conclusions at the level of ITE estimation.

## 5 IntervalGP-VAE

### 5.1 Unified Bayesian Formulation of IntervalGP-VAE

#### 5.1.1 Design Motivation of IntervalGP

The proposed IntervalGP is designed specifically for interval-valued latent inference. Its design is motivated by the observation that latent-variable causal inference differs significantly from standard Gaussian process regression. In conventional GP regression, both the inputs and the corresponding function values are directly observed, and learning proceeds by maximizing the GP marginal log-likelihood over the joint covariance matrix of all observations. In the present problem, however, the learning objective is very different. As defined in the problem setting of Section 2, the proposed causal model consists of two coupled generative processes: the structural proxy model (Eq. 4), which generates the observed proxy variables from the latent confounder, and the structural outcome model (Eq. 1), which generates the observed outcome from the latent confounder, treatment, and observed outcome covariates. Consequently, the objective is not to regress

directly observed function values, but rather to recover the latent confounder from its observed proxies and subsequently estimate the individualized outcome using the recovered latent representation.

Accordingly, the IntervalGP component is introduced to regularise the inverse recovery of the latent confounder from the observed proxies. The proposed IntervalGP therefore models the latent recovery function $F : \mathbb{R}^k \to \mathbb{R}^d$, which maps the observed proxy variables $Z$ to the latent confounder representation $U$. Since this recovery function is defined solely through the proxy-generation mechanism, the GP naturally takes the observed proxy variables $Z$ as its input. The outcome variable $Y$ is not part of the latent recovery function itself; instead, it enters the framework through the causal likelihood, guiding the learning of the latent representation during variational inference. Consequently, the proposed framework models the posterior latent recovery $p(U \mid Z)$, rather than directly constructing a GP posterior of the form $p(U \mid Z, Y)$. This design preserves the interpretation of the GP as modelling proxy-based latent recovery, while allowing the outcome information to influence the recovered latent representation through the causal decoder.

Another distinction arises from the fact that the latent recovery function is never directly observed. Unlike standard GP regression, where the function values are available as observed outputs, here the latent function values remain hidden. Consequently, the standard GP marginal likelihood cannot be optimized directly. Instead, the variational framework introduces the proxy reconstruction likelihood (i.e.the first term in the objective function), and the causal outcome likelihood (i.e. the second term in the objective function), which follow naturally from the two generative processes defined in Section 2. These likelihood terms provide the information required to infer the latent representation, while the GP prior regularises the latent recovery function. The resulting variational objective is presented in Eq. 16.

Furthermore, since the encoder produces interval-valued posterior summaries, the IntervalGP prior is reformulated in terms of interval containment probabilities. Rather than maximizing the standard GP marginal likelihood for point-valued observations, the IntervalGP prior evaluates the probability that each latent function value lies within its corresponding encoder interval. Consequently, the GP regularisation is defined pointwise over the interval-valued latent observations, and the training objective depends only on the marginal GP distributions at individual proxy observations rather than the full cross-sample covariance matrix. The corresponding interval-GP regularisation is defined in Eq. 17.

The role of the full GP covariance is instead reserved for posterior prediction. Although the off-diagonal covariance terms do not enter the training objective, they are fully exploited during inference to interpolate the latent recovery function for previously unseen proxy observations. This interpolation is justified by the smoothness assumption on the posterior recovery map $m(z) = \mathbb{E}[U \mid Z = z]$, introduced in the *Smoothness of the posterior recovery map* in Section 5.1.2, which states that nearby proxy observations induce nearby posterior latent representations.

Next, we present the details of InterGP-VAE.

### 5.1.2 IntervalGP-VAE

**Vector-valued latent function prior.** Since the latent confounder satisfies $U^i \in \mathbb{R}^d$, the GP prior is defined over a vector-valued latent recovery function. In the current realization, we model each latent dimension independently using a scalar-output GP:

$$F_r \sim \mathcal{GP}(0, k(\cdot, \cdot)), \qquad r = 1, \ldots, d, \tag{10}$$

and define

$$U^i = F(Z^i) = \left( F_1(Z^i), \ldots, F_d(Z^i) \right). \tag{11}$$

For a finite collection $\{Z^i\}_{i=1}^n$, each latent dimension induces the finite-dimensional Gaussian marginal

$$U_r = (U_r^1, \ldots, U_r^n) \sim \mathcal{N}(0, K_Z), \qquad r = 1, \ldots, d, \tag{12}$$

where

$$(K_Z)_{ij} = k(Z^i, Z^j). \tag{13}$$

**Variational posterior and interval-valued latent representation.** The variational encoder defines an amortized approximation to the latent posterior:

$$q_\phi(U^i \mid Z^i) = \mathcal{N}\left(\mu_u^i, (\sigma_u^i)^2\right), \tag{14}$$

where

$$\mu_u^i = \mu_\phi(Z^i) \in \mathbb{R}^d, \qquad \sigma_u^i = \sigma_\phi(Z^i) \in \mathbb{R}^d.$$

The encoder posterior induces the interval-valued latent representation

$$I_u^i = \left[\mu_u^i - \sigma_u^i, \mu_u^i + \sigma_u^i\right] \subseteq \mathbb{R}^d, \tag{15}$$

where the interval is interpreted componentwise. The interval $I_u^i$ is a deterministic uncertainty summary derived from the variational posterior in Eq. 14, rather than an additional latent random variable.

**Likelihood terms.** The decoder defines the proxy reconstruction likelihood

$$p_\theta(Z^i \mid U^i),$$

where $\theta$ denotes decoder parameters and $\epsilon^i$ denotes additive latent noise variables. The outcome model defines

$$p_\psi(Y^i \mid T^i, U^i, Z_Y^i),$$

where $\psi$ denotes the outcome model parameters.

**IntervalGP regularisation and ELBO.** The exact posterior over latent confounders is intractable. We therefore optimize an approximate variational objective. For a single observation,

$$\mathcal{L}^i = \mathbb{E}_{q_\phi(U^i \mid Z^i)}\left[\log p_\theta(Z^i \mid U^i) + \log p_\psi(Y^i \mid T^i, U^i, Z_Y^i)\right] - \mathrm{KL}\left(q_\phi(U^i \mid Z^i) \,\|\, p_{\mathrm{GP}}(U^i \mid Z^i)\right). \tag{16}$$

In IntervalGP-VAE, the GP term is adapted to interval-valued latent observations produced by the encoder. Specifically, for each latent dimension $r$, the interval-GP contribution evaluates the probability that the latent function value lies within this learned interval:

$$\log p_{\mathrm{GP}}\left(U_r^i \in I_{u,r}^i \mid Z^i\right) = \log\left[\Phi\left(\frac{\mu_{u,r}^i + c\sigma_{u,r}^i - \mu_{\mathrm{GP},r}^i}{\sigma_{\mathrm{GP},r}^i}\right) - \Phi\left(\frac{\mu_{u,r}^i - c\sigma_{u,r}^i - \mu_{\mathrm{GP},r}^i}{\sigma_{\mathrm{GP},r}^i}\right)\right], \tag{17}$$

where $\Phi(\cdot)$ is the standard Gaussian CDF. This interval probability acts as a tractable surrogate for the GP-prior part of the KL regularisation in Eq. 16. The resulting training objective is

$$\mathcal{L} = \sum_{i=1}^{n} \mathbb{E}_{q_\phi(U^i \mid Z^i)}\left[\log p_\theta(Z^i \mid U^i) + \log p_\psi(Y^i \mid T^i, U^i, Z_Y^i)\right] + \lambda \sum_{i=1}^{n} \sum_{r=1}^{d} \log p_{\mathrm{GP}}\left(U_r^i \in I_{u,r}^i \mid Z^i\right).$$

where $\lambda > 0$ controls the strength of the IntervalGP regularisation. The model is not trained by maximizing the standard GP marginal likelihood for point-valued outputs. Instead, the VAE learns interval-valued latent posterior summaries through the encoder, and the IntervalGP term regularises these learned intervals by assigning high probability to plausible latent function values under the GP marginal distribution.

**Role of kernel hyperparameters during training.** The current implementation uses a fixed lengthscale parameter $l$ within the RBF kernel

$$k_\eta(z, z') = \sigma_f^2 \exp\left(-\frac{\|z - z'\|^2}{2\ell^2}\right).$$

In standard GP models, the training objective depends on the full covariance structure $k_\eta(Z^i, Z^j)$, which allows the lengthscale to be learned through cross-sample interactions. By contrast, the current IntervalGP training objective uses only local marginal interval regularisation terms Eq. 17 associated with individual latent observations. Consequently, the training objective depends only on marginal quantities of the form

$k_\eta(Z^i, Z^i)$. Therefore, the kernel lengthscale does not act as a learned training-time smoothness parameter in the current implementation.

Also, unlike standard point-valued GP regression, uncertainty in the proposed framework is already represented through the encoder posterior in Eq. 14, which induces the interval-valued latent representation in Eq. 15. The interval width therefore provides a learned, data-adaptive representation of individual-level latent uncertainty. Consequently, the proposed IntervalGP framework does not require separate learning of GP variance parameters for output uncertainty calibration. In particular, the latent uncertainty width is already learned through the encoder posterior variance $(\sigma_u^i)^2$, so the current realization uses a fixed kernel variance parameter $\sigma_f^2$ and a fixed observation-noise term of the form $\sigma_n^2 I$. In this sense, the encoder-derived interval posterior absorbs the role typically played by output and observation-noise variance parameters in standard point-valued GP regression.

**Smoothness of the posterior recovery map.** For inference-time GP posterior prediction, we assume that the posterior recovery map

$$m(z) = \mathbb{E}[U \mid Z = z]$$

is locally smooth on the support of the observed proxy distribution. This is consistent with the smooth proxy-generation model in Eq. 4. Indeed, under $Z = g(U) + \epsilon_Z$, where $g$ is $C^1$, if the latent density $p_U$ and noise density $p_\epsilon$ are sufficiently smooth and $p_Z(z) > 0$ on the region of interest, then the posterior density

$$p(u \mid z) = \frac{p_\epsilon(z - g(u))\, p_U(u)}{p_Z(z)}$$

varies smoothly with $z$. Furthermore, if differentiation under the expectation integral is justified, then

$$m(z) = \mathbb{E}[U \mid Z = z] = \int u\, p(u \mid z)\, du$$

is locally smooth. Under these regularity conditions, the posterior recovery map is locally Lipschitz on compact regions of the proxy manifold. We express this regularity condition as

$$\|m(z) - m(z')\| \le L_m \|z - z'\|, \tag{18}$$

for nearby proxy values $z, z'$ lying on the proxy manifold induced by Eq. 4. This regularity condition justifies the use of kernel-based GP posterior interpolation during inference, since nearby proxy observations are expected to induce nearby posterior latent representations, enabling stable posterior smoothing between neighbouring observations.

**Latent confounder posterior prediction.** After training, the encoder-derived latent intervals are treated as noisy interval-valued observations of the latent recovery function. For a test proxy vector $Z^*$, full GP posterior conditioning is applied independently to each latent dimension:

$$p(U_r^* \mid Z^*, \mathcal{D}) = \mathcal{N}\left(\mu_{u,r}^*, (\sigma_{u,r}^*)^2\right), \qquad r = 1, \ldots, d, \tag{19}$$

where $\mu_{u,r}^* = K_s^\top K^{-1} \mu_{u,r}, \qquad (\sigma_{u,r}^*)^2 = K_{ss} - K_s^\top K^{-1} K_s$. Here:

- $K \in \mathbb{R}^{n \times n}$ denotes the kernel matrix over training proxy observations,

- $K_s \in \mathbb{R}^{n \times 1}$ denotes the cross-covariance vector between training inputs and the test input $Z^*$,

- and $K_{ss}$ denotes the prior variance at $Z^*$.

The GP posterior therefore induces the interval-valued latent posterior set:

$$I_u^* = I_{u,1}^* \times \cdots \times I_{u,d}^* \subseteq \mathbb{R}^d,$$

**ITE posterior prediction via interval propagation.** The posterior latent interval $I_u^*$ is propagated through the outcome model to obtain uncertainty-aware counterfactual outcome intervals:

$$Y^*(1) = f_\psi(1, U^*, Z_Y^*), \qquad Y^*(0) = f_\psi(0, U^*, Z_Y^*),$$

The posterior latent interval $I_u^*$ is propagated through the causal outcome model, inducing the interval-valued posterior ITE representation

$$\hat{\tau}_{\text{GP}}^* = \left[ \inf_{U^* \in I_u^*} \left( f_\psi(1, U^*, Z_Y^*) - f_\psi(0, U^*, Z_Y^*) \right), \right.$$

$$\left. \sup_{U^* \in I_u^*} \left( f_\psi(1, U^*, Z_Y^*) - f_\psi(0, U^*, Z_Y^*) \right) \right], \tag{20}$$

$$(\sigma_\tau^*)^2 = K_{ss} - K_s^\top K^{-1} K_s.$$

The interval bounds correspond to the minimum and maximum counterfactual treatment effects over admissible latent values satisfying $U^* \in I_u^*$, thereby propagating interval-valued latent posterior uncertainty through the causal outcome model. The accompanying variance term $(\sigma_\tau^*)^2$ quantifies predictive uncertainty associated with the GP latent recovery function during posterior smoothing.

**Summary of the probabilistic framework.** The Bayesian components of the proposed IntervalGP–VAE are summarised as follows.

- **Observed data.** The observed data are

$$\mathcal{D} = \{(Z^i, T^i, Y^i)\}_{i=1}^n,$$

  whereas the latent variables are the unobserved confounders

$$\{U^i\}_{i=1}^n.$$

- **Prior.** The prior is the IntervalGP prior placed on the latent recovery function

$$F : \mathbb{R}^k \to \mathbb{R}^d,$$

  as defined in Eq. 17, which regularises the recovery of latent confounders from the observed proxy variables.

- **Likelihood.** The likelihood consists of the proxy reconstruction likelihood

$$p_\theta(Z \mid U)$$

  and the causal outcome likelihood

$$p_\psi(Y \mid T, U, Z_Y),$$

  corresponding to the first and second likelihood terms in Eq. 16. These likelihoods follow directly from the structural proxy model and structural outcome model introduced in Section 2.

- **Posterior approximation.** We approximate the latent posterior

$$p(U \mid Z)$$

  by the variational posterior

$$q_\phi(U \mid Z),$$

  through the variational objective in Eq. 16. During inference, the encoder posterior summaries are further described in Eq. 19 and Eq. 20

- **Model parameters.** The trainable model parameters are the encoder parameters $\phi$, decoder parameters $\theta$, and causal prediction parameters $\psi$.

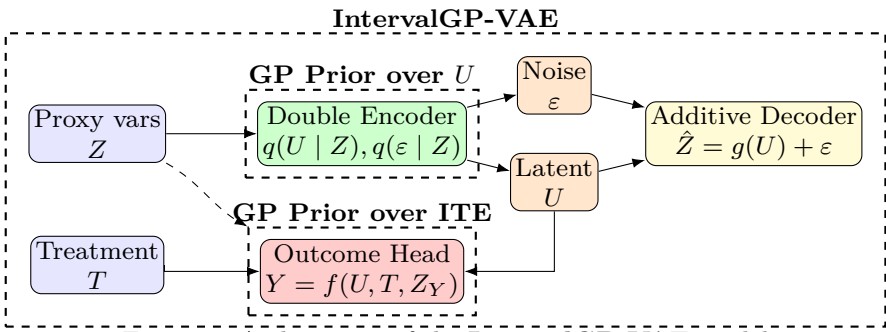

Figure 2: Architecture of the **IntervalGP-VAE** model.

## 5.2 Compatibility of IntervalGP Posterior Inference with Identifiability

Theorem 1 establishes identifiability of the latent recovery problem under the structural proxy assumptions. Specifically, it characterises when the latent confounder can be recovered from the proxy distribution up to the equivalence class induced by smooth invertible reparameterisations. In practice, however, the underlying distribution is not observed directly. Instead, we observe a finite sample $\{(Z^i, T^i, Y^i)\}_{i=1}^n$, and seek to recover individual latent representations through posterior inference. The role of IntervalGP-VAE is therefore to provide a finite-sample approximate inference procedure for learning posterior latent representations $q_\phi(U^i \mid Z^i)$ from observed data. Within this framework, the IntervalGP component acts as an interval-valued latent prior/regularisation mechanism during training and as a posterior smoothing mechanism during inference, rather than as an additional structural assumption required for identifiability. Proposition 1 formalises that introducing this IntervalGP-based inference prior does not alter the identifiability result of Theorem 1 and remains compatible with the latent reparameterisation freedom implied by the identifiable equivalence class.

---

**Proposition 1 (Compatibility of IntervalGP priors with identifiability and latent reparameterisations)**

Let $U \in \mathbb{R}^d$ denote the latent confounder satisfying the assumptions of Theorem 1, and let

$$F_r \sim \mathcal{GP}(0, k), \qquad r = 1, \dots, d,$$

define the vector-valued latent recovery functions used by the IntervalGP framework. Then:

1. Introducing IntervalGP-based latent priors does not alter the identifiability of the individual treatment effect (ITE) established in Theorem 1.

2. For the reparameterisation $h : \mathbb{R}^d \to \mathbb{R}^d$, defined in Eq. (8), replacing $U$ by $\tilde{U} = h(U)$ does not affect the validity of $k$ as a covariance kernel over the latent representation.

---

*Proof.* See Appendix B for the detailed proof of Proposition 1. □

The practical role of the IntervalGP component is therefore distinct from the identifiability analysis. Theorem 1 establishes when latent confounder recovery is theoretically possible from proxies, while the IntervalGP framework provides a probabilistic mechanism for posterior smoothing and uncertainty propagation over latent representations within the identifiable equivalence class.

### 5.3 IntervalGP-VAE Framework

#### 5.3.1 Model Architecture

The proposed *IntervalGP-VAE* extends the standard Variational Autoencoder (VAE) framework by incorporating structured Gaussian Process (GP) priors over both the latent confounder space $U$ and the Individual Treatment Effect (ITE) space. This architecture, conceptualized in Fig. 2, enables the model to yield both accurate point estimates and calibrated uncertainty intervals for latent variables and causal effects. The core components are structured as follows:

- **Encoder:** Maps each proxy input $Z^i$ to a variational posterior over $U$, defining a latent interval $u^i \in [\mu_u^i - \sigma_u^i, \; \mu_u^i + \sigma_u^i]$ via Eq. 15 to explicitly capture epistemic uncertainty within the latent representation.

- **IntervalGP over $U$:** A GP prior that regularizes the latent space by maximizing the interval-based log-likelihood $\log p_{\mathrm{GP}}(\mu_u \pm \sigma_u \mid Z)$ as specified in Eq. 17.

- **Decoder:** Reconstructs the proxy variables via $\hat{\mathbf{z}} = g_{\mathrm{dec}}(u, \epsilon)$ using sampled latent variables and additive noise.

- **Outcome Head:** Predicts the potential outcomes $y$ from the latent confounder $u$ and treatment $t$, thereby facilitating ITE estimation.

- **IntervalGP over ITE:** A secondary GP regressor tasked with predicting calibrated ITE intervals, effectively propagating inference uncertainty from the latent space to the final causal effect estimation.

#### 5.3.2 Staged Optimization

The model is trained using the objective introduced in Section 5.1, which combines proxy reconstruction, outcome prediction, variational regularisation, and IntervalGP latent regularisation. To stabilize latent confounder recovery, enhance outcome prediction accuracy, and streamline uncertainty propagation within the same objective family, the model is optimized in three consecutive stages:

1. **Joint Training:** Jointly optimize the VAE components and the causal head using the full loss $\mathcal{L}_{\mathrm{IntervalGP\text{-}VAE}}$.

2. **Causal-Head Tuning:** Freeze the VAE parameters and exclusively fine-tune the causal head to sharpen outcome predictions.

3. **VAE Refinement:** Freeze the causal head and refine the VAE parameters to achieve tighter alignment of the latent structures.

The training procedure is summarised in Algorithm 1. Implementation-specific parameters, weights, and hyperparameter configurations are detailed in the Experiments section.

### 5.4 Computational complexity

Although exact GP inference scales as $\mathcal{O}(n^3)$, the practical cost in our model is substantially lower. The interval-GP prior operates in the low-dimensional latent space, and all kernel evaluations, Cholesky factorizations, and predictive computations are performed on minibatches (128 in all experiments), yielding small kernel matrices and inexpensive decomposition steps. As a result, the computational overhead is comparable to standard VAE training. For higher-dimensional latent spaces or larger latent sets, the framework supports sparse GP approximations, such as inducing-point or structured, kernel methods—which reduce complexity

---

**Algorithm 1:** IntervalGP–VAE Algorithm

---

**Input:** Training set $\mathcal{D}_{\text{train}} = \{(Z^i, T^i, Y^i)\}_{i=1}^n$, test set $\mathcal{D}_{\text{test}} = \{Z^j\}_{j=1}^m$, training epochs $(E_{\text{joint}}, E_{\text{head}}, E_{\text{vae}})$

**Output:** Estimated ITEs $\{\hat{\tau}^j\}_{j=1}^m$ and interval estimates $\{[\hat{\tau}_{\text{lower}}^j, \hat{\tau}_{\text{upper}}^j]\}_{j=1}^m$

**1 Initialize** encoder $q_\phi(U \mid Z)$, decoder $p_\theta(Z \mid U)$, and causal head $f_\psi(T, U, Z_Y)$;

**2 Stage 1: Joint Training (VAE and Causal Head)**;

**3 for** epoch $= 1$ **to** $E_{\text{joint}}$ **do**

**4**      **for** *each mini-batch* $\{(Z^i, T^i, Y^i)\}$ **do**

**5**          Infer latent posterior $q_\phi(U \mid Z)$ and sample latent representations;

**6**          Reconstruct proxies $\hat{Z} \sim p_\theta(Z \mid U)$;

**7**          Predict outcomes $\hat{Y} = f_\psi(T, U, Z_Y)$;

**8**          Compute the reconstruction term, outcome term, and IntervalGP regularisation term;

**9**          Compute the total objective and update $(\phi, \theta, \psi)$;

**10**      **end**

**11 end**

**12 Stage 2: Causal Head Refinement**;

**13 for** epoch $= 1$ **to** $E_{\text{head}}$ **do**

**14**      Update the causal head while keeping the latent representation fixed;

**15 end**

**16 Stage 3: Latent Representation Refinement**;

**17 for** epoch $= 1$ **to** $E_{\text{vae}}$ **do**

**18**      Refine the latent representation while keeping the causal head fixed;

**19 end**

**20 Post-training inference**;

**21 for** *each test proxy observation* $Z^j$ **do**

**22**      Compute the IntervalGP posterior for the latent confounder to obtain the latent interval $I_u^j$;

**23**      Propagate $I_u^j$ through the causal outcome model to obtain the posterior ITE estimate $\hat{\tau}^j$ and interval $[\hat{\tau}_{\text{lower}}^j, \hat{\tau}_{\text{upper}}^j]$ with predictive uncertainty;

**24 end**

**25 return** $\{\hat{\tau}^j\}_{j=1}^m$ *and* $\{[\hat{\tau}_{\text{lower}}^j, \hat{\tau}_{\text{upper}}^j]\}_{j=1}^m$;

---

to $\mathcal{O}(m^3)$ with $m \ll n$. These approximations do not affect our identifiability results, which rely on proxy structure rather than the specific GP implementation.

Notably, in standard Gaussian Process regression, the full kernel matrix across all training inputs is essential, and minibatching is nontrivial. However, our model employs an *interval-valued GP prior*, which is conceptually and mathematically distinct from the posterior inference typically associated with GP regression.

Specifically, as defined in Equation (10), our prior imposes a structured regularization over the encoder output $U \sim q(U \mid Z) = \mathcal{N}(\mu_u(Z), \sigma_u^2(Z))$, treating each latent variable $u_i$ as lying within the interval $[\mu_u^i - \sigma_u^i, \ \mu_u^i + \sigma_u^i]$. The GP prior then computes the marginal likelihood of each interval under the GP prior $U(Z) \sim \mathcal{GP}(0, K(Z, Z))$, and regularizes the encoder by maximizing the total log probability that each $u_i$ lies within this interval. Importantly, the resulting prior term, $\log p_{\mathrm{GP}} \left( u^i \in [\mu_u^i \pm \sigma_u^i] \mid Z^i \right)$, depends only on the marginal GP distribution at each point $Z^i$, i.e., $\mathcal{N}(0, k(Z^i, Z^i))$, and not on cross-point correlations. This makes the prior computation fully local, univariate, and independent across data points, thereby fully compatible with minibatch training. As a result, training complexity is significantly reduced compared to standard GP methods. As such, the resulting regularizer is fully decoupled across samples.

This form of prior emerges when the latent variable is represented not as a point estimate, but as an interval-valued random variable. In such cases, the probabilistic constraint that a latent sample $u_i \in [\mu_u^i - \sigma_u^i, \ \mu_u^i + \sigma_u^i]$ under a GP prior corresponds to evaluating the Gaussian probability mass within that interval. i.e., the difference between two CDF values of a univariate normal distribution centered at zero with variance $k(Z^i, Z^i)$. Since this prior only depends on the marginal variance at each $Z^i$, it avoids the need to evaluate full joint Gaussian densities across data points.

This pointwise structure makes the intervalGP prior *naturally compatible with mini-batch training*. The computational complexity is therefore drastically reduced, and the model can scale to large datasets using stochastic optimization. This is particularly important for practical implementation on high-dimensional proxy data $Z$, and aligns with the goal of enabling uncertainty-aware latent regularization without sacrificing scalability.

We emphasize that this GP prior is used during training for regularization. At test time, we revert to a full posterior GP computation (Equations 11–12), which recovers standard GP properties, such as smoothness, confidence calibration, and extrapolation behavior, by leveraging the full kernel structure. This clarifies that the simplification only affects the training prior, not the inference procedure, ensuring both efficiency and fidelity.

### 5.5 From Structural Identifiability to Finite-Sample Estimation

Theorem 1 establishes that, under the conditional independence, variability, and proxy-count assumptions, the latent confounder $U$ is identifiable from the observed proxy distribution $p(Z)$ up to a smooth invertible reparameterisation $h(U)$. However, practical learning is performed using finite samples and neural function approximators. Therefore, the theoretical identifiability result must be connected to the empirical learning procedure used to infer $\hat{U}$ and estimate individual treatment effects (ITEs).

In this section, we introduce several regularity assumptions commonly used in statistical learning theory and nonlinear latent-variable models, and then establish finite-sample error bounds linking latent recovery, outcome estimation, and ITE estimation.

**Regularity assumptions for finite-sample estimation.** We assume:

- **Compact support of the observed and latent variables.** The latent confounder support $\mathcal{U} \subset \mathbb{R}^d$, the proxy support $\mathcal{Z} \subset \mathbb{R}^k$, and the outcome support $\mathcal{Y} \subset \mathbb{R}$ are compact. Consequently, there exist constants $M_U, M_Z, M_Y > 0$ such that

$$\sup_{u \in \mathcal{U}} \|u\|_2 \leq M_U, \qquad \sup_{z \in \mathcal{Z}} \|z\|_2 \leq M_Z, \qquad \sup_{y \in \mathcal{Y}} |y| \leq M_Y.$$

- **Lipschitz continuity of $f$.** The structural outcome function $f : \{0,1\} \times \mathbb{R}^d \times \mathbb{R}^{k'} \to \mathbb{R}$ is $L_f$-Lipschitz continuous with respect to its latent confounder argument $u$. That is, for all $u_1, u_2 \in \mathcal{U}$, fixed $t \in \{0,1\}$, and fixed $Z_y$,

$$|f(t, u_1, Z_y) - f(t, u_2, Z_y)| \le L_f \|u_1 - u_2\|_2.$$

- **Bounded empirical objective.** Let $\ell_{\mathrm{ELBO}}(Z, Y, T; \phi, \theta, \psi)$ denote the training objective induced by Eq. 16, including the proxy likelihood, outcome likelihood, and IntervalGP regularisation term. We assume that this objective is uniformly bounded, i.e.,

$$-B_{\mathrm{ELBO}} \le \ell_{\mathrm{ELBO}}(Z, Y, T; \phi, \theta, \psi) \le B_{\mathrm{ELBO}}$$

for some constant $B_{\mathrm{ELBO}} < \infty$, uniformly over the data support and parameter space.

This bounded empirical objective assumption is satisfied when $Z$ and $Y$ have bounded support, the encoder–decoder and outcome-head parameter spaces are compact, the likelihood variances are bounded away from zero and infinity, and the IntervalGP interval probability is bounded away from zero.

**Outcome Head Estimation Error**  Given the learned representation $\hat{u}$, the outcome head $f_\psi(t, \hat{u}, Z_y)$ is trained to approximate the structural outcome function $f(t, u, Z_y)$. Over the transformed latent space $u' = h(u)$, the corresponding target function is

$$\tilde{f}(t, u', Z_y) = (f \circ h^{-1})(t, u', Z_y) = f(t, h^{-1}(u'), Z_y). \tag{21}$$

Under the Bi-Lipschitz Regularity of $h$ and the Lipschitz Continuity of $f$, the transformed outcome function remains Lipschitz continuous:

$$|(f \circ h^{-1})(t, u_1', Z_y) - (f \circ h^{-1})(t, u_2', Z_y)| \le L_f L_{h^{-1}} \|u_1' - u_2'\|_2. \tag{22}$$

Hence perturbations in the learned latent representation propagate in a controlled manner through the outcome model. In particular, if the learned representation $\hat{u}$ approximates the transformed latent variable $h(u)$ with small error, then the induced error in the predicted potential outcomes and ITE estimates is correspondingly bounded.

This provides the key connection between latent recovery quality and downstream causal estimation accuracy: stable recovery of the latent confounder under the bi-Lipschitz transformation directly induces stable approximation of the structural counterfactual functions.

**Latent Approximation**  The VAE learning procedure can be viewed as a finite-sample approximation to the population-level proxy recovery problem characterised by Theorem 1. Under the identifiability assumptions of Theorem 1, together with the Bi-Lipschitz Regularity of $h$, the Compact Support assumption, and the Bounded Empirical Objective assumption, standard uniform-convergence arguments based on Rademacher complexity imply that the empirical training objective admits a generalisation gap of order

$$\mathcal{O}(N^{-1/2}).$$

The learned latent representation $\hat{u}$ may be viewed as a finite-sample approximation to the identifiable equivalence class $h(u)$. Specifically, we write

$$\mathbb{E}\|\hat{u} - h(u)\|_2 \le \epsilon_U(N), \tag{23}$$

where $\epsilon_U(N)$ decreases with sample size, model complexity, and optimisation accuracy. Eq. 34 shows the decomposition of $\epsilon_U(N)$.

By the Bi-Lipschitz Regularity of $h$, the identifiable latent representation $h(U)$ preserves the information contained in $U$. Therefore, if $\hat{U}$ approximates $h(U)$ up to the finite-sample error $\epsilon_U(N)$, then $\hat{U}$ acts as an approximate confounding-sufficient representation. In this sense, $\hat{U}$ may be viewed as a finite-sample surrogate for the unobserved confounder $U$.

**ITE Estimation Bound**   Now we show that the ITE estimation error under our setting is bounded. Given the learned representation $\hat{u}$, the outcome head $f_\psi(t, \hat{u}, Z_y)$ is trained to approximate the structural outcome function $f(t, u, Z_y)$. Over the transformed latent space $u' = h(u)$, the corresponding target function is

$$\tilde{f}(t, u', Z_y) = (f \circ h^{-1})(t, u', Z_y) = f(t, h^{-1}(u'), Z_y). \tag{24}$$

For each treatment value $t \in \{0, 1\}$, the outcome prediction error can be decomposed as

$$|f_\psi(t, \hat{u}, Z_y) - f(t, u, Z_y)| \leq |f_\psi(t, \hat{u}, Z_y) - \tilde{f}(t, \hat{u}, Z_y)| + |\tilde{f}(t, \hat{u}, Z_y) - f(t, u, Z_y)|. \tag{25}$$

The first term corresponds to the finite-sample estimation error of the outcome head. We write

$$\mathbb{E}\big[|f_\psi(t, \hat{u}, Z_y) - \tilde{f}(t, \hat{u}, Z_y)|\big] \leq \mathcal{E}_{\text{gen}} + \mathcal{E}_{\text{approx}} + \mathcal{E}_{\text{opt}}, \tag{26}$$

where $\mathcal{E}_{\text{gen}}$ denotes the statistical generalisation error, $\mathcal{E}_{\text{approx}}$ denotes the approximation error of the neural function class, and $\mathcal{E}_{\text{opt}}$ denotes the optimisation error.

The generalisation error satisfies

$$\mathcal{E}_{\text{gen}} \leq \sup_{f \in \mathcal{F}_\Psi} \left|\mathcal{R}(f) - \hat{\mathcal{R}}_N(f)\right|,$$

which upper-bounds the discrepancy between the population risk $\mathcal{R}(f)$ and the empirical risk $\hat{\mathcal{R}}_N(f)$ over the hypothesis class. Under standard bounded-loss and bounded-complexity assumptions on the hypothesis class,

$$\mathcal{E}_{\text{gen}} = O(\mathfrak{R}_N(\mathcal{F}_\Psi)) = O(N^{-1/2}), \tag{27}$$

where $\mathfrak{R}_N(\mathcal{F}_\Psi)$ denotes the Rademacher complexity of the outcome-head hypothesis class.

The approximation error

$$\mathcal{E}_{\text{approx}} = \inf_{f_\psi \in \mathcal{F}_\Psi} \|f_\psi - \tilde{f}\|_\infty$$

reflects the expressive capacity of the outcome-head architecture. For a fixed architecture, this acts as an architecture-dependent bias term. Under standard neural-network approximation theory on compact domains, $\mathcal{E}_{\text{approx}}$ decreases as the model capacity increases, with rates depending on the smoothness of $\tilde{f}$, the specific network class, and the dimension of the transformed latent space. Notably, this term is not a function of the sample size $N$ for a fixed architecture.

The optimisation error

$$\mathcal{E}_{\text{opt}} = \hat{\mathcal{R}}_N(f_\psi) - \min_{f \in \mathcal{F}_\Psi} \hat{\mathcal{R}}_N(f)$$

captures the residual empirical-risk gap after finite optimisation. This term depends on the optimisation algorithm, learning rate, and number of gradient steps. Under standard stochastic first-order optimisation assumptions, it decreases as training progresses and may vanish in the limit of successful optimisation. Consequently, it depends on the choice of optimizer and the number of training iterations rather than directly on $N$.

The second term corresponds to latent recovery error. By definition of $\tilde{f}$,

$$\begin{aligned}
|\tilde{f}(t, \hat{u}, Z_y) - f(t, u, Z_y)| &= |f(t, h^{-1}(\hat{u}), Z_y) - f(t, u, Z_y)| \\
&\leq L_f \|h^{-1}(\hat{u}) - u\|_2 = L_f \|h^{-1}(\hat{u}) - h^{-1}(h(u))\|_2 \\
&\leq L_f L_{h^{-1}} \|\hat{u} - h(u)\|_2,
\end{aligned} \tag{28}$$

where the first inequality follows from the Lipschitz Continuity of $f$, and the second inequality follows from the Bi-Lipschitz Regularity of $h$. Taking expectations and applying Equation equation 23 gives

$$\mathbb{E}[|f_\psi(t, \hat{u}, Z_y) - f(t, u, Z_y)|] \leq \mathcal{E}_{\text{gen}} + \mathcal{E}_{\text{approx}} + \mathcal{E}_{\text{opt}} + L_f L_{h^{-1}} \epsilon_U(N). \tag{29}$$

Finally, the true and estimated ITEs are

$$\tau(u, Z_y) = f(1, u, Z_y) - f(0, u, Z_y), \tag{30}$$

and

$$\hat{\tau}(\hat{u}, Z_y) = f_\psi(1, \hat{u}, Z_y) - f_\psi(0, \hat{u}, Z_y). \tag{31}$$

By the triangle inequality,

$$|\hat{\tau}(\hat{u}, Z_y) - \tau(u, Z_y)| \leq |f_\psi(1, \hat{u}, Z_y) - f(1, u, Z_y)| + |f_\psi(0, \hat{u}, Z_y) - f(0, u, Z_y)|. \tag{32}$$

Applying the previous bound for $t = 1$ and $t = 0$ yields

$$\mathbb{E}[|\hat{\tau}(\hat{u}, Z_y) - \tau(u, Z_y)|] \leq 2\mathcal{E}_{\text{gen}} + 2\mathcal{E}_{\text{approx}} + 2\mathcal{E}_{\text{opt}} + 2L_f L_{h^{-1}} \epsilon_U(N). \tag{33}$$

The latent approximation error in Equation equation 23 can be further decomposed into:

$$\epsilon_U(N) = \mathcal{E}_{\text{gen}}^U + \mathcal{E}_{\text{approx}}^U + \mathcal{E}_{\text{opt}}^U, \tag{34}$$

where the three terms have the same interpretations as those defined above for the outcome-head estimation error.

Thus, the ITE bound can be interpreted as

$$\mathbb{E}[|\hat{\tau}(\hat{u}, Z_y) - \tau(u, Z_y)|] \leq O(N^{-1/2}) + 2\mathcal{E}_{\text{approx}} + 2\mathcal{E}_{\text{opt}} + 2L_f L_{h^{-1}} \left[ O(N^{-1/2}) + \mathcal{E}_{\text{approx}}^U + \mathcal{E}_{\text{opt}}^U \right].$$

Therefore, as $N$ grows, the statistical generalisation components vanish at the usual $N^{-1/2}$ rate, while the remaining terms are controlled by model expressiveness and optimisation quality. This bound shows that the ITE estimation error decomposes into two sources: (i) finite-sample estimation error of the outcome head, and (ii) latent recovery error inherited from the finite-sample approximation of the identifiable latent representation. Consequently, improvements in either latent recovery accuracy or outcome-model estimation directly improve the accuracy of the estimated ITE.

**Relation to latent ignorability.** The latent ignorability assumption $(Y(0), Y(1)) \perp T \mid U$ is a standard assumption in causal inference that enables identification of causal effects once the latent confounder is known. It is important to note that Theorem 1 does *not* rely on this assumption. Rather, Theorem 1 is purely a latent recovery result: it establishes conditions under which the latent confounder can be identified from the observed proxy distribution through the proxy model, conditional independence, variability, and bounded completeness assumptions, without involving the treatment or outcome variables.

The role of latent ignorability arises only in the subsequent causal identification stage. Specifically, Theorem 1 shows that the recovered latent representation is identifiable up to a smooth invertible transformation, namely $\hat{U} = h(U)$. Since $h$ is invertible, conditioning on $\hat{U}$ is theoretically equivalent to conditioning on $U$, and therefore the latent ignorability assumption is preserved at the population level:

$$(Y(0), Y(1)) \perp T \mid U \quad \iff \quad (Y(0), Y(1)) \perp T \mid \hat{U}.$$

In practice, however, the learned latent representation only approximates the population solution because of finite-sample estimation error. This section therefore studies this practical setting by deriving an upper bound on the ITE estimation error as a function of the latent recovery error. This analysis above has quantified how deviations between the estimated latent representation and the population solution propagate to downstream causal effect estimation.

# 6 Experiments

We conduct synthetic, semi-synthetic, and synthetic-to-real experiments to evaluate the effectiveness of our proposed method. All experiments are implemented on an Apple MacBook Pro laptop equipped with an M5 Pro chip (comprising a 15-core CPU and a 16-core GPU), 24 GB of unified memory, and a 1 TB SSD.

Table 2: Treatment mechanisms and proxy/outcome functions used in synthetic experiments.

| | Functions |
|---|---|
| **Proxy Functions** | $\{u,\ u^2,\ u^3,\ \max(1.2u, 0.2u),\ 0.2\sinh(u),\ \mathrm{sgn}(u)\log(1+|u|)\}$, $\{u,\ \sin(u),\ \tanh(0.5u),\ \exp(0.3u),$ $\log(1+u^2),\ \sigma(u)\}$, $\{u,\ u^2,\ \sin(u),\ \tanh(u),\ \exp\!\left(-0.2\,u^2\right),\ \sin(2u)\}$ $\{u,\ u^3,\ \tanh(u),\ \sigma(u),\ \log(1+u^2),\ \sin(u)\}$ $\{u,\ \tanh(0.5u),\ \sin(u),\ \exp\!\left(-0.3\,u^2\right),\ \sigma(u),\ u^2\}$ $\{u,\ \log(1+e^u),\ \exp(0.5u),\ \tanh(u),\ \sigma(u),\ \log(1+u^2)\}$ |
| **Treatment Functions** | $\{\mathrm{Bernoulli}\!\left(\sigma(0.8u)\right)\}$, $\{\mathrm{Bernoulli}\!\left(\sigma(0.4u+0.3)\right)\}$ |
| **Outcome Functions** | $\{0.5u + 0.2\sin(u) + t\,(0.8+0.3u) + \varepsilon\}$, $\{0.5u + 0.3\cos(u) + t\,(0.5+0.5u) + \varepsilon\}$ |

## 6.1 Synthetic Experiments

We first evaluate the proposed *IntervalGP-VAE* framework on synthetic datasets generated from a controlled latent-confounder setting. For each unit, a hidden confounder is sampled via

$$U \sim \mathcal{N}(0,1).$$

To verify the robust performance of our framework under diverse data distributions, we consider 24 distinct synthetic settings. These settings correspond to a $6 \times 2 \times 2$ combination of proxy functions, binary treatment mechanisms, and outcome functions, as summarised in Table 2. Crucially, these settings are deliberately constructed to satisfy the identifiability conditions outlined in Theorem 1. Under this identifiable regime, *IntervalGP-VAE* is expected to recover the unobserved confounder up to an invertible transformation, thereby enabling the faithful propagation of measurement-induced uncertainty from noisy proxy variables to downstream counterfactual outcomes and individual treatment effect (ITE) estimates.

Existing proxy-based latent-confounder methods, such as *CEVAE* (Louizos et al., 2017), *TEDVAE* (Zhang et al., 2021), and *DFPV* (Xu et al., 2021), primarily focus on recovering latent representations or estimating treatment effects as point values. Consequently, they do not directly address whether the interval component introduced in *IntervalGP-VAE* yields additional utility for uncertainty-aware causal inference. This distinction is vital in our setting because noisy and imperfect proxy variables inherently introduce epistemic uncertainty into the recovered latent confounder. As established in our problem formulation, proxy-based identifiability guarantees consistency in the large-sample limit but does not eliminate finite-sample variability or proxy noise. Thus, reliable individual-level decision support demands not only accurate point-wise ITE predictions but also well-calibrated uncertainty quantification (UQ). To evaluate both dimensions, we extend our baseline comparison to encompass both point-estimation and interval-producing methods, structured to answer two complementary research questions:

1. Does *IntervalGP-VAE* maintain competitive point-estimation precision compared with representative latent-confounder and proxy-learning benchmarks?

2. Are its interval estimates more informative and better calibrated than standard uncertainty baselines, rather than merely inflating interval widths to artifactually boost empirical coverage?

To address these questions, we benchmark *IntervalGP-VAE* against five distinct baselines spanning different causal paradigms:

- **Deep Latent and Proxy Methods (Point Estimation):** We include *CEVAE* (Louizos et al., 2017) as a classical VAE-based latent-confounder baseline, and a *DeepPCL*-like representation framework directly related to proxy-based causal learning. We also retain *TEDVAE* (Zhang et al., 2021) as the primary deep latent-variable baseline due to its close relevance to treatment-effect estimation with learned representations.

- **Uncertainty-Aware Baselines (Interval Estimation):** We incorporate a bootstrap tree-ensemble baseline inspired by the nonparametric causal response surface design of BART (Hill, 2011) and BCF (Hahn et al., 2020), alongside a conformalized TARNet baseline (Shalit et al., 2017).

To establish a competitive upper bound for the uncertainty baselines, we implement a split conformal calibration protocol for *TEDVAE* to serve as an *oracle uncertainty baseline*. Formally, let $\hat{\tau}_i^{\mathrm{TEDVAE}}$ denote the point estimate for a sample $i$ in a held-out calibration set $\mathcal{D}_{\mathrm{cal}}$. The corresponding conformity score is defined as

$$s_i = \left| \tau_i - \hat{\tau}_i^{\mathrm{TEDVAE}} \right|. \tag{35}$$

Throughout the experiments, we use nominal 90% ITE intervals. Thus, $1 - \alpha = 0.90$, $\alpha = 0.10$, and two-sided Gaussian intervals use the quantile $z_{1-\alpha/2} = z_{0.95}$. Given this nominal coverage level, the conformal quantile $\hat{q}_{1-\alpha}$ is computed over the calibration scores via

$$\hat{q}_{1-\alpha} = Q_{\lceil (n_{\mathrm{cal}}+1)(1-\alpha) \rceil / n_{\mathrm{cal}}} \left( \{s_i\}_{i=1}^{n_{\mathrm{cal}}} \right). \tag{36}$$

The calibrated conformal ITE interval for a test sample $j \in \mathcal{D}_{\mathrm{test}}$ is subsequently constructed as

$$\left[ \hat{\tau}_j^{\mathrm{TEDVAE}} - \hat{q}_{1-\alpha}, \; \hat{\tau}_j^{\mathrm{TEDVAE}} + \hat{q}_{1-\alpha} \right]. \tag{37}$$

It is critical to note that this conformal calibration procedure relies on the true ITEs ($\tau_i$) within the calibration set, which are accessible exclusively in synthetic environments. Consequently, this protocol must be interpreted as an oracle calibration baseline rather than a directly applicable procedure for real observational studies, where true ITEs are strictly unobserved.

Following standard causal inference literature (Hill, 2011; Shalit et al., 2017; Louizos et al., 2017; Yao et al., 2018), we evaluate point-estimation accuracy across all methods using two metrics on the test set of size $m$:

1. **Precision in Estimation of Heterogeneous Effect (PEHE):** Gauges individual-level estimation accuracy, defined as

$$\mathrm{PEHE} := \sqrt{\frac{1}{m} \sum_{j=1}^{m} (\hat{\tau}_j - \tau_j)^2}. \tag{38}$$

2. **Average Treatment Effect Error (ATE-error):** Measures population-level bias, where the estimated ATE ($\widehat{\mathrm{ATE}}$) is computed as the sample expectation of ITE:

$$\widehat{\mathrm{ATE}} = \frac{1}{m} \sum_{j=1}^{m} \hat{\tau}_j, \qquad \mathrm{ATE\text{-}error} = \left| \widehat{\mathrm{ATE}} - \mathrm{ATE}_{\mathrm{true}} \right|. \tag{39}$$

For methods with interval-producing components (*IntervalGP-VAE*, the BART/BCF-style ensemble, conformalized *TARNet*, and conformalized *TEDVAE*), we evaluate the calibration–efficiency trade-off by reporting the empirical coverage rate of the nominal 90% ITE interval, the mean interval width, and the interval score. The interval score for individual $j$ is defined as

$$S_{\alpha,j} = \left( \hat{\tau}_j^{\mathrm{upper}} - \hat{\tau}_j^{\mathrm{lower}} \right) + \frac{2}{\alpha} \left( \hat{\tau}_j^{\mathrm{lower}} - \tau_j \right) \mathbb{I} \left\{ \tau_j < \hat{\tau}_j^{\mathrm{lower}} \right\} + \frac{2}{\alpha} \left( \tau_j - \hat{\tau}_j^{\mathrm{upper}} \right) \mathbb{I} \left\{ \tau_j > \hat{\tau}_j^{\mathrm{upper}} \right\}. \tag{40}$$

The average interval score over the test set is then given by

$$\mathrm{IS}_\alpha = \frac{1}{m} \sum_{j=1}^{m} S_{\alpha,j}. \tag{41}$$

Finally, training time is recorded across all models to quantify computational efficiency. The hyperparameters used for the *IntervalGP-VAE* method are listed in Table 3. The comprehensive evaluation results are summarized in Table 4 and visualized in Figure 3.

Overall, *IntervalGP-VAE* achieves the strongest point-estimation performance among the compared methods, obtaining the lowest average PEHE and ATE error. The interval-estimation results reveal clearer differences among the uncertainty-aware methods. The BART/BCF-style ensemble produces relatively sharp intervals, but its empirical coverage is substantially below the nominal level, indicating severe under-calibration.

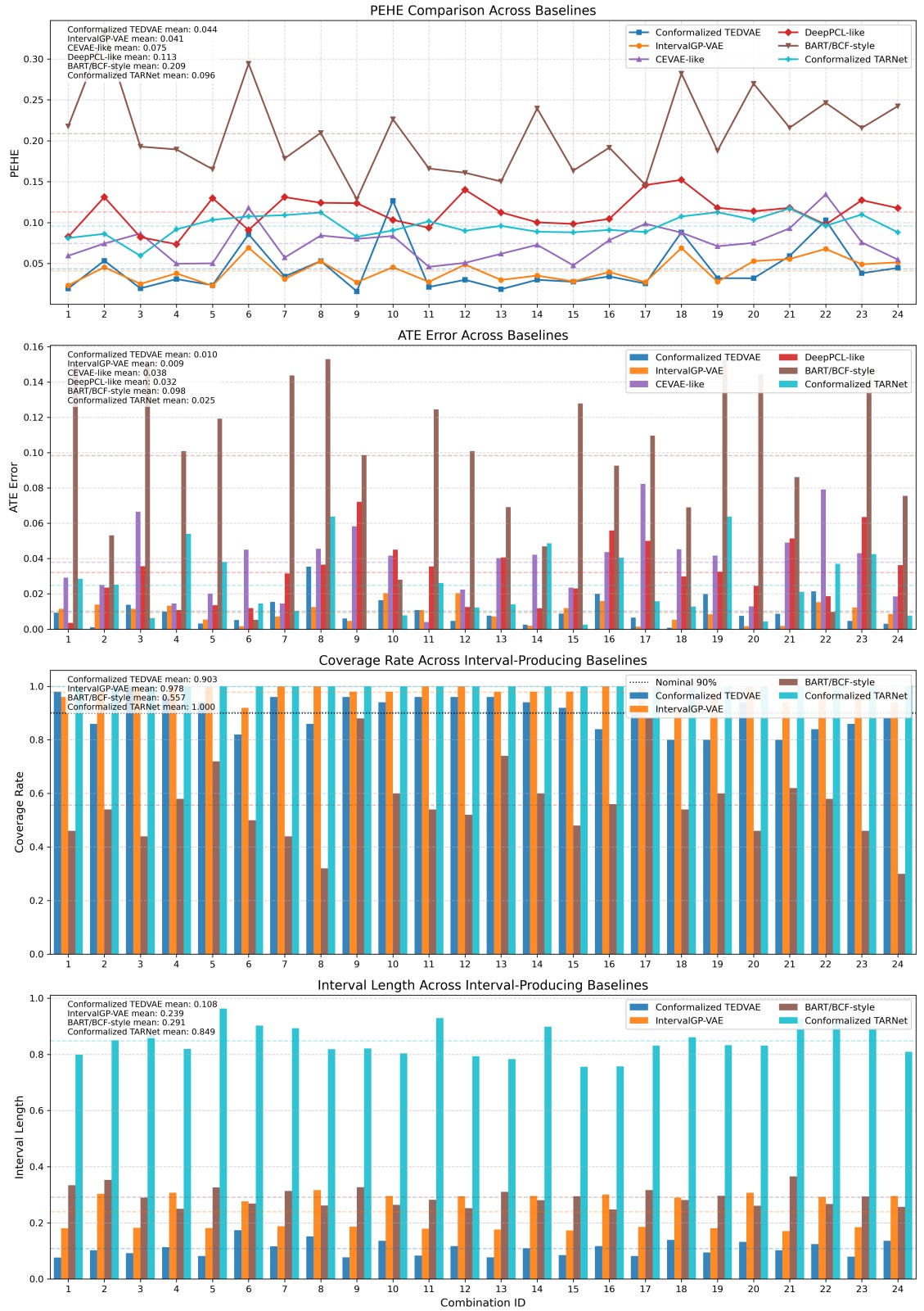

Figure 3: Extended baseline comparison on synthetic settings.

Table 3: IntervalGP-VAE hyperparameters used in the synthetic experiments.

| Notation | Value | Notation | Value |
|---|---|---|---|
| $d$ | 1 | $k$ | 6 |
| $n$ | 1000 | $m$ | 50 |
| $H_{\text{vae}}$ | 64 | $H_\psi$ | 64 |
| $\ell$ | 7.0 | $\sigma_f^2$ | 2.0 |
| $\sigma_n^2$ | $10^{-4}$ | $E_{\text{joint}}$ | 200 |
| $E_{\text{head}}$ | 100 | $E_{\text{vae}}$ | 50 |
| $B$ | 128 | $\eta_{\text{joint}}$ | $10^{-3}$ |
| $\eta_{\text{head}}$ | $10^{-3}$ | $\eta_{\text{vae}}$ | $10^{-4}$ |
| $\lambda_{\text{wd}}$ | $10^{-5}$ | $1 - \alpha$ | 0.90 |
| $S$ | 100 | $q_\phi(u, \epsilon, u_{a0}, u_{a1} \mid z)$ | MLP |
| $p_\theta(z \mid u, \epsilon, u_{a0}, u_{a1})$ | MLP | $f_\psi(u, t, z_Y, u_{a1})$ | MLP |
| $k_\eta(z, z')$ | RBF | Optimizer | AdamW |

Table 4: Baseline comparison on synthetic settings.

| Method | PEHE | ATE Error | Coverage | Width | Interval Score | Time (s) |
|---|---|---|---|---|---|---|
| BART/BCF-style ensemble | $0.2088 \pm 0.0509$ | $0.0983 \pm 0.0460$ | $0.5567 \pm 0.1427$ | $0.2914 \pm 0.0331$ | $1.1375 \pm 0.4627$ | $1.38 \pm 0.08$ |
| CEVAE | $0.0746 \pm 0.0222$ | $0.0379 \pm 0.0204$ | – | – | – | $2.08 \pm 0.11$ |
| Conformalized TARNet | $0.0959 \pm 0.0131$ | $0.0249 \pm 0.0194$ | $\mathbf{1.0000 \pm 0.0000}$ | $0.8486 \pm 0.0572$ | $0.8486 \pm 0.0572$ | $\mathbf{1.29 \pm 0.07}$ |
| DeepPCL | $0.1130 \pm 0.0207$ | $0.0321 \pm 0.0179$ | – | – | – | $1.88 \pm 0.09$ |
| IntervalGP-VAE | $\mathbf{0.0412 \pm 0.0150}$ | $\mathbf{0.0094 \pm 0.0057}$ | $0.9783 \pm 0.0257$ | $0.2393 \pm 0.0604$ | $0.2741 \pm 0.0863$ | $2.42 \pm 0.09$ |
| Conformalized TEDVAE | $0.0436 \pm 0.0292$ | $0.0101 \pm 0.0081$ | $0.9033 \pm 0.0637$ | $\mathbf{0.1083 \pm 0.0269}$ | $\mathbf{0.1914 \pm 0.1280}$ | $9.40 \pm 0.25$ |

This suggests that bootstrap-style tree variation alone is insufficient to capture proxy-induced measurement uncertainty. In contrast, conformalized *TARNet* attains very high coverage, but only by producing overly conservative intervals. Thus, it provides coverage at the cost of poor interval efficiency. Compared with these alternatives, *IntervalGP-VAE* provides a more favourable calibration–efficiency trade-off. It maintains high empirical coverage while producing substantially sharper intervals than conformalized *TARNet*, and it achieves a lower interval score than most competing uncertainty-aware baselines. These results indicate that *IntervalGP-VAE* can propagate latent-confounder recovery uncertainty to downstream ITE intervals without relying on brute-force conformal widening. Although conformalized *TEDVAE* achieves strong interval efficiency under oracle conformal calibration, it incurs a substantially higher computational cost. By contrast, *IntervalGP-VAE* delivers slightly better point accuracy, stronger empirical coverage, and much lower runtime. Taken together, these results show that *IntervalGP-VAE* offers the most robust balance between ITE estimation accuracy, uncertainty calibration, interval efficiency, and computational cost.

To inspect the alignment between confounder recovery and ITE interval calibration more closely, Figure 4 illustrates the estimated 90% intervals for a representative synthetic replicate corresponding to Combination 12. As shown in Figure 4a, the recovered hidden-confounder intervals wrap closely around the true latent state trajectory. This precise alignment directly translates into the well-calibrated, non-collapsing ITE intervals displayed in Figure 4b. The near-nominal coverage of these intervals empirically demonstrates the capability of *IntervalGP-VAE* to enforce latent-space coherence through the GP prior and smoothly propagate proxy-level uncertainty into reliable, well-bounded treatment effect estimations. Specifically, this strict geometric alignment occurs because the structural outcome equation for Combination 12 is governed by

$$Y = 0.5U + 0.3\cos(U) + T(0.5 + 0.5U) + \epsilon_Y, \quad \epsilon_Y \sim \mathcal{N}(0, \sigma_Y^2). \tag{42}$$

By evaluating the potential outcomes under counterfactual regimes, the true structural individual treatment effect (ITE) simplifies linearly to a function of the unobserved confounder:

$$\tau(U) = Y(1) - Y(0) = 0.5 + 0.5U. \tag{43}$$

Consequently, the trajectory of the true structural ITE is an exact, positive monotonic affine transformation of the true hidden confounder $U$. This underlying causal mechanism directly explains the identical geometric

profiles shared between Figure 4a and Figure 4b, validating that *IntervalGP-VAE* successfully captures the structural SCM invariant mapping.

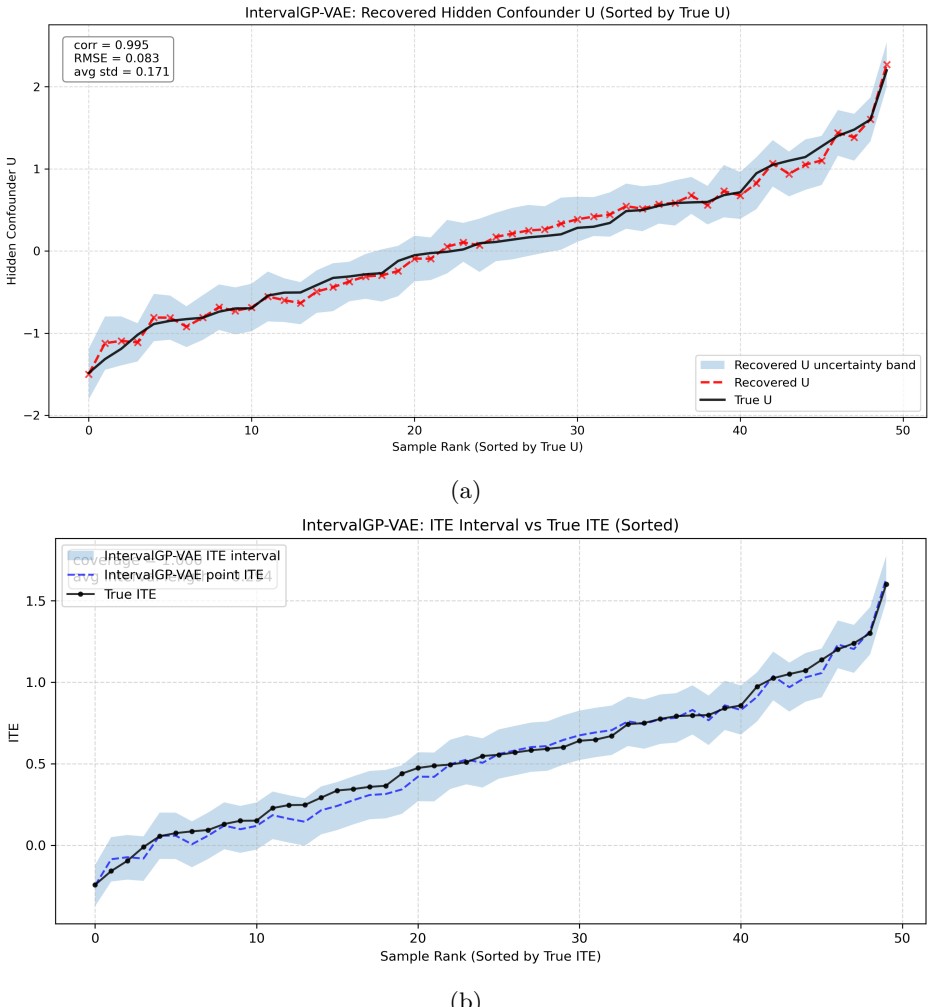

(a)

(b)

Figure 4: Results for Combination 12: (a) recovered hidden confounder intervals and (b) ITE intervals.

### 6.1.1 Ablation on Latent Prior Regularization

Table 5: Ablation results on latent prior regularisation.

| ID | PEHE ↓ | ATE Error ↓ | Coverage ↑ | Width ↓ | Cal. Error ↓ | Interval Score ↓ | Latent |Corr.| ↑ | Time (s) ↓ |
|---|---|---|---|---|---|---|---|---|
| A0 | 0.041 | **0.008** | **0.987** | 0.270 | 0.087 | 0.287 | 0.995 | **2.38** |
| A1 | 0.338 | 0.113 | 0.333 | 0.284 | 0.568 | 3.369 | 0.454 | 2.51 |
| A2 | 0.041 | 0.009 | 0.983 | 0.237 | **0.083** | **0.259** | **0.995** | 2.46 |
| A3-NLL | **0.039** | 0.010 | 0.578 | **0.058** | 0.322 | 0.259 | 0.995 | 2.70 |
| A3-KL | 0.043 | 0.010 | 0.567 | 0.062 | 0.333 | 0.289 | 0.993 | 2.87 |
| A4-32 | 0.046 | 0.010 | 0.603 | 0.064 | 0.297 | 0.283 | 0.993 | 2.87 |
| A4-64 | 0.040 | 0.009 | 0.597 | 0.061 | 0.303 | 0.262 | 0.994 | 2.90 |
| A4-128 | 0.040 | 0.010 | 0.584 | 0.061 | 0.316 | 0.260 | 0.995 | 3.08 |

We conduct ablation experiments to isolate the effects of different latent prior regularisers in *IntervalGP-VAE*. To ensure a controlled comparison, all variants share the same architecture, training protocol, hyperparameters, and data-generating settings. The variants differ only in the regularisation imposed on the latent

confounder posterior $q(u \mid z)$. We compare eight variants: **A0**, a standard VAE baseline with an isotropic Gaussian prior $\mathcal{N}(0, I)$; **A1**, an input-dependent Gaussian prior $\mathcal{N}(0, s(z)^2)$; **A2**, the default per-sample IntervalGP regulariser used in our IntervalGP-VAE framework; **A3-NLL**, a joint mini-batch GP prior trained with the GP negative log-likelihood objective; **A3-KL**, a joint mini-batch GP prior trained with the GP KL regularisation objective; and **A4-32**, **A4-64**, and **A4-128**, sparse inducing GP approximations with $M_{\mathrm{ind}} = 32$, 64, and 128 inducing points, respectively. All variants are evaluated on the same synthetic proxy-based causal settings. We report PEHE, ATE error, empirical coverage of the nominal 90% ITE intervals, mean interval width, calibration error, approximate negative log-likelihood, interval score, latent recovery accuracy, and runtime. The results are summarised in Table 5.

The ablation shows that the choice of latent prior regulariser has a substantial effect on both ITE estimation and interval calibration. The standard VAE prior (**A0**) is a strong conservative baseline: it achieves accurate point estimation and high empirical coverage, but its intervals are relatively wider. In contrast, the input-dependent Gaussian prior (**A1**) performs poorly across point-estimation, calibration, and latent-recovery metrics. This indicates that simply allowing the prior variance to depend on the input is insufficient for recovering a coherent latent confounder representation or for producing reliable ITE intervals. The default per-sample IntervalGP regulariser (**A2**) provides the best calibration–efficiency trade-off. It preserves the strong point-estimation accuracy and high empirical coverage of the standard VAE prior, while producing sharper intervals and achieving the most favourable calibration and interval score. It also maintains highly accurate latent recovery. These results support **A2** as an effective default regulariser: it improves interval efficiency without sacrificing coverage or point-estimation accuracy. The joint mini-batch GP variants (**A3-NLL** and **A3-KL**) achieve competitive point-estimation accuracy and very narrow intervals. However, their empirical coverage drops substantially, indicating that these variants tend to produce overconfident uncertainty estimates. Thus, although joint GP regularisation can be beneficial for point estimation, it does not provide the same level of interval calibration as the per-sample IntervalGP regulariser in this experiment. The sparse inducing-point variants (**A4-32**, **A4-64**, and **A4-128**) also achieve competitive PEHE and strong latent recovery, especially as the number of inducing points increases. However, they consistently under-cover the true ITEs and produce overly narrow intervals. This suggests that sparse inducing-point regularisation is a computationally feasible approximation with good point-estimation behaviour, but it does not match the calibration performance of the per-sample IntervalGP regulariser in this small-sample ablation.

Overall, the results support three main conclusions. First, the per-sample IntervalGP regulariser (**A2**) achieves the most favourable balance between point accuracy, interval sharpness, empirical coverage, calibration, and latent recovery. Second, joint and sparse GP regularisers can match or improve point-estimation accuracy, but they tend to produce overly narrow intervals and therefore suffer from under-coverage. Third, the standard VAE prior (**A0**) remains a strong conservative baseline, but **A2** improves interval efficiency while maintaining reliable coverage. These findings justify the use of the per-sample IntervalGP regulariser as the default choice in *IntervalGP-VAE*, especially when calibrated ITE uncertainty is as important as point-estimation accuracy.

### 6.1.2 Scalability Analysis with Sparse Inducing-Point GP Approximation

Section 5.4 discusses the computational cost of GP-based regularisation in *IntervalGP-VAE*. Since exact full GP inference scales cubically with the number of samples, it becomes computationally prohibitive for large training sets. To make the framework computationally realistic, we adopt a lightweight implementation that avoids constructing the full GP posterior covariance matrix and instead uses subset-based diagonal GP inference during the latent smoothing and ITE-space inference stages. We further investigate whether sparse inducing-point IntervalGP regularisation can improve scalability while preserving ITE estimation accuracy and interval behaviour. We conduct a scalability experiment on synthetic datasets generated from the same structural causal setting used in the previous experiments. The training sample size is varied as

$$n \in \{1000, 5000, 10000\},$$

and we compare the current per-sample IntervalGP prior with sparse inducing-point IntervalGP variants using $M_{\mathrm{ind}} \in \{32, 64, 128\}$ inducing points. All models use the same encoder, decoder, causal head, latent dimension, training schedule, learning rate, and data-generating mechanism. We report training time, PEHE,

Table 6: Scalability analysis results for different IntervalGP regularisation strategies.

| $n$ | Method | $M_{\text{ind}}$ | Time (s) ↓ | PEHE ↓ | Coverage ↑ | Width ↓ |
|---|---|---|---|---|---|---|
| 1000 | Per-sample IntervalGP | – | 0.499 | 0.124 | 0.897 | 0.536 |
| 1000 | Sparse IntervalGP | 32 | **0.470** | 0.122 | 0.895 | 0.512 |
| 1000 | Sparse IntervalGP | 64 | 0.467 | 0.119 | 0.899 | **0.511** |
| 1000 | Sparse IntervalGP | 128 | 0.499 | **0.117** | **0.902** | 0.514 |
| 5000 | Per-sample IntervalGP | – | **2.012** | **0.068** | **0.997** | 0.643 |
| 5000 | Sparse IntervalGP | 32 | 2.260 | 0.088 | 0.993 | **0.554** |
| 5000 | Sparse IntervalGP | 64 | 2.285 | 0.084 | 0.995 | 0.561 |
| 5000 | Sparse IntervalGP | 128 | 2.424 | 0.079 | 0.996 | 0.561 |
| 10000 | Per-sample IntervalGP | – | **4.025** | **0.066** | **1.000** | **0.619** |
| 10000 | Sparse IntervalGP | 32 | 4.535 | 0.085 | 0.996 | 0.675 |
| 10000 | Sparse IntervalGP | 64 | 4.581 | 0.077 | 0.999 | 0.675 |
| 10000 | Sparse IntervalGP | 128 | 4.856 | 0.077 | 0.999 | 0.662 |

empirical ITE interval coverage, and mean interval width. All results are averaged over five random seeds and are summarised in Table 6.

Overall, the results show that the computationally realistic implementation scales efficiently to the largest sample size considered. The per-sample IntervalGP prior remains the most accurate method at larger sample sizes, achieving the lowest PEHE in the medium and large data regimes. Sparse inducing-point regularisation provides competitive performance, and increasing the number of inducing points generally improves PEHE among the sparse variants. However, under the current lightweight implementation, the sparse variants do not outperform the per-sample prior in terms of ITE point-estimation accuracy. The computational cost grows approximately linearly with the training sample size for all variants. The sparse inducing-point methods are slightly more expensive than the per-sample prior at larger sample sizes because they require additional kernel computations involving the inducing points. As expected, the overhead increases with the number of inducing points, but it remains moderate in this setting. These results indicate that the proposed implementation avoids the cubic bottleneck of exact full GP inference and remains practical for larger synthetic datasets. In terms of interval behaviour, all methods provide high empirical coverage. At the smallest sample size, coverage is close to the nominal level, while at larger sample sizes the intervals become conservative across all variants. This suggests that the lightweight interval construction is reliable in terms of coverage, but may overestimate uncertainty as the sample size increases. The interval-width results show that sparse inducing-point regularisation can produce narrower intervals in some regimes, but this effect is not consistent across all sample sizes. Taken together, the scalability analysis shows that the per-sample IntervalGP prior offers a strong accuracy–efficiency trade-off under the current implementation. Sparse inducing-point regularisation is computationally feasible and can be useful as an alternative scalable approximation, but it does not consistently improve either point-estimation accuracy or interval sharpness in this experiment. Therefore, we retain the per-sample IntervalGP prior as the default implementation, while viewing sparse inducing-point variants as a promising extension for larger-scale or more complex settings.

### 6.1.3 Kernel Ablation for Latent GP Regularization

Proposition 1 shows that introducing the IntervalGP-based latent prior does not alter the identifiability result of Theorem 1 and remains compatible with the smooth invertible reparameterisation freedom implied by the identifiable latent equivalence class. In addition, the theoretical role of the GP component is to impose latent-space coherence and uncertainty propagation, rather than to rely on a specific kernel form. Although we use the RBF kernel as a numerically stable default in the main experiments, the framework is in principle compatible with other positive-definite kernels, such as Matérn and rational quadratic kernels. We therefore conduct a kernel ablation study to examine whether the empirical performance of *IntervalGP-VAE* depends critically on the RBF kernel, or whether the model remains stable under alternative notions of latent-space similarity. We keep the encoder, decoder, outcome head, latent dimension, training schedule, batch size, learning rate, and data-generating mechanisms fixed, and only replace the kernel used in the latent

Table 7: Kernel ablation results for latent GP regularization.

| Kernel | PEHE ↓ | ATE Error ↓ | Coverage ↑ | Width | NLL ↓ | Interval Score ↓ | corr($\hat{U}, U$) ↑ | Time (s) ↓ |
|---|---|---|---|---|---|---|---|---|
| RBF | **0.045** | 0.009 | 0.983 | 0.249 | -1.056 | 0.286 | **0.996** | 2.57 |
| Matérn-1/2 | 0.054 | 0.010 | **0.994** | 0.394 | -1.108 | 0.421 | 0.992 | 2.45 |
| Matérn-3/2 | 0.054 | 0.008 | 0.980 | 0.263 | -0.890 | 0.323 | 0.994 | 2.50 |
| Matérn-5/2 | 0.051 | **0.006** | 0.981 | 0.258 | -0.957 | 0.317 | 0.995 | 2.50 |
| Rational Quadratic | 0.048 | 0.008 | 0.980 | 0.251 | **-1.230** | 0.295 | 0.996 | 2.47 |
| Linear + RBF | 0.053 | 0.018 | 0.981 | **0.231** | -0.971 | **0.282** | 0.988 | **2.45** |

GP regularizer. We evaluate six kernels: RBF, Matérn-1/2, Matérn-3/2, Matérn-5/2, rational quadratic, and linear-plus-RBF. The experiment is conducted on four representative synthetic settings selected from the original synthetic design, covering different proxy transformations, treatment mechanisms, and outcome functions. Specifically, we select the combinations with IDs 1, 6, 11, and 16 from the full synthetic experiment loop, corresponding to

$$(\mathcal{P}_1, T_1, Y_1), \quad (\mathcal{P}_2, T_1, Y_2), \quad (\mathcal{P}_3, T_2, Y_1), \quad (\mathcal{P}_4, T_2, Y_2),$$

where $\mathcal{P}_j = \{g_{j,1}, \ldots, g_{j,k}\}$ denotes the $j$-th proxy-function set, $T_1$ and $T_2$ denote the two treatment assignment mechanisms, and $Y_1$ and $Y_2$ denote the two outcome mechanisms. For each setting and kernel, we report PEHE, ATE error, empirical coverage of the nominal 90% ITE interval, mean interval width, approximate negative log-likelihood (NLL), interval score, latent-confounder recovery correlation, and training time. The results are summarised in Table 7.

Overall, *IntervalGP-VAE* remains stable across all tested kernels. Across the kernel choices, PEHE remains low, empirical coverage stays above the nominal level, and the recovered latent confounder remains highly correlated with the true latent confounder. These results indicate that the uncertainty calibration and latent recovery performance of *IntervalGP-VAE* are not tied to a specific kernel choice. Among the tested kernels, the RBF kernel provides the strongest overall performance, achieving consistently accurate ITE estimation, reliable coverage, and robust latent recovery. The rational quadratic and Matérn-5/2 kernels also perform competititively, suggesting that several smooth positive-definite kernels are suitable within the proposed framework. The Matérn-1/2 kernel produces more conservative intervals, while the linear-plus-RBF kernel yields sharper intervals but shows slightly weaker treatment-effect estimation and latent recovery. The Matérn-3/2 kernel remains stable but is not uniformly superior to the other smooth kernels. The average training time is similar across kernels, indicating that kernel choice does not introduce a substantial computational burden in this experimental setting. Taken together, these results support the kernel flexibility of *IntervalGP-VAE*, while confirming the RBF kernel as a reliable default choice due to its strong point-estimation accuracy, stable coverage, and robust latent-confounder recovery.

### 6.1.4 Ablation on ITE Interval Uncertainty Sources

Table 8: Ablation results for ITE interval sources.

| ID | Method | PEHE ↓ | ATE Error ↓ | Coverage ↑ | Width ↓ | Interval Score ↓ |
|---|---|---|---|---|---|---|
| B0 | Point only | 0.046 | **0.009** | 0.000 | 0.000 | 0.684 |
| B1 | Encoder MC latent sampling | 0.045 | **0.009** | 0.971 | 0.237 | 0.295 |
| B2 | Latent-GP smoothed MC | 0.045 | **0.009** | 0.398 | **0.042** | 0.363 |
| B3 | FullGP GP-ITE posterior | **0.041** | 0.009 | **0.978** | 0.239 | **0.274** |

To clarify how different uncertainty sources contribute to ITE interval estimation, we conduct an ablation study on the same synthetic settings used above. We compare four variants of *IntervalGP-VAE*:

- **B0**: a point-only baseline with degenerate intervals, where the lower and upper bounds are both set to the point ITE estimate.

- **B1**: constructs empirical intervals from Monte Carlo samples of the encoder posterior $q(U \mid Z)$, thereby isolating uncertainty from the latent posterior.

- **B2**: first applies latent GP posterior smoothing to the recovered confounder and then constructs Monte Carlo ITE intervals from the smoothed latent distribution.

- **B3**: corresponds to the full core GP-ITE posterior mechanism of *IntervalGP-VAE*, where interval-valued ITE pseudo-observations are propagated to test points through a FullGP posterior in the ITE space.

The same evaluation metrics as above are used, and Table 8 reports the results at the 90% nominal level. The results reveal a clear distinction between point-estimation accuracy, interval sharpness, and empirical calibration. The point-only baseline **B0** achieves competitive PEHE and ATE error, but its intervals have zero width and therefore provide no empirical coverage. This confirms that accurate point estimation alone is insufficient for quantifying ITE uncertainty. **B1** achieves high empirical coverage of 0.97, with a mean interval width of 0.24. This indicates that direct Monte Carlo sampling from the encoder posterior can effectively propagate latent confounder uncertainty to the ITE estimate. The resulting intervals are slightly conservative relative to the 90% nominal level, but they provide reliable coverage and a substantially lower interval score than the point-only baseline. **B2** applies latent GP posterior smoothing before constructing Monte Carlo ITE intervals. Although it obtains similar PEHE and ATE error and produces the narrowest non-degenerate intervals, its empirical coverage drops sharply to 0.40. This suggests that latent GP smoothing alone can reduce posterior variability too aggressively, leading to overly narrow intervals and severe under-coverage when no additional ITE-space posterior is used. In contrast, **B3**, the full GP-ITE posterior variant, provides the best overall trade-off. It achieves the lowest PEHE, near-nominal empirical coverage, and the lowest interval score. Compared with **B1**, **B3** slightly improves point-estimation accuracy and interval score while maintaining reliable coverage. Compared with **B2**, it avoids the severe under-coverage caused by latent smoothing alone by propagating interval-valued ITE pseudo-observations through a FullGP posterior in the ITE space. These results show that the ITE-space GP posterior plays a crucial role in restoring calibrated uncertainty after latent GP smoothing.

Overall, the ablation demonstrates the contribution of each uncertainty component in *IntervalGP-VAE*. **B0** removes uncertainty modelling and therefore cannot provide coverage. **B1** captures encoder posterior uncertainty and yields conservative but reliable intervals. **B2** introduces latent GP smoothing, but without the ITE-space FullGP posterior its intervals become overly narrow and under-cover. **B3** combines latent posterior sampling, latent GP smoothing, and FullGP posterior propagation in the ITE space, achieving the best balance between point accuracy, interval width, empirical coverage, and interval score. These findings support the full GP-ITE posterior mechanism as the core uncertainty estimator in *IntervalGP-VAE*.

### 6.1.5 Scaling Analysis of Proxy Count and Hidden Confounder Recovery

We conduct a controlled synthetic experiment to examine how the number of proxy variables affects the recovery of latent confounders and ITE estimation. The main goal is to empirically evaluate the implication of the proxy-identifiability condition in Theorem 1, which suggests that the number of proxies should satisfy $k \geq 2d + 4$ for a $d$-dimensional latent confounder under suitable identifiability assumptions. We consider latent confounders $U \in \mathbb{R}^d$ with dimensions $d \in \{1, 2, 3, 4, 5\}$. For each latent dimension, we vary the proxy count $k$ below, at, and above the theoretical threshold $2d + 4$. Specifically, the proxy-count grids are

$$
\begin{aligned}
d = 1 : \quad & k \in \{1, 2, 3, 4, 5, 6, 7\}, \\
d = 2 : \quad & k \in \{1, 2, 3, 4, 5, 6, 7, 8, 9\}, \\
d = 3 : \quad & k \in \{1, 2, 3, 4, 5, 6, 7, 8, 9, 10, 11\}, \\
d = 4 : \quad & k \in \{1, 2, 3, 4, 5, 6, 7, 8, 9, 10, 11, 12, 13\}, \\
d = 5 : \quad & k \in \{1, 2, 3, 4, 5, 6, 7, 8, 9, 10, 11, 12, 13, 14, 15\}.
\end{aligned}
$$

The theoretical threshold is $k = 2d + 4$, corresponding to $6, 8, 10, 12$, and $14$, respectively. For each pair $(d, k)$, we repeat the experiment over five random seeds. The training and test sample sizes are increased with the

latent dimension, as higher-dimensional latent confounder recovery requires greater sample complexity, as shown below.

| $d$ | $n_{\text{train}}$ | $n_{\text{test}}$ |
|---|---|---|
| 1 | 1200 | 400 |
| 2 | 1600 | 500 |
| 3 | 2200 | 600 |
| 4 | 3000 | 800 |
| 5 | 4000 | 1000 |

The latent confounder vector is sampled as $U = (U_1, \ldots, U_d) \sim \mathcal{N}(0, I_d)$, where $d$ denotes the number of latent confounder components and $U_1, \ldots, U_d$ are independent standard Gaussian variables. The proxy variables are constructed sequentially from three groups of transformations: linear coordinate anchors, coordinate-wise nonlinear transformations, and additional mixed nonlinear projections. Specifically, the first group contains linear coordinate anchors. When sufficient proxy coordinates are available, we include

$$Z_j = U_j + \epsilon_j, \qquad j = 1, \ldots, d, \tag{44}$$

together with an additional weighted linear combination

$$Z_{d+1} = w_1^\top U + \epsilon_{d+1}, \tag{45}$$

where $w_1 \in \mathbb{R}^d$ is a fixed deterministic weight vector. The second group contains coordinate-wise injective nonlinear transformations, given by

$$Z_{d+1+j} = \exp(\rho U_j) + \epsilon_{d+1+j}, \qquad j = 1, \ldots, d, \tag{46}$$

where $\rho = 0.30$, together with one additional mixed nonlinear proxy

$$Z = w_2^\top U + 0.10 \sum_{j=1}^d U_j^3 + \epsilon, \tag{47}$$

where $w_2 \in \mathbb{R}^d$ is another fixed deterministic weight vector. If further proxy coordinates are required, the remaining proxies are generated as mixed projection-based nonlinear transformations. For each additional proxy, we sample two normalized projection directions $a_i, b_i \in \mathbb{R}^d$ and compute

$$s_i = a_i^\top U, \qquad q_i = b_i^\top U. \tag{48}$$

The extra proxy coordinates then cycle through the following nonlinear forms:

$$g_i(U) \in \left\{ s_i, \ s_i + 0.25 s_i^3, \ \sin(s_i) + 0.20 q_i, \ \tanh(s_i) + 0.20 q_i, \ \exp(\rho(s_i + 0.5 q_i)), \ s_i - q_i + 0.10 s_i^2 \right\}. \tag{49}$$

The observed proxy variable is generated as

$$Z_i = g_i(U) + \epsilon_i, \qquad \epsilon_i \sim \mathcal{N}(0, \sigma_z^2), \tag{50}$$

where the proxy noise level is fixed as $\sigma_z = 0.10$. This construction allows us to compare sub-threshold and at/above-threshold proxy regimes in a controlled manner. When $k \geq 2d+4$, the generated proxy coordinates can be partitioned as

$$Z = (Z_{L_1}, Z_{L_2}, Z_{L_3}), \tag{51}$$

with

$$|L_1| = d + 1, \qquad |L_2| = d + 1, \qquad |L_3| \geq 2. \tag{52}$$

When $k < 2d + 4$, the same generation rule is still applied, but the block-size condition is intentionally violated. These sub-threshold settings are included to evaluate how the method behaves when the theoretical proxy-count condition does not hold.

The treatment assignment follows

$$T \sim \text{Bernoulli}\left(\sigma(a_T^\top U)\right), \tag{53}$$

where $a_T$ is a normalized treatment direction and $\sigma(\cdot)$ denotes the sigmoid function. The outcome is generated from

$$Y = b(U) + T\tau(U) + \epsilon_Y, \qquad \epsilon_Y \sim \mathcal{N}(0, \sigma_y^2), \tag{54}$$

where $b(U)$ denotes the baseline outcome function, $\tau(U)$ denotes the individual treatment effect function, and the outcome noise standard deviation is fixed as $\sigma_y = 0.10$. Equivalently, under the same realization of the exogenous outcome noise, the potential outcomes are

$$Y(0) = b(U) + \epsilon_Y, \qquad Y(1) = b(U) + \tau(U) + \epsilon_Y. \tag{55}$$

Although the latent confounder is identifiable only up to an invertible reparameterisation, our empirical evaluation uses linear alignment to account for common linear ambiguities such as rotation, scaling, sign changes, and permutation. Specifically, both the true and inferred latent variables are standardized, and a least-squares linear mapping is fitted from $\hat{U}$ to $U$. The aligned latent recovery error is computed as

$$\text{RMSE}_{\text{aligned}} = \sqrt{\frac{1}{n_{\text{test}}d} \left\| U_{\text{std}} - \widetilde{U}_{\text{aligned}} \right\|_F^2}, \tag{56}$$

where $U_{\text{std}}$ denotes the standardized true latent confounder and $\widetilde{U}_{\text{aligned}}$ denotes the linearly aligned inferred latent representation. We compute this metric for both the raw encoder posterior mean and the GP-smoothed latent representation. We also report the average absolute latent correlation,

$$\text{Corr}_{\text{abs}} = \frac{1}{d} \sum_{j=1}^{d} \max_\ell \left| \text{corr}\left(U_j, \hat{U}_\ell\right) \right|. \tag{57}$$

This metric measures whether each true latent dimension is captured by at least one inferred latent dimension, while being insensitive to sign and permutation ambiguities. For each $(d, k)$ setting, we repeat the experiment over five random seeds and report the mean values of PEHE, ATE error, ITE interval coverage, average interval width, aligned latent RMSE, and average absolute latent correlation. The results are visualized as heatmaps over the latent dimension $d$ and proxy count $k$, with red boxes marking the theoretical threshold $k = 2d + 4$. The results are summarized in Figures 5–9 and Tables 9–10.

Table 9 summarizes the results by separating the sub-threshold regime, $k < 2d + 4$, from the at/above-threshold regime, $k \geq 2d + 4$. The results show that increasing the number of proxies generally improves latent-confounder recovery and ITE estimation for higher-dimensional latent confounders, although the transition at $k = 2d + 4$ is gradual rather than a sharp phase transition. In the one-dimensional case, the task is already relatively easy below the theoretical threshold. When $d = 1$, the sub-threshold regime achieves a PEHE of 0.06, coverage of 0.97, and smoothed latent correlation of 0.99. Moving to the at/above-threshold regime gives a PEHE of 0.08, coverage of 0.87, and smoothed latent correlation of 0.92. This indicates that, for a scalar hidden confounder, a small number of informative nonlinear proxies can already provide sufficient information for latent recovery and ITE estimation, and additional proxies do not necessarily improve finite-sample performance. A diagnostic analysis further shows that, when $d = 1$, the additional proxies generated for larger $k$ are mostly highly correlated nonlinear transformations of the same scalar latent direction. Consequently, the effective rank of the clean proxy matrix remains low, so these additional proxies do not provide genuinely new latent information. This explains why performance at (k=7) can be slightly worse than at (k=6) in the one-dimensional case, as shown in Figures 5–9: the threshold condition provides a sufficient structural proxy-richness guideline, but finite-sample performance also depends on the numerical informativeness and non-redundancy of the generated proxy functions. For higher-dimensional latent confounders, the benefit of moving into the proxy-rich regime is more visible. When $d = 2$, moving from $k < 8$ to $k \geq 8$ reduces PEHE from 0.15 to 0.07, decreases ATE error from 0.07 to 0.01, increases coverage from 0.87 to 0.98, and improves latent correlation from 0.81 to 0.90. The same pattern is observed for $d = 3, 4, 5$: moving to the at/above-threshold regime consistently reduces PEHE and ATE error, improves empirical coverage, and yields better latent recovery. These results indicate that proxy richness becomes increasingly important

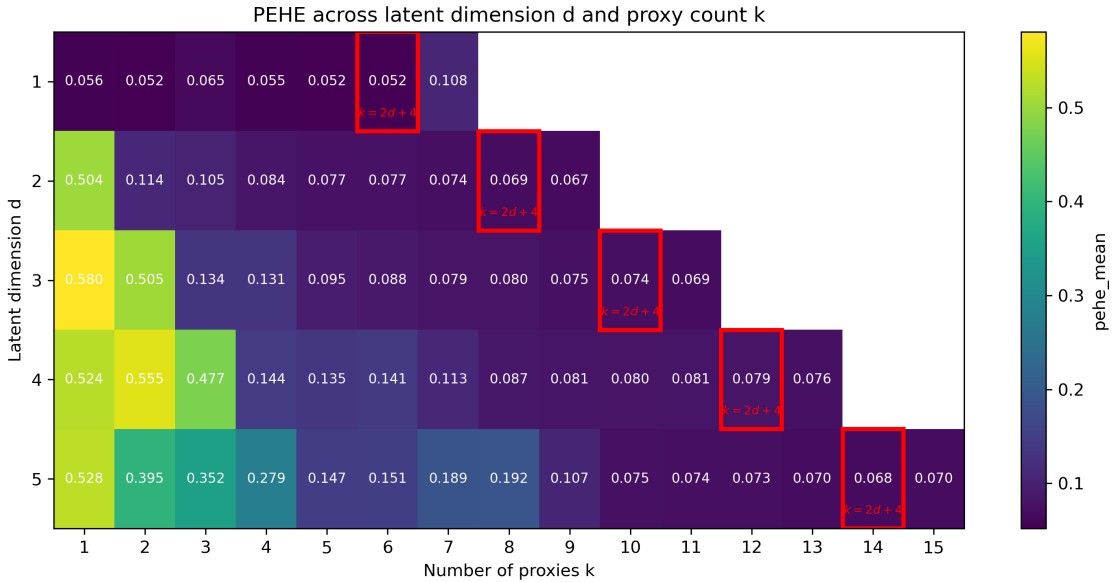

Figure 5: PEHE across proxy counts and latent dimensions.

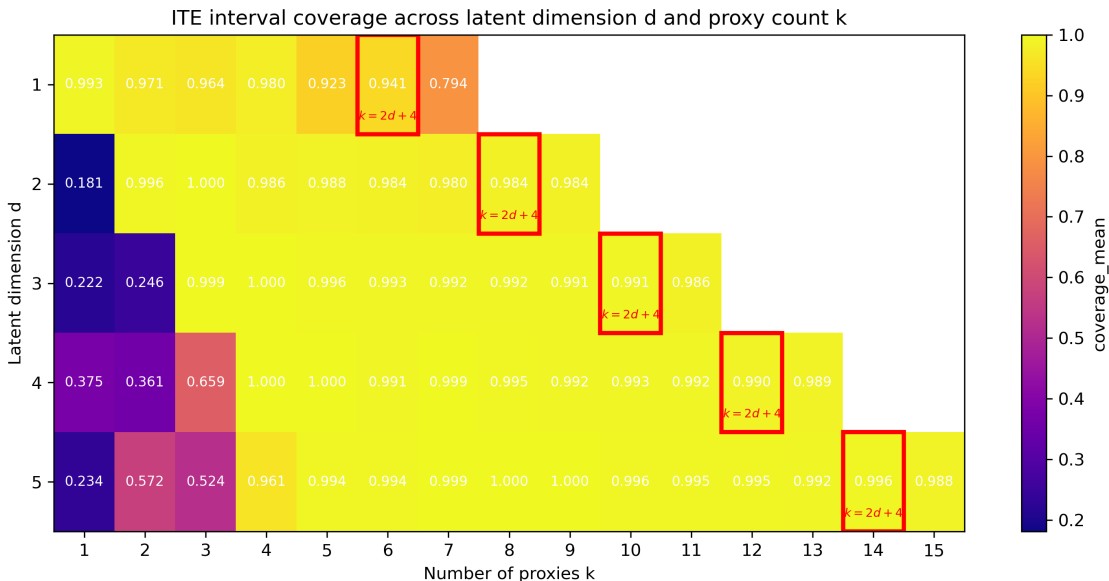

Figure 6: ITE interval coverage rate across proxy counts and latent dimensions.

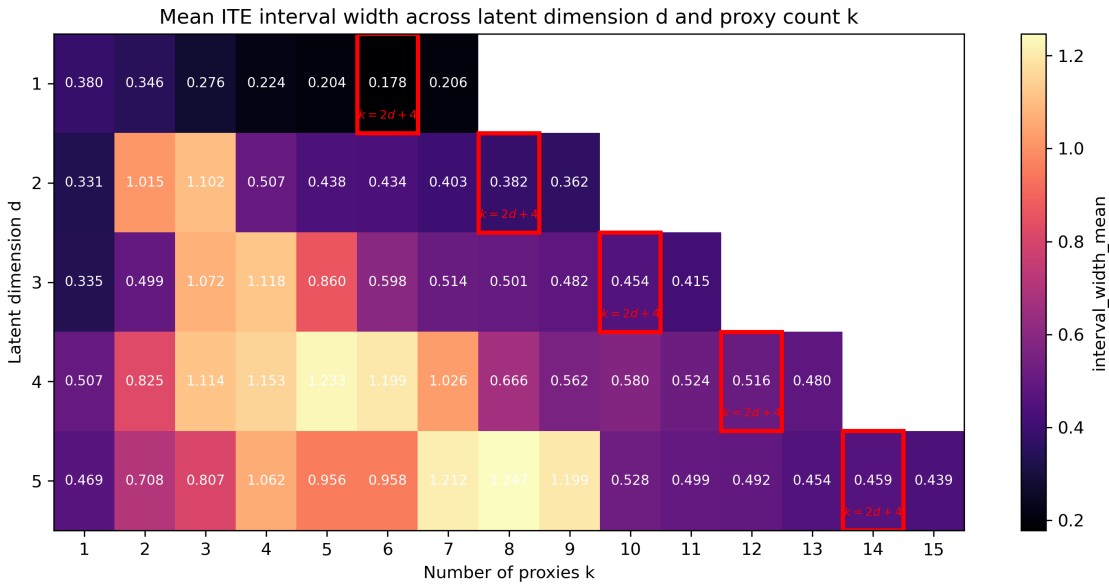

Figure 7: ITE interval width across proxy counts and latent dimensions.

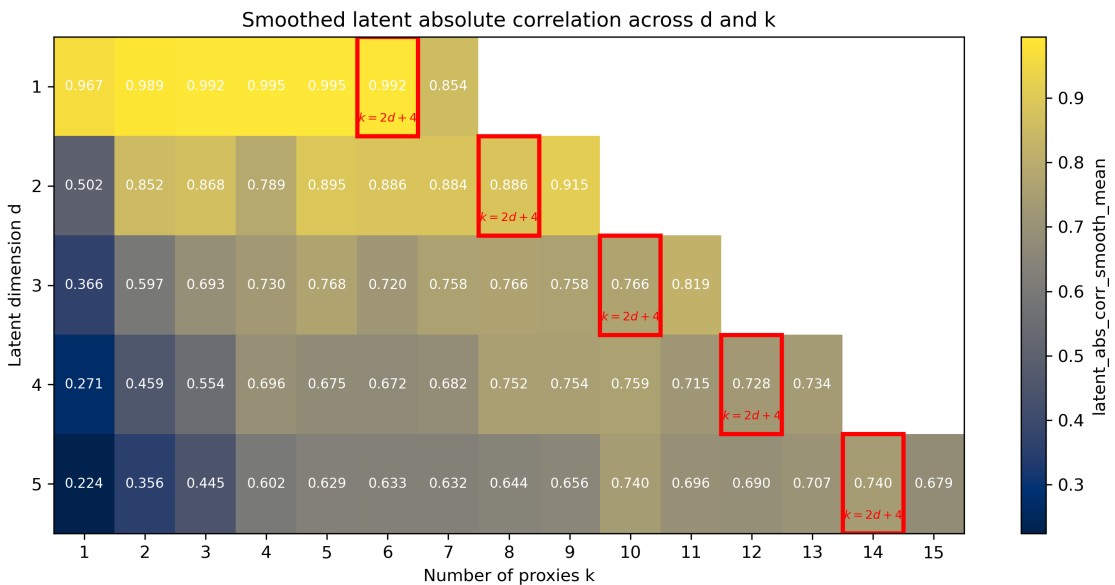

Figure 8: Absolute latent correlation across proxy counts and latent dimensions.

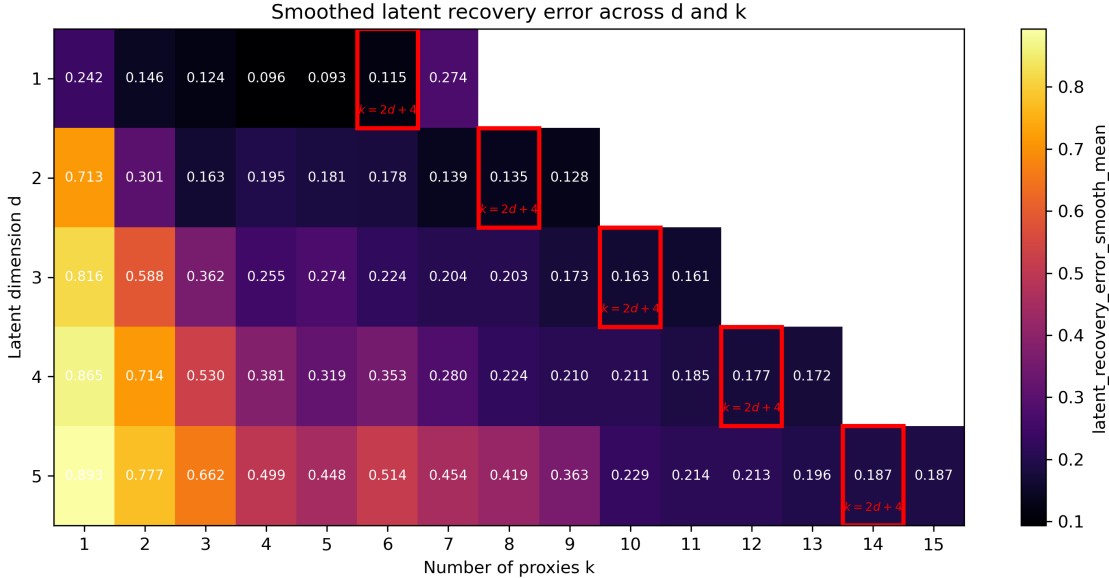

Figure 9: Latent recovery error across proxy counts and latent dimensions.

Table 9: Comparison between sub-threshold and at/above-threshold proxy-count regimes.

| $d$ | Regime | PEHE ↓ | ATE Error ↓ | Coverage ↑ | Width | Latent RMSE ↓ | Latent Corr. ↑ |
|-----|--------|--------|-------------|------------|-------|---------------|----------------|
| 1 | $k < 2d + 4$ | 0.056 | 0.012 | 0.966 | 0.286 | 0.140 | 0.988 |
| 1 | $k \geq 2d + 4$ | 0.080 | 0.022 | 0.868 | 0.192 | 0.195 | 0.923 |
| 2 | $k < 2d + 4$ | 0.148 | 0.068 | 0.874 | 0.604 | 0.267 | 0.811 |
| 2 | $k \geq 2d + 4$ | 0.068 | 0.005 | 0.984 | 0.372 | 0.131 | 0.900 |
| 3 | $k < 2d + 4$ | 0.196 | 0.122 | 0.826 | 0.664 | 0.344 | 0.684 |
| 3 | $k \geq 2d + 4$ | 0.072 | 0.006 | 0.988 | 0.435 | 0.162 | 0.793 |
| 4 | $k < 2d + 4$ | 0.220 | 0.144 | 0.851 | 0.854 | 0.389 | 0.635 |
| 4 | $k \geq 2d + 4$ | 0.077 | 0.009 | 0.990 | 0.498 | 0.175 | 0.731 |
| 5 | $k < 2d + 4$ | 0.202 | 0.128 | 0.866 | 0.815 | 0.452 | 0.589 |
| 5 | $k \geq 2d + 4$ | 0.069 | 0.010 | 0.992 | 0.449 | 0.187 | 0.710 |

Table 10: Results at the exact theoretical threshold $k = 2d + 4$.

| $d$ | $k = 2d + 4$ | PEHE ↓ | ATE Error ↓ | Coverage ↑ | Width | Latent RMSE ↓ | Latent Corr. ↑ |
|-----|--------------|--------|-------------|------------|-------|---------------|----------------|
| 1 | 6 | 0.052 | 0.005 | 0.942 | 0.178 | 0.115 | 0.992 |
| 2 | 8 | 0.069 | 0.003 | 0.984 | 0.382 | 0.135 | 0.886 |
| 3 | 10 | 0.074 | 0.005 | 0.991 | 0.454 | 0.163 | 0.766 |
| 4 | 12 | 0.079 | 0.008 | 0.990 | 0.516 | 0.177 | 0.728 |
| 5 | 14 | 0.068 | 0.010 | 0.996 | 0.459 | 0.187 | 0.740 |

as the latent confounder becomes multidimensional, because additional proxies provide more information for recovering different latent directions.

The threshold-specific results in Table 10 further show that the exact threshold $k = 2d + 4$ provides strong empirical performance for $d \geq 2$, but should be interpreted as a structural proxy-richness guideline rather than as a deterministic finite-sample guarantee. This interpretation is consistent with Theorem 1, which states that "a sufficient condition for identifiability is $k \geq 2d + 4$." Hence, the proxy-count condition should be understood as a conservative sufficient condition for latent-confounder identifiability, not as a necessary condition. Consequently, settings below this threshold are not necessarily unidentifiable; rather, they fall outside the regime for which Theorem 1 provides a formal identifiability guarantee. The best-performing proxy counts also indicate that empirical performance is not strictly monotonic in $k$. The best PEHE is obtained at $k = 2$ for $d = 1$, $k = 9$ for $d = 2$, $k = 11$ for $d = 3$, $k = 13$ for $d = 4$, and $k = 14$ for $d = 5$. Similarly, the highest coverage values are obtained at $k = 1, 3, 4, 4,$ and 8 for $d = 1, 2, 3, 4,$ and 5, respectively. For $d \geq 2$, moving from the sub-threshold regime to the at/above-threshold regime substantially improves PEHE, ATE error, coverage, and smoothed latent correlation. However, the empirical transition around $k = 2d + 4$ remains gradual rather than abrupt. In particular, the one-dimensional case shows that strong finite-sample performance can still be achieved below the threshold when a few proxies are already highly informative. As $d$ increases, more proxies become increasingly helpful because they provide additional information about different latent directions.

### 6.1.6 Analysis of Proxy Quality and Assumption Violation

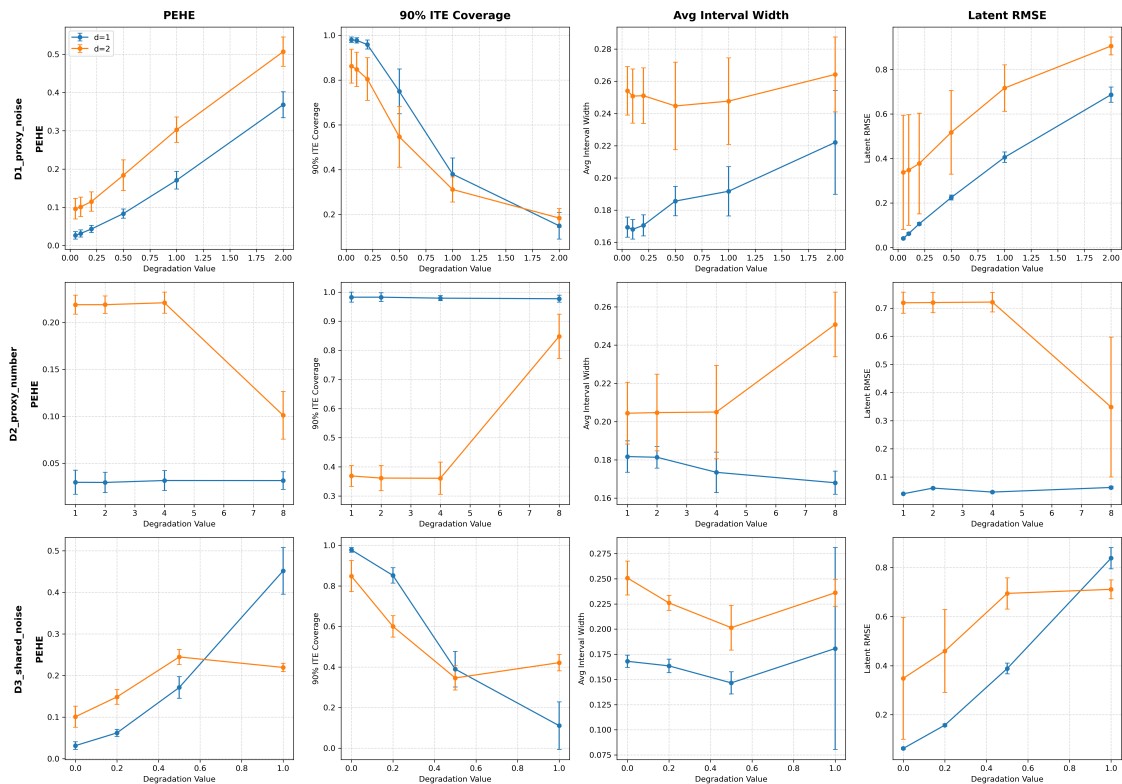

Figure 10: Proxy quality and assumption violation results.

The preceding synthetic experiments are constructed to satisfy the proxy identifiability conditions in Theorem 1, and therefore represent a favourable regime in which the latent confounder is recoverable from multiple complementary proxy variables. In contrast, real-world semi-synthetic datasets such as IHDP are not designed to satisfy these proxy assumptions. In particular, the observed covariates may be noisy, weakly informative, redundant, or statistically dependent beyond the latent confounder. This creates a gap be-

tween the identifiable synthetic setting and the more realistic IHDP setting, where the empirical coverage of the 90% ITE intervals is substantially lower (see Section 6.2). To bridge this gap, we conduct a controlled proxy-quality and assumption-violation analysis. The goal is not to show that *IntervalGP-VAE* remains calibrated under arbitrary violations, but to examine how ITE estimation and interval calibration degrade as the assumptions required for proxy-based identifiability are gradually weakened. To isolate the effect of proxy quality, the treatment and outcome mechanisms are fixed across all settings, while the proxy mechanism is systematically varied. We consider latent dimensions $d \in \{1, 2\}$ and fix the total number of proxies to $k = 8$. We evaluate three types of degradation. First, we vary the independent proxy noise level $\sigma_z \in \{0.05, 0.10, 0.20, 0.50, 1.00, 2.00\}$ while keeping all proxies informative. This setting tests the effect of increasingly noisy proxy measurements. Second, we reduce the number of informative proxies while keeping the total proxy dimension fixed. Specifically, only $r \in \{k, k/2, k/4, 1\}$ proxies are generated as informative nonlinear functions of $U$, while the remaining proxies are redundant copies of the same proxy function. This setting tests the importance of non-redundant proxy variability. Third, we introduce shared proxy noise,

$$Z_i = g_i(U) + \eta_{\text{shared}} + \epsilon_i, \qquad \eta_{\text{shared}} \sim \mathcal{N}(0, \sigma_{\text{shared}}^2), \tag{58}$$

with $\sigma_{\text{shared}} \in \{0.00, 0.20, 0.50, 1.00\}$. This directly violates the conditional independence assumption among proxies given $U$. For each setting, we report PEHE, empirical 90% ITE interval coverage, mean interval width, and aligned latent recovery error. All results are averaged over three random seeds. The results of this proxy-quality and assumption-violation analysis are visualized in Figure 10.

The results reveal three complementary patterns. First, increasing independent proxy noise consistently worsens latent recovery and ITE point estimation. In the one-dimensional setting, as $\sigma_z$ increases from 0.05 to 2.00, PEHE increases from 0.03 to 0.37, and the aligned latent RMSE increases from 0.04 to 0.69. Empirical coverage decreases from 0.98 to 0.15, while the average interval width only increases mildly from 0.17 to 0.22. A similar pattern is observed in the two-dimensional setting. These results show that severe independent proxy noise can substantially degrade both point estimation and interval calibration, particularly when the learnt ITE intervals do not expand enough to reflect weaker proxy information.

Second, reducing non-redundant proxy number has little effect in the one-dimensional setting but causes substantial degradation in the two-dimensional setting. When $d = 1$, reducing the number of informative proxies from 8 to 1 leaves PEHE almost unchanged, from 0.03 to 0.03, while coverage remains high and the interval width changes only slightly. This suggests that, for a scalar latent confounder, repeated noisy measurements of the same latent direction can still provide sufficient information for recovering the relevant confounding signal. In contrast, when $d = 2$, reducing the number of informative proxies from 8 to 1 increases PEHE from 0.10 to 0.22, increases aligned latent RMSE from 0.35 to 0.72, and reduces coverage from 0.85 to 0.37. This indicates that redundant proxies are particularly harmful when the latent confounder is multidimensional.

Third, violating conditional independence through shared proxy noise leads to substantial calibration degradation. In the one-dimensional setting, increasing $\sigma_{\text{shared}}$ from 0.00 to 1.00 increases PEHE from 0.03 to 0.45 and increases aligned latent RMSE from 0.06 to 0.84, while empirical coverage drops from 0.98 to 0.11. The interval width changes only from 0.17 to 0.18, which is insufficient to compensate for the deterioration in latent recovery. In the two-dimensional setting, shared noise also worsens both point estimation and calibration: increasing $\sigma_{\text{shared}}$ from 0.00 to 1.00 increases PEHE from 0.10 to 0.22, increases aligned latent RMSE from 0.35 to 0.71, and reduces coverage from 0.85 to 0.42. These results show that shared proxy noise can systematically distort the recovered latent representation, while the resulting ITE intervals remain overconfident under this structural violation.

Taken together, this experiment shows that proxy quality affects ITE uncertainty through both latent recovery and interval calibration. When proxies become noisier, less informative, more redundant, or conditionally dependent beyond the latent confounder, the learnt latent representation becomes less accurate, which directly worsens ITE point estimation. At the same time, the ITE intervals may fail to widen sufficiently to reflect the weakened proxy information, leading to under-coverage. These findings provide a controlled synthetic-to-real explanation for the lower coverage observed on IHDP (see Section 6.2): real-world covariates may only partially satisfy the conditional-independence, non-redundancy, and informativeness assumptions required for proxy-based identifiability. Thus, the lower IHDP coverage should be interpreted as evidence

Table 11: IntervalGP-VAE hyperparameters used in the semi-synthetic experiments.

| Notation | Value | Notation | Value |
|---|---|---|---|
| $d$ | 4 | $k$ | 6 |
| $n$ | 672 | $m$ | 75 |
| $H_{\text{vae}}$ | 64 | $H_\psi$ | 192 |
| $\ell$ | 1.85 | $\sigma_f^2$ | 3.00 |
| $\sigma_n^2$ | 0.012 | $E_{\text{joint}}$ | 650 |
| $E_{\text{head}}$ | 250 | $E_{\text{vae}}$ | 75 |
| $B$ | 128 | $\eta_{\text{joint}}$ | $10^{-3}$ |
| $\eta_{\text{head}}$ | $10^{-3}$ | $\eta_{\text{vae}}$ | $10^{-4}$ |
| $\lambda_{\text{wd}}$ | $10^{-5}$ | $1 - \alpha$ | 0.90 |
| $S$ | 100 | $q_\phi(u, \epsilon, u_{a0}, u_{a1} \mid z)$ | MLP |
| $p_\theta(z \mid u, \epsilon, u_{a0}, u_{a1})$ | MLP | $f_\psi(u, t, u_{a1})$ | MLP |
| $k_\eta(z, z')$ | RBF | Optimizer | AdamW |

of partial proxy-assumption violation in this benchmark, rather than as a failure of interval construction in the identifiable proxy regime.

## 6.2 Semi-synthetic Experiments

### 6.2.1 Experimental Setup

To evaluate the proposed *IntervalGP-VAE* under a realistic causal inference scenario with known ground-truth counterfactuals, we conduct a 100-replication study on the Infant Health and Development Program (IHDP) benchmark dataset (Hill, 2011). We report PEHE, ATE error, empirical coverage, and the average width of the 90% ITE uncertainty intervals. To reduce experimental variance and facilitate reproducible comparisons, we retain only the six continuous covariates and exclude the remaining 19 discrete or binary variables. In this setting, the observed continuous covariates are treated as proxy variables $Z$ ($k = 6$) for the latent socio-demographic confounding factors $U$. Mechanistically, *IntervalGP-VAE* leverages these six continuous covariates to infer latent confounder representation, which is then utilized for treatment-effect estimation and uncertainty quantification. The detailed hyperparameter configurations for the semi-synthetic experiments are summarized in Table 11.

### 6.2.2 Theoretical Boundaries and Latent Dimensionality

The application of *IntervalGP-VAE* to the IHDP dataset should be interpreted in light of the proxy-identifiability assumptions underlying our framework. Theorem 1 establishes that the latent confounder is identifiable from the proxy distribution up to a smooth invertible reparameterisation when the proxies satisfy conditional independence, sufficient variability, bounded completeness, and a sufficient proxy-count condition. In particular, the theorem gives the conservative sufficient condition $k \geq 2d + 4$, reflecting the amount of independent proxy information needed to distinguish latent states in a $d$-dimensional confounder space. Thus, for a fixed number of observed proxies, the admissible latent dimension should be chosen consistently with this proxy-richness requirement. In the IHDP setup used in our experiments, restricting the input to continuous covariates yields only $k = 6$ proxy variables. Under the conservative sufficient condition $k \geq 2d + 4$, this proxy count formally supports only $d \leq 1$. Nevertheless, to allow sufficient representation capacity on IHDP and to assess the method beyond the regime fully covered by the theory, we empirically set the latent dimension to $d = 4$ in the *IntervalGP-VAE* architecture. This choice should therefore be viewed as an empirical modelling choice and a practical stress test under partial proxy-assumption violation, rather than as a configuration fully guaranteed by Theorem 1. This interpretation is also consistent with the scaling analysis in Section 6.1.5. There, we observed that the theoretical threshold $k = 2d + 4$ is a conservative sufficient condition rather than a sharp finite-sample boundary. For example, when $d = 4$, the formal threshold is $k = 12$, but competitive finite-sample performance can already appear with fewer proxies, depending on the evaluation metric and the informativeness of the proxy functions. This suggests

that the choice $d = 4$ on IHDP is not intended to claim full theoretical identifiability under $k = 6$, but rather to evaluate how the method behaves in a realistic semi-synthetic setting where the proxy conditions are only partially satisfied. Conversely, while baselines such as *TEDVAE* may use a high-dimensional latent space, for example $d = 20$, to improve empirical representation capacity, such choices do not by themselves provide formal identifiability guarantees within a strict proxy-based framework. In principle, recovering a high-dimensional latent structure is beneficial only when the observed proxies provide sufficient, informative, and non-redundant information to identify it. Otherwise, over-parameterisation may lead to biased estimation and miscalibrated uncertainty intervals under misspecified proxy mechanisms.

### 6.2.3 Results and Discussion

Given that Conformalized TEDVAE is among the most competitive baselines in the synthetic experiments, we use TEDVAE as the primary comparator for the semi-synthetic IHDP benchmark. This focused comparison allows us to examine the calibration–sharpness and accuracy–efficiency trade-offs against a strong latent-variable baseline. Figure 11 presents the comparative evaluation on the IHDP benchmark over 100 replications, highlighting distinct trade-offs among pointwise accuracy, uncertainty calibration, and interval sharpness. Due to space limitations, runtime results are not shown in the figure.

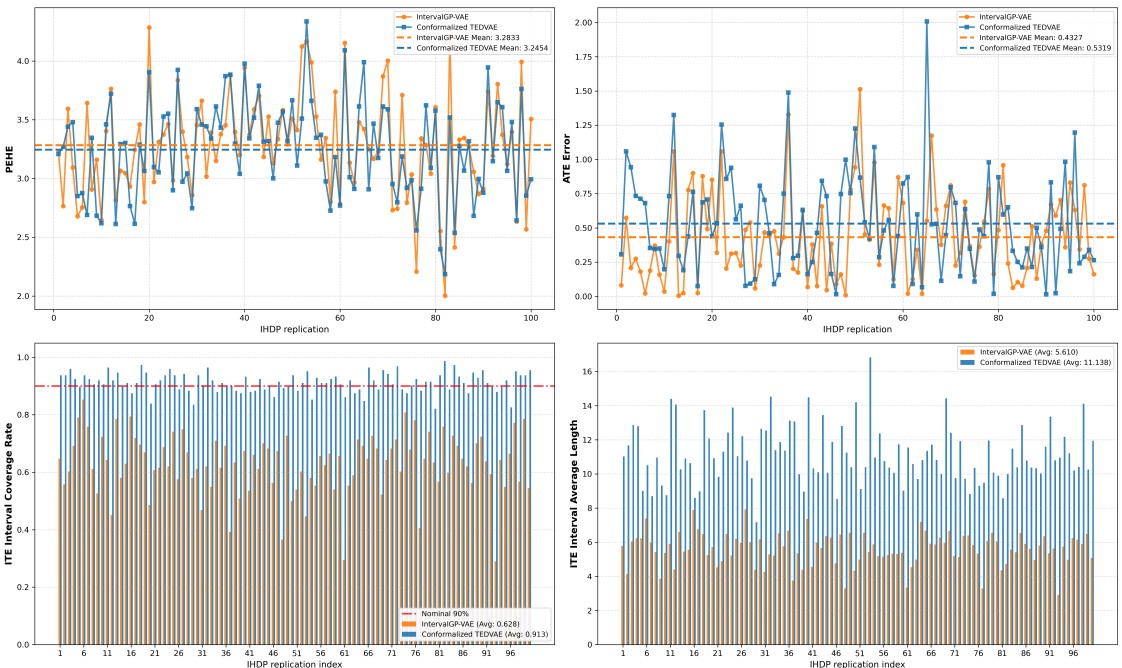

Figure 11: Baseline comparison on the semi-synthetic IHDP dataset.

- **Treatment Effect Estimation:** In terms of individual-level estimation, *TEDVAE* yields a slightly lower test PEHE of 3.25 compared to 3.28 for *IntervalGP-VAE*, indicating a minor advantage in pointwise ITE prediction. However, *IntervalGP-VAE* achieves a lower ATE absolute error (0.43 vs. 0.53 for *TEDVAE*), representing an approximate 18.87% relative reduction in population-level treatment effect error.

- **Uncertainty Quantification:** *TEDVAE* with conformal calibration achieves an empirical coverage of 0.91 (close to the nominal 90% level) but requires a substantial average interval width of 11.14. In contrast, the uncalibrated (direct) *IntervalGP-VAE* yields much sharper intervals with an average width of 5.61 (a 49.64% reduction), though its empirical coverage drops to 0.63 due to the aforementioned proxy limitations of IHDP.

- **Computational Cost:** *IntervalGP-VAE* demonstrates remarkable computational efficiency, requiring an average execution time of only 3.73 seconds per replication, compared to 8.72 seconds for Conformalized *TEDVAE*.

These empirical findings closely align with our theoretical analysis. Our custom synthetic datasets are explicitly constructed with multiple complementary, nonlinear, and non-redundant proxy functions to support latent recovery under the proxy-identifiability assumptions. In contrast, the six continuous covariates in IHDP may be weakly informative, redundant, or statistically dependent beyond the latent confounders, thereby degrading the calibration of direct interval estimates. This highlights an important distinction between empirical representation capacity and proxy-based identifiability: strong predictive representations may still perform well on a given benchmark, but formal recovery of latent confounders requires structured proxy conditions. Thus, *IntervalGP-VAE* is primarily tailored to scenarios where confounders are genuinely unobserved and must be recovered through identifiable proxy mechanisms. Consequently, our main contribution is not to claim universal empirical dominance across all potentially misspecified benchmarks, but to provide explicit identifiability conditions and principled, interval-based uncertainty quantification, which are traditionally absent in VAE-based causal models.

## 7 Conclusion

This work demonstrates that *IntervalGP-VAE* provides a principled framework for causal inference in the presence of unobserved confounding. By combining latent variable modelling with interval-valued Gaussian process inference, the framework enables both recovery of latent confounders from noisy proxy measurements and coherent quantification of the uncertainty that such recovery inevitably entails. In contrast to approaches that produce only point estimates of individual treatment effects (ITEs), our model provides calibrated ITE intervals, thereby explicitly representing uncertainty arising from latent confounding and imperfect proxy measurements.

A central contribution of this work lies in the identifiability analysis. We show that, under structural assumptions on the proxy generation mechanism, the latent confounder can be recovered up to a smooth invertible transformation. Importantly, we further establish that counterfactual treatment effects remain invariant under such transformations, ensuring that causal conclusions are preserved despite this residual ambiguity. These results provide theoretical justification for proxy-based causal inference and clarify the conditions under which unmeasured confounding can be addressed using noisy proxy variables. Nevertheless, a limitation of our framework is that these identifiability guarantees rely on structural assumptions about the proxy generation mechanism which, in practice, may be challenging to verify. Developing data-driven methods to test or relax these assumptions remains an important direction for future work.

Empirically, the results support our central hypothesis: integrating VAE-based latent representation learning with interval-valued GP inference yields reliable and uncertainty-aware estimation of ITEs when the proxies satisfy the identifiability conditions. Experiments on synthetic and semi-synthetic benchmarks demonstrate that the proposed approach produces well-calibrated uncertainty intervals while maintaining competitive estimation accuracy.

Several promising directions follow from this work. Extending the framework to multivariate or structured latent confounders would broaden its applicability in more complex causal systems. Incorporating temporal or spatial structures could further enable counterfactual reasoning in longitudinal health, education, or environmental studies. In addition, improving robustness to incomplete or partially informative proxies—through adaptive kernel learning or causal feature selection—may enhance the applicability of proxy-based causal inference in real-world observational datasets. Finally, connecting interval-valued ITE estimates with decision-theoretic objectives, such as risk-sensitive treatment recommendation or fairness-aware policy optimisation, offers a promising direction for developing uncertainty-aware causal models that directly support practical decision making. A limitation of our framework is that the identifiability guarantees rely on structural assumptions about the proxy generation mechanism. In practice, verifying these assumptions may be challenging and remains an important direction for future work.

## 8 Broader Impact Statement

The proposed IntervalGP-VAE framework is motivated by applications in which individual treatment effect estimates may support consequential decisions, such as healthcare, economics, and social sciences. In such settings, uncertainty quantification is important because treatment-effect estimates based on noisy or incomplete proxy variables may otherwise lead to overconfident conclusions. By explicitly modelling uncertainty in latent confounder recovery and propagating this uncertainty to ITE intervals, the proposed method can help users identify cases where the available proxy information is insufficient for reliable individual-level causal estimation.

At the same time, the method should not be interpreted as an automated decision-making tool. Its validity depends on structural assumptions about the proxy-generation mechanism, including proxy informativeness, conditional independence, non-redundancy, and sufficient proxy variability. These assumptions may be difficult to verify in real-world applications. If the observed proxies are weakly informative, biased, redundant, or affected by unmodelled dependence, the recovered latent representation may be unreliable, and the resulting ITE intervals may be misleading or overconfident. Therefore, the method should be used only with careful domain-specific validation, sensitivity analysis, and human oversight.

More broadly, uncertainty-aware causal estimation has the potential to support more cautious and transparent decision-making by making model uncertainty explicit. However, such estimates should be treated as decision-support evidence rather than definitive recommendations. In high-stakes applications, conclusions drawn from IntervalGP-VAE should be combined with substantive domain knowledge, external validation, fairness considerations, and ethical review before being used to inform policy or individual-level interventions.

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

# Appendix

## A   Proof of Theorem 1

> **Theorem 1** (Identifiability of the latent confounder from noisy proxies)
>
> Consider the proxy–latent relationship defined in Eq. 4. Suppose the proxy variables satisfy conditional independence (Eq. 5), the variability condition (Eq. 6, 7), and the bounded completeness condition (Eq. 8). A sufficient condition for identifiability is
>
> $$k \geq 2d + 4.$$
>
> Under these conditions, the latent confounder $U \in \mathbb{R}^d$ is identifiable from the observed proxy distribution $p(Z)$ up to a smooth, invertible reparameterisation.

*Proof.* **Step 1: Finite anchor sets.** Let

$$\mathcal{U}_r = \{u^0, \ldots, u^d\}$$

be any collection of $d + 1$ distinct latent values in the support of $U$. Set

$$r = d + 1.$$

By Eq. 7, the corresponding conditional proxy distributions $p(Z \mid U = u^0), \ldots, p(Z \mid U = u^d)$ are linearly independent as functions of $Z$.

**Step 2: Proxy partition.** By the partitionable proxy variability condition in Eq. 6, there exists a fixed partition of the proxy coordinates into three disjoint blocks,

$$Z = (Z_{L_1}, Z_{L_2}, Z_{L_3}),$$

such that the block conditional families over the anchor set $\mathcal{U}_r$ satisfy the rank requirements needed below.

**Step 3: Block conditional matrices and Kruskal ranks.** For each block $L_j$, define the conditional matrix

$$M_j(\ell, m) := \mathbb{P}(Z_{L_j} = m \mid U = u^\ell), \qquad \ell = 0, \ldots, d,$$

where $m$ indexes the discretised joint outcomes of the proxy block $Z_{L_j}$.

By conditional independence of the proxies given $U$, each block conditional law factorises as

$$p(Z_{L_j} \mid U = u^\ell) = \prod_{i \in L_j} p(Z_i \mid U = u^\ell).$$

By Eq. 6, the block conditional distributions in $L_1$ are linearly independent across the $r = d + 1$ anchor states, and the same holds for $L_2$. After sufficiently fine discretisation, this gives

$$\mathrm{krank}(M_1) = r, \qquad \mathrm{krank}(M_2) = r.$$

For the third block, Eq. 6 ensures that at least two block conditional distributions are linearly independent, so

$$\mathrm{krank}(M_3) \geq 2.$$

**Step 4: Finite-anchor identifiability.** The joint distribution of $(Z_{L_1}, Z_{L_2}, Z_{L_3})$ admits the finite-mixture representation

$$p(Z_{L_1} = m_1, Z_{L_2} = m_2, Z_{L_3} = m_3) = \sum_{\ell=0}^{d} \pi_\ell M_1(\ell, m_1) \, M_2(\ell, m_2) \, M_3(\ell, m_3),$$

where

$$\pi_\ell = p(U = u^\ell)$$

in the finite-anchor representation.

Since

$$\mathrm{krank}(M_1) + \mathrm{krank}(M_2) + \mathrm{krank}(M_3) \geq r + r + 2 = 2r + 2,$$

Kruskal's theorem (Kruskal, 1977; Allman et al., 2009) implies that the finite-anchor decomposition is unique, up to permutation of the anchor labels. Therefore, for every anchor set

$$\mathcal{U}_r = \{u^0, \ldots, u^d\},$$

the corresponding anchor conditionals and mixing weights are identifiable.

**Step 5: Continuous identification.** We now consider the continuous latent case, where the observed proxy distribution admits the representation

$$p(Z = z) = \int p(z \mid u) \, p_U(u) \, du.$$

The finite-anchor argument in Steps 1–4 applies to arbitrary anchor sets of fixed size $r = d + 1$ throughout the support of $U$. These local finite-anchor identifications constrain both the conditional family and the mixing distribution across the latent support.

By the bounded completeness assumption (Eq. 8), the associated integral operator is injective (Hu & Schennach, 2008). Consequently, two distinct continuous mixing distributions cannot generate the same observed proxy distribution through the same proxy kernel.

Under the structural proxy model (Eq. 4), the conditional family is restricted to the shifted noise family

$$p(Z \mid U = u) = p_\epsilon(Z - g(u)).$$

Together with Eq. 7, this prevents distinct latent states from inducing the same conditional proxy law and prevents alternative conditional families from matching all finite-anchor decompositions.

Therefore, the continuous mixing distribution $p_U$ and the conditional proxy family $\{p(Z \mid U = u)\}_u$ are jointly identified from $p(Z)$ up to a reparameterisation of $u$. This extends the finite-anchor uniqueness established in Step 4 to the continuous latent setting.

**Step 6: Recovering the latent coordinate.** By the joint injectivity and regularity conditions in Eq. 7, distinct latent values induce distinct conditional proxy laws, and the identified conditional family can be indexed by the latent coordinate $u$ itself, up to a smooth invertible change of coordinates. Hence any alternative latent variable $U'$ generating the same observed proxy distribution must satisfy $U' = h(U)$ for some smooth invertible map $h$.

Thus $U$ is identifiable from $p(Z)$ up to smooth invertible reparameterisation.

**Step 7: Conclusion.** Since $r = d + 1$, Kruskal's condition is satisfied whenever

$$k \geq 2r + 2 = 2(d + 1) + 2 = 2d + 4.$$

Together with bounded completeness and the variability of the proxy mechanism, this yields identifiability of the continuous latent confounder $U$ from $p(Z)$ up to smooth invertible reparameterisation.

The bound is a sufficient condition arising from the finite-anchor construction, and is not claimed to be necessary or tight.

$\square$

## B  Proof of Proposition 1

> **Proposition 1 (Compatibility of IntervalGP priors with identifiability and latent reparameterisations)**
>
> Let $U \in \mathbb{R}^d$ denote the latent confounder satisfying the assumptions of Theorem 1, and let
>
> $$F_r \sim \mathcal{GP}(0, k), \qquad r = 1, \dots, d,$$
>
> define the vector-valued latent recovery functions used by the IntervalGP framework. Then:
>
> 1. Introducing IntervalGP-based latent priors does not alter the identifiability of the individual treatment effect (ITE) established in Theorem 1.
>
> 2. For the reparameterisation $h : \mathbb{R}^d \to \mathbb{R}^d$, defined in Eq. (8), replacing $U$ by $\tilde{U} = h(U)$ does not affect the validity of $k$ as a covariance kernel over the latent representation.

*Proof.* (1)  Theorem 1 establishes identifiability of the latent recovery problem under the structural proxy assumptions. The IntervalGP prior is introduced only within the finite-sample variational inference procedure used to learn the posterior latent representation $q_\phi(U \mid Z)$. Since the IntervalGP prior does not modify the structural proxy mechanism implied by Eq. 4, the conditional independence assumption in Eq. 5, or the variability condition in Eq. 7 used in Theorem 1, it does not alter the identifiability result established by Theorem 1.

(2)  By Theorem 1, the latent confounder is identifiable only up to a smooth invertible reparameterisation $h : \mathbb{R}^d \to \mathbb{R}^d$. Thus, an observationally equivalent representation may replace $U$ by $\tilde{U} = h(U)$.

The IntervalGP framework uses a covariance kernel over the observed proxy inputs:

$$\text{Cov}(U_r^i, U_r^j) = k(Z^i, Z^j), \qquad r = 1, \dots, d.$$

The kernel $k$ is defined on the proxy space and depends only on the observed proxy inputs $(Z^i, Z^j)$, rather than on a particular coordinate representation of the latent variable.

Therefore, replacing $U$ by the equivalent representation $\tilde{U} = h(U)$ does not invalidate the use of the same proxy-space kernel in the finite-sample inference procedure. The IntervalGP prior may be applied to whichever representative of the identifiable equivalence class is recovered by the variational encoder.

Consequently, the IntervalGP prior remains compatible with the equivalence class $\{h(U)\}$ of latent representations and does not restrict the reparameterisation freedom implied by the identifiability analysis.

$\square$

*Remark* 1 (IntervalGP posterior smoothing and uncertainty propagation). The proposed IntervalGP framework introduces kernel-based posterior smoothing over latent representations while remaining compatible with the identifiable latent equivalence class established in Theorem 1. Since the latent confounder is identifiable only up to a smooth invertible transformation, the role of the IntervalGP prior is not to enforce a unique latent parameterisation, but rather to provide stable posterior interpolation and uncertainty-aware

inference within the admissible latent space. In addition, the IntervalGP posterior enables interval-valued uncertainty quantification through the encoder-derived latent intervals. These latent intervals are subsequently propagated through the causal outcome model to induce interval-valued posterior uncertainty over the ITE estimates. This provides a principled mechanism for propagating proxy uncertainty into counterfactual prediction when the observed proxies are noisy.

*Remark* 2 (Flexibility of kernel design). Proposition 1 does not require a kernel specifically tailored to the structural proxy functions The kernel instead governs posterior smoothing and local neighbourhood structure within the latent inference space. While an RBF kernel is used as a numerically stable default in the current implementation, alternative kernels (e.g. Matérn, rational quadratic, or non-stationary kernels) may also be employed without affecting the identifiability guarantees established in Theorem 1 or the conclusions of Proposition 1. Since identifiability is derived from the conditional proxy mechanism together with the conditional independence and variability assumptions, the choice of kernel affects only the geometry of posterior inference and uncertainty propagation, rather than the underlying identifiability result itself.

