# OpenReview forum: "IntervalGP-VAE: Learning Unobserved Confounders with Uncertainty for Personalized Causal Effect Estimation"
_TMLR — Decision pending for TMLR_

### Review · Reviewer_EiGS · 2026-04-01

**Summary Of Contributions:**

This paper deals with complexities of confounding adjustment for individual treatment effects (ITEs) when a key confounder is not observed, but several proxies are. As defined in the paper, proxies, $Z$, are variables that causally determine the unmeasured confounder $U$, while $U$ is the confounder in the sense of causally determining treatment $T$ and outcome $Y$. Additional constraints are specified on the proxies, such as mutual independence in Equation 5.

The idea is that perhaps adjustment for proxies will be sufficient to eliminate confounding due to $U$ if only we could recover the structure of $U$  based on the proxies $Z$. The authors propose doing this with a variance of Gaussian Processes to ensure a type of "distance-based" mapping so that patients with similar proxies $Z$ get mapped to similar $U$.

**Audience:**

Yes

**Audience Explanation:**

Topics related to causal inference are likely of broad interest to TMLR readership.

**Claims And Evidence:**

No

**Claims Explanation:**

My expertise is primarily in causal inference so I focus my comments to the causal identifiability component of the papers, which has some deficiencies.

First, the authors make a strong assumption of cross-world independence in Equation 3 defining the ITE. As they say the error $\epsilon_{Y,j}$ for individual $j$ is held constant across the two potential outcomes $Y_j(1), Y_j(0)$. This assumes independence of within-subject potential outcomes which is unrealistic: of course if I told you a patient's outcome under treatment $Y_j(1)$, that probably gives you information about what would have happened under treatment zero $Y_j(0)$. Cross-world independence assumptions are highly controversial because unlike ignorability on page 4, which is at least falsifiable in principle (you could designed a randomized trial), there is no experiment you can ever design where $Y_j(1) \perp Y_j(0)$ is forced to hold. That is, it is in some sense unscientific.

Finally identifiability depends on the temporal ordering between $Z$ and $T$. In the paper, authors assume that $Z\rightarrow U$. But if instead $U \rightarrow Z$, then the existence of a path $Z \leftarrow U_1 \rightarrow Y$ through some other unmeasured factors $U_1$ can be problematic. Specifically, adjusting for $Z$ would open a backdoor path $T\leftarrow U \rightarrow Z leftarrow U \rightarrow Y$. It's tough to draw a DAG in the review box, but try on paper and you'll see. Unfortunately, the DAGs where $Z \rightarrow U$ or $U\rightarrow  Z$ are observationally equivalent -i.e. and so no method, however flexible, can distinguish from the observed data which DAG is true.

This illustrates that identifiability does not hold under the assumptions listed in the paper alone, but rather depends on the topology of the DAG - which cannot be estimated from data in general. The possible topologies become numerous when you consider there are $d$ such unmeasured components in $U$ with observationally equivalent causal relations among them. Thus I remain skeptical of the accuracy and even utility of the main identification results in practice

**Requested Changes:**

I have two suggestions both of which are critical to securing recommendation for aceptance.


1) Instead of thinking about a single vector of finitely many unmeasured confound*ers* I encourage the authors to think about unmeasured confound*ing*

That is, if Z is the set of measured covariates (here "proxies"), unmeasured confounding means that $Y(t) \not \perp T \mid Z$. This implies that even after adjusting for proxies $Z$, the distribution of $Y(t)$ still differs between treatment groups: e.g. $\Delta(z, t) = E[Y(t) \mid T=1, Z] - E[Y(t) \mid T=0, Z] \neq 0$ for each $t$. $\Delta(z, t)$ is what Robins calls the confounding function. See Equation 7 https://onlinelibrary.wiley.com/doi/epdf/10.1002/sim.1657 .

That is, $\Delta(z, t)$ encodes the amount of unmeasured confounding - its direction and magnitude captures the size and gravity of biases induced by departing from "no unmeasured confounding assumption" (i.e. $\Delta(z,t)=0$). This approach is much more general than what authors propose as it does not even require positing a specific finite dimensional latent $U$ and then having to worry about the graphical topology of the $U$ node relative to other nodes. It side-steps unmeasured confounders to just reason about the amount of unmeasured confounding - regardless of the nature of the $U$ that induces it.

Therefore, I encourage the authors to rework their approach around the confounding function and think carefully about how different assumptions about the joint $f(u, z)$ impacts the sign and magnitude of $\Delta(x, t)$. E.g., if $U$ is sufficient to control for confounding, then $U=Z$ with probability one should implies $\Delta(x, t)=0$.


2) Since a major claim of the paper is to extend on existing proximal inference methods (which target-population level quantities) to handle individual-level quantities, I suggest the paper fully grapples with cross-world dependence between potential outcomes rather than naively assuming the errors are the same. This is, in fact, the main identification challenge that distinguishes population vs. unit-level inference but is avoided here. This paper may be helpful in thinking about how population-level quantities like $\Delta$ may be translated to unit-level effects: https://arxiv.org/abs/2508.15016

---

> ### Author Response · Authors · 2026-05-27
> **Acknowledgement of Reviews and Upcoming Revision**
>
> Dear Reviewers,
>
> We sincerely appreciate your high-quality reviews, which are very helpful to us. We are currently working intensively on additional experiments and theoretical clarifications to address your concerns. We will post our revised manuscript and detailed responses shortly to begin the discussion.
>
> Thank you again for your patience and valuable feedback.
>
> Best regards,
> The Authors

---

> > ### Author Response · Authors · 2026-06-22
> >
> > Dear Reviewers,
> >
> > We have added many new experiments and made substantial revisions to the manuscript. We are now at the final revision stage and will upload the revised version this week.
> >
> > Many thanks for your patience.
> >
> > Best regards, The Authors

---

> ### Author Response · Authors · 2026-06-26
>
> **Cross-world dependence and SCM:** We thank the reviewer for the comments. It is important to clarify that in our paper cross-world dependence is modeled through a structural causal model (SCM).
> There are multiple equivalent ways to formalise cross-world dependence in causal inference.
> The work of Oganisian et al. (Oganisian, 2025)(cited by the reviewer) adopts a *direct modeling* approach, in which the joint distribution of $(Y(1),Y(0))$ is specified explicitly, e.g., via a parametric family such as a bivariate normal distribution with a dependence parameter.
> In contrast, our work adopts the SCM framework (Pearl, 2009), , which provides a principled and widely used alternative to direct joint modeling.
> Within SCM, counterfactual outcomes are defined through structural equations of the form
> $Y = f(T,U,\varepsilon),$
> where $U$ represents latent confounders and $\varepsilon$ denotes exogenous noise.
> Under this formulation, the joint distribution of $(Y(1),Y(0))$ is not specified directly, but is instead *induced* through shared latent variables:$Y(1) = f(1,U,\varepsilon), \quad Y(0) = f(0,U,\varepsilon).$
> Thus, cross-world dependence arises naturally from the fact that both potential outcomes are generated using the same realization of $(U,\varepsilon)$.
>
> We emphasise that our problem setting is conceptually consistent with the literature highlighted by the reviewer.
> Both direct joint modeling approaches (e.g., Oganisian et al.(Oganisian, 2025)) and SCM-based approaches recognise that individual treatment effects cannot be recovered from marginal distributions alone, but depend critically on cross-world dependence between potential outcomes. The difference lies in how this dependence is modeled. In practice, both approaches introduce inductive biases, such as, Gaussianity and linearity in direct modeling by Oganisian et al.(Oganisian, 2025), versus structural assumptions such as ANM in SCM. In the revised manuscript, we have added a new subsection 3.4 to clarify this in detail.
>
> The use of shared exogenous variables across potential outcomes is not an unique assumption introduced in our work, but follows directly from the standard definition of counterfactuals in SCM.
> In the classical abduction–action–prediction framework (Pearl, 2009) one first infers the latent variables $(U,\varepsilon)$ for an individual (abduction), and then evaluates counterfactual outcomes under different treatment values while holding these variables fixed (action and prediction).
> This ensures that counterfactual outcomes differ only in the treatment variable, while all other latent factors remain unchanged, which is essential for meaningful individual-level comparisons. **Importantly, this construction induces *dependence*, rather than independence, between $Y({1})$ and $Y({0})$.**
> Therefore, our framework does not rely on any cross-world independence assumption.
> On the contrary, cross-world dependence is explicitly encoded through shared latent variables, consistent with standard SCM semantics. In the revised manuscript, we added a new paragraph in Section 2 to clarify this in details.
>
> Our modeling choice is aligned with a large body of existing work on individual-level causal inference.
> In the revised manuscript under Section ~3.1 we have provided an improved description of the related work, including many exist papers that
> adopt latent-variable formulations in which unobserved confounders are inferred from proxy variables and subsequently used for treatment effect estimation.
> Our work builds on this line of research by providing identifiability guarantees for latent confounder recovery and incorporating uncertainty-aware modeling via interval-valued Gaussian processes - see the Introduction section for our contribution summary.
>
> We further note that our use of an additive noise model (ANM) is a standard structural assumption in causal inference.
> ANM simplifies the functional form of the structural equation and has been widely used in both causal discovery and causal effect estimation due to its favorable identifiability properties.
> While SCM-based counterfactual inference in general requires an abduction step to infer exogenous noise variables, the ANM structure allows us to focus on the recovery of latent confounders, which is the central challenge addressed in this work.
>
> Finally, we fully agree that the identifiability analysis does not hold under any topological structures. Proxy-based approaches need to assume a structure. We give more details in the answer to the next comment.
>
> Reference: Oganisian, A. (2025). Untangling Sample and Population Level Estimands in Bayesian Causal Inference. https://api.semanticscholar.org/CorpusID:280699860
>
> Reference: Pearl, J. (2009). Causality: Models, Reasoning, and Inference. 2nd edition. Cambridge University Press.

---

> > ### Author Response · Authors · 2026-06-26
> >
> > **Structural Assumptions and Identifiability:** We thank the reviewer for the comments. We agree that the role of structural assumptions should be clarified more explicitly. Our paper does *not* claim that Theorem 1 holds for arbitrary causal graphs. Rather, the theorem is stated relative to the proxy-based causal structure already specified in the manuscript through the structural equations and the corresponding SCM (Equation 1 and 4). The key issue here is not whether structural assumptions are needed, but which structural assumptions are appropriate for the target problem. Our work studies the proxy-based latent-confounding setting, where a causal structure linking $U$, $T$, $Y$, and $Z$ is assumed, and then derives additional conditions under which recovery of $U$ and hence ITE estimation become possible. To make this assumption explicit, we have added a structural diagram in Figure 1 of the revised manuscript, illustrating the proxy-generating SCM under which Theorem~1 is stated. Also, key assumptions including the variability assumption have been updated. A new bounded completeness assumption is added.
> >
> > In our opinion, there is nothing unusual about pre-specifying such a structure. Proxy-based causal inference necessarily requires assumptions about how the observed variables relate to the latent confounder. In our setting, the observed variables $Z$ are not arbitrary covariates, but noisy proxy measurements of an underlying latent confounder $U$. Without such a structural relationship, the notion of a ``proxy'' is not well defined, and there is no basis for recovering a latent confounder from observed variables. This is standard in proxy-based methods, including those latent-variable approaches we included in Section 3.1
> >
> >
> > This also clarifies
> > conditioning on $Z$ alone may fail to identify the causal effect does not invalidate our theorem. We fully agree that conditioning on $Z$ alone does not, in general, remove confounding. In fact, this is precisely the reason additional conditions are needed. The role of Theorem~1 is not to justify adjustment on $Z$, but to establish when the latent confounder $U$ can be recovered from the proxy variables $Z$ under further assumptions, namely conditional independence (Equation 5), variability (Equation 6 and 7), bounded completeness (Equation 8) and a sufficient number of proxies. Once $U$ is recovered (up to invertible reparameterisation), individual-level counterfactual inference becomes possible. We have added more discussions in Section ~4.1 to clarify this issue.
> >
> > This point is especially important because the primary goal of our work is *individual treatment effect* (ITE) estimation rather than population-level adjustment. In the presence of hidden confounding, ITE estimation requires access to a confounding-sufficient latent representation: without recovering $U$ (or an equivalent representation), the joint distribution of $(Y(1),Y(0))$ cannot be determined, even if some marginal or population-level quantities are identifiable. Thus, the need to recover $U$ is a consequence of the problem of individual-level causal inference under hidden confounding itself.
> >
> > This distinction also explains why sensitivity-analysis approaches based on confounding functions (e.g., (Brumback et al., 2004)) are not directly applicable in our setting. Such approaches are designed to quantify residual confounding at the population level, typically in terms of biases in mean causal effects such as ATE, while reasoning directly from observed variables.
> > While this may be appropriate for population-level sensitivity analysis, it is insufficient for proxy-based ITE estimation: one cannot simply take an arbitrary set of observed variables $Z$ and infer a latent confounder from them without specifying $Z$ are structurally related to (hence the proxies of) $U$.
> >
> > Reference: Brumback, B. A., Hernan, M. A., Haneuse, S. J. P. A., and Robins, J. M. (2004). Sensitivity Analyses for Unmeasured Confounding Assuming a Marginal Structural Model for Repeated Measures. Statistics in Medicine, 23(5), 749--767.

---

> > ### Comment · Reviewer_EiGS · 2026-07-02
> >
> > Thank you for responding to the comments - I have reread the revised manuscript. The additional paragraph on page 4 starting with "We clarify that holding $\epsilon_{Y,j}$ constant..." is very clarifying. In fact, holding it constant is exactly the cross-world assumption underlying your model. Holding it constant ensures actually that the POs are deterministically related  given the output of $f$ and $\epsilon_{Y,j}$ such that, if you know these, you can always recover the counterfactual PO with certainty if given the factual PO.
> >
> > On page 8, the authors write that all "approaches recognise that individual treatment effects cannot be recovered from marginal distributions alone, but depend critically on cross-world dependence between potential outcomes. The difference lies in how this dependence is modeled: directly via parametric assumptions on $(Y (1), Y (0))$, or indirectly via structural equations and latent variables."  I agree but, if the authors also agree, I would consider making the more specific point that it is done indirectly via assumption of a common error term $\epsilon_{Y,j}$. The authors claim this is not an assumption, but I do think it is in fact the crucial assumption that governs cross-world relationship.

---

> > > ### Author Response · Authors · 2026-07-03
> > >
> > > **Clarification of the Cross-World Assumption and Common Error Term.** We thank the reviewer for this helpful suggestion. We agree that a common error term constitutes a modelling assumption underlying our framework and therefore deserve to be stated more explicitly.
> > >
> > > To address this point, we have revised Section~2 by introducing an assumption explicitly, namely *SCM-based Counterfactual Consistency*. This assumption states that both potential outcomes are generated from the same realization of the latent confounders and exogenous noise, so that the cross-world dependence between the potential outcomes is induced through the shared latent variables and exogenous noise within the SCM.
> > >
> > > In addition, we have revised the paragraph in Section~2 where we previously stated that holding $\epsilon_{Y,j}$ fixed is ``not an additional assumption''. We agree that this wording could be misleading. The revised text now explains that the cross-world dependence is induced through the shared latent confounders and exogenous noise in the SCM, and that holding (U) and $\epsilon_{Y,j}$ fixed across interventions is the standard SCM mechanism for defining counterfactual outcomes.
> > >
> > > The newly added text is highlighted in **magenta**.

---

### Review · Reviewer_LLQ5 · 2026-05-02

**Summary Of Contributions:**

The paper proposes IntervalGP-VAE, a VAE-based framework for ITE estimation under unobserved confounding, with two interval-valued GP components: a GP prior over the latent confounder $U$ as a function of the proxies $Z$, and a GP regression head that produces ITE intervals. On the theory side, the authors claim (Theorem 1) that under conditional independence of proxies and a variability condition, $k \geq 2d + 4$ proxies suffice for identifying $U \in \mathbb{R}^d$ up to a smooth invertible reparameterization, and (Theorem 2) that the structural ITE is invariant under such reparameterizations. Experiments cover 24 synthetic configurations and 100 IHDP replications, against TEDVAE.

The composition is sensible — propagating proxy noise through to ITE intervals via an interval-output GP head is a reasonable engineering idea, and as far as I know this combination is novel. Proposition 1 (GP prior compatible with identifiability) is correct and useful framing.

That said, I have several concerns. The proof of Theorem 1 has a gap in the discrete-to-continuous limit. Theorem 2 is essentially a tautology. The training-time GP prior collapses to a per-sample univariate penalty (the authors note this in Sec 5.4), which contradicts the motivational claim that the prior induces *latent-space coherence* during learning. Empirically, only TEDVAE is used as a baseline, $d = 1$ throughout, and on IHDP the empirical 90% interval coverage is 53%. Finally, the paper sits oddly relative to the proximal causal inference literature it gestures at but doesn't really engage with.

**Additional Comments:**

A few smaller things. Table 1 lists $\hat Y(t)$ but the paper uses both $\hat Y(t)$ and $Y_j(t)$ — please harmonize. Some inline math in Eqs. 10 -12 has spacing glitches in the PDF; in particular, please confirm whether a noise variance is being added at test time in $K_{ss} = k(Z^{\star}, Z^{\star}) + \sigma^2$.

Sec 5.3 mentions a "staged strategy" but only a single joint training stage is described — either remove "staged" or describe the stages. The level $\alpha$ used in Eq. 13 isn't specified in the main text (it appears to be 5%).

**Audience:**

Yes

**Audience Explanation:**

Uncertainty-aware ITE estimation under proxy-recovered confounding is something readers working on GP-VAE hybrids, latent-variable causal models, and uncertainty-aware policy evaluation would be interested in.

**Broader Impact Concerns:**

The paper has no Broader Impact statement --- or at least I did not see one. That said, since the motivating examples (healthcare, economics, social sciences) involve consequential decisions and the stated value-add is *calibrated uncertainty for downstream decision-making*, a short discussion would be appropriate.

**Claims And Evidence:**

No

**Claims Explanation:**

**Theorem 1.** The proof combines a discretization argument with Kruskal's theorem (via Allman–Matias–Rhodes) and then takes a limit to recover the continuous model. The discrete step is fine, but the limiting argument in Step 5 needs a regularity condition that's never invoked. Kruskal gives identifiability of a finite mixture of dimension $r$ when $k \geq 2r+2$. To upgrade to a continuous latent you need $r \to \infty$ with $k$ fixed, and Kruskal's rank condition fails at sufficiently fine resolutions. The standard fix is a completeness or analyticity assumption --- Hu and Schennach (Econometrica 2008) is the canonical reference, and Kivva et al. (NeurIPS 2022) handle a closely related case. Without something in this neighborhood, I can't see how the limit goes through.

This connects to the second issue: the variability condition (Eq. 6) is stated as "$p(Z_i \mid U = u_1) \neq p(Z_i \mid U = u_2)$ for many $i$," which is informal. What you actually need is something like bounded completeness of $Z$ wrt $U$ --- i.e., the conditional expectation operator $g \mapsto E[g(U) \mid Z]$ is injective on the relevant function space. Pointwise non-equality of conditionals is necessary but not sufficient; two distributions can be pointwise distinct while still allowing nontrivial $g(U)$ to average to zero under conditioning. So Eq. 6 isn't strong enough to make the proof work, and "for many $i$" needs to be made precise — at minimum tracing the implied lower bound through the Khatri–Rao rank computation in Step 3.

There's also an issue with Step 7's choice of $r = d+1$. The justification ("to preserve $d$ degrees of freedom of $\mathcal{M}$") is a topological appeal rather than an argument, and it's what produces the headline $2d+4$ bound. Why not $r = d+2$ giving $k \geq 2d+6$? The bound is presented as tight, but its tightness isn't established.

**Unstated assumptions on $g$.** The proxy model in Eq. 4 only requires additive independent noise, with the $g_i$ described as "unknown but sufficiently non-redundant." But Step 6 of the proof says "the noise distribution and the image manifold $\{g(u)\}$ are determined up to a smooth change of coordinates," which seems to require quite a bit more than the paper states. A few questions to help me understand what's being assumed:

- Is the joint map $g = (g_1, \ldots, g_k): \mathbb{R}^d \to \mathbb{R}^k$ assumed to be injective? The variability condition (Eq. 6) is at the level of marginal conditionals $p(Z_i \mid U)$, which is weaker than joint injectivity, but joint injectivity seems necessary for the recovery argument to go through. Could you state this explicitly if it's intended?
- What smoothness on $g_i$ is assumed? "Image manifold" and "smooth change of coordinates" suggest at least $C^1$ with non-degenerate Jacobian, but this isn't stated. Closely related work (Khemakhem et al. AISTATS 2020; Hyvarinen et al. AISTATS 2019) is explicit about this --- it would help to follow that convention.
- Is there a non-degeneracy condition on the $g_i$ collectively? The Khatri-Rao rank computation in Step 3 needs the per-proxy conditional matrices to have generic rank $\geq 2$, which translates into a condition on how each $g_i$ varies locally. The synthetic experiments (Table 2) all use smooth, mostly-monotone functions — it would be useful to know which features of those choices are doing the work, and what would break the argument.
If these conditions are intended, stating them up front would let readers gauge applicability to their own settings; if they're not intended, the proof needs adjustment.

**Theorem 2.** Once $\hat U = h(U)$ is plugged into the same structural equation $f$ via the implicit reparameterization $f \circ h^{-1}$, invariance follows by composition. This is correct but doesn't say much. The substantive question — whether the *learned* outcome head, fit to data drawn through some unknown $h$, recovers the correct ITE up to the invariance class — is an estimation question, and it's not addressed.

**Latent-space coherence vs. the actual training prior.** Section 5.1 motivates the GP prior by saying that "individuals with similar proxy profiles induce similar latent posterior distributions." But §5.4 explicitly states that the training-time prior "depends only on the marginal GP distribution at each point $Z^i$ … and not on cross-point correlations," making it "fully local, univariate, and independent across data points." So during training the regularizer is a sum of per-point Gaussian-mass-in-an-interval terms — equivalent up to constants to a per-point KL toward $\mathcal{N}(0, k(Z^i, Z^i))$. There's no joint kernel structure during training, so there's no mechanism for "similar $Z$ produces similar $\hat U$" beyond what any per-point prior with input-dependent variance would give. The "GP-induced coherence" only enters at test time. Without an ablation against (i) a plain $\mathcal{N}(0, I)$ prior with input-dependent scaling and (ii) a true joint GP prior, I can't tell whether the GP is doing the work the paper says it does.

**Calibration and comparative claims.** The abstract claims "calibrated ITE intervals" and that the method "outperforms existing methods." Synthetic coverage is ~95% for nominal 90% (overcoverage). IHDP coverage is 53%. The paper attributes the IHDP gap to identifiability assumption violations, which is fair, but that concession is more damaging than the authors acknowledge — calibration is the headline contribution, and it's shown to fail on the only realistic benchmark. On performance, Figure 4 shows TEDVAE achieving lower PEHE (3.39 vs. 3.52) and lower ATE error on IHDP, so "outperforms" isn't supported either.

**Positioning relative to proximal causal inference.** This is more of a framing issue but it matters for evaluating the contribution. The paper cites Miao et al. (2018) but doesn't really engage with the proximal CI program (Cui-Pu-Shi-Miao-Tchetgen Tchetgen on semiparametric proximal CI; Tchetgen Tchetgen and collaborators on the broader program; Kompa et al. on uncertainty in counterfactual prediction). Proximal CI uses two structurally distinct proxies (treatment-side and outcome-side) and identifies effects via bridge functions, sidestepping recovery of $U$ entirely under bounded completeness. This paper is doing something different — it's in the latent-variable measurement-error tradition (Kuroki-Pearl 2014; Allman-Matias-Rhodes 2009; CEVAE; iVAE; Khemakhem et al. AISTATS 2020), where many redundant proxies all measure $U$ and you recover $U$ itself up to an equivalence class. Both lineages need a "$Z$ varies enough w.r.t. $U$" condition, but they cash it out differently — completeness in proximal CI, generic Kruskal-rank conditions plus a regularity assumption in the latent-variable approach. The paper would be much clearer if it positioned itself explicitly in the second lineage and explained why it isn't doing proximal CI.

A related point: the role of $Z_Y$ in Eq. 1 is unclear. If $Z_Y \subseteq Z$ is generated by $U$ via Eq. 4 and *also* directly affects $Y$, you have a mediated path $U \to Z_Y \to Y$ in addition to $U \to Y$, and Footnote 1 holds $Z_Y$ fixed across counterfactual worlds without much discussion. This deserves explicit treatment — what is $Z_Y$, exactly, and how does it differ from the rest of $Z$?

**Requested Changes:**

The proof of Theorem 1 needs to be repaired or replaced. Either invoke a completeness or analyticity assumption that justifies the discrete-to-continuous limit (Hu–Schennach completeness is the standard reference), or adapt a result that handles the continuous case directly. Make the variability condition (Eq. 6) precise — "for many $i$" needs an explicit lower bound that traces through the rank computation in Step 3. And justify why $r = d+1$ is the smallest sufficient choice in Step 7, or restate the result conditionally as "for some $r \geq d+1$, $k \geq 2r+2$" and discuss tightness honestly.

State the assumptions on the proxy mechanism $g$ explicitly. As written, Eq. 4 only requires additive independent noise, but the proof seems to need joint injectivity of $g = (g_1, \ldots, g_k)$, smoothness (likely $C^1$ with non-degenerate Jacobian), and a non-degeneracy condition on how each $g_i$ varies locally. Following the convention of Khemakhem et al. (AISTATS 2020) and stating these up front would let readers gauge applicability to their own data.

I would either downgrade Theorem 2 to a remark or replace it with a substantive estimation result.

The contradiction between the motivational claims and the actual training prior needs to be resolved. Three options seem reasonable: rewrite §1, §5.1, §5.2 to drop the "latent-space coherence during training" framing; replace the per-sample regularizer with a true joint GP prior (or an SVGP approximation) and re-run experiments; or add an ablation comparing the current per-sample prior, a plain $\mathcal{N}(0, I)$ KL with input-dependent variance, and a joint GP prior. Whichever path, make the active ingredient unambiguous.

A single comparison to TEDVAE isn't enough for a paper centered on uncertainty quantification. At minimum: CEVAE (Louizos et al. 2017), Deep Proxy Causal Learning (Xu et al. NeurIPS 2021), and at least one method that natively produces ITE intervals — BART (Hill 2011), Bayesian Causal Forest (Hahn et al. 2020), or conformal CATE (Lei and Candès 2021; Alaa et al. 2023). Without an interval-producing baseline I genuinely can't tell whether the *interval* part of IntervalGP-VAE is doing useful work, or whether any reasonable Bayesian or conformal CATE method would deliver better-calibrated intervals on the same data.

Run experiments at $d \geq 2$. The whole quantitative content of the identifiability theorem is the proxy-count requirement scaling with $d$, and at $d = 1$ the bound asks for 6 proxies — barely a constraint. $d = 2, 3$ with the corresponding $k$ would actually stress the theory.

The IHDP result needs to be characterized more honestly. 53% empirical coverage for nominal 90% is a big gap. Either add a benchmark — ACIC 2016/2018, or a synthetic setup where the $g_i$ are deliberately partially informative — that interpolates between the synthetic regime where calibration holds and the IHDP regime where it fails, or qualify the calibration claim in the abstract and intro to explicitly condition on the identifiability assumptions holding. The "outperforms existing methods" claim should be removed or substantiated.

Position the paper relative to proximal CI explicitly. This isn't proximal causal inference — it's latent-variable causal inference in the Kuroki-Pearl / Allman-Matias-Rhodes / CEVAE tradition with sharper identifiability bookkeeping. Saying so directly, and citing the relevant proximal CI work (Cui et al., Tchetgen Tchetgen and collaborators, Kompa et al.) to clarify what this paper is *not* doing, would help readers locate the contribution. Relatedly, please clarify what $Z_Y$ is and how it differs from the rest of $Z$.

Finally, please clarify which ITE interval is reported at test time. Eq. 13 defines the bounds via quantiles of samples from the outcome head; Eq. 12 gives a separate GP-based posterior interval; Algorithm 1 references only Eq. 12. A short paragraph and an updated Algorithm 1 would resolve this.

The paper notes compatibility with Matérn, rational quadratic, and non-stationary kernels but uses RBF throughout — a small ablation across kernels would substantiate the claim. Similarly, an experiment at $n \geq 10{,}000$ using a sparse / inducing-point GP would validate the scalability claim in §5.4.

The title's "Personalized" overstates what's shown; something like "Uncertainty-Aware ITE Estimation under Proxy Confounding" would more accurately reflect the contribution. In Figure 2, the "Sample Index" axis is uninformative — sorting by true ITE would make the calibration visually interpretable. Reporting TEDVAE's training time on the same hardware would let readers assess the cost of the GP components. Per TMLR's reproducibility expectations, an anonymous code link should be added.

---

> ### Author Response · Authors · 2026-05-27
> **Acknowledgement of Reviews and Upcoming Revision**
>
> Dear Reviewers,
>
> We sincerely appreciate your high-quality reviews, which are very helpful to us. We are currently working intensively on additional experiments and theoretical clarifications to address your concerns. We will post our revised manuscript and detailed responses shortly to begin the discussion.
>
> Thank you again for your patience and valuable feedback.
>
> Best regards,
> The Authors

---

> > ### Author Response · Authors · 2026-06-22
> >
> > Dear Reviewers,
> >
> > We have added many new experiments and made substantial revisions to the manuscript. We are now at the final revision stage and will upload the revised version this week.
> >
> > Many thanks for your patience.
> >
> > Best regards, The Authors

---

> ### Author Response · Authors · 2026-06-26
>
> Many thanks and the comments are very helpful in improving the paper. We have addressed all the issues - please see more details below.
>
> **Completeness assumption for the continuous latent setting.** Following the reviewer's suggestion and the related literature, we now introduce an explicit *Bounded Completeness Assumption* in Section 2 to justify the transition from the finite-mixture setting to the continuous latent setting.
>
> **Revised variability assumption.**  We agree that the original formulation of Eq. (6) was informal and insufficient for the proof. The variability condition has therefore been reformulated as a collection of precise structural assumptions, including: (i) joint injectivity of the proxy mechanism, (ii) partitionable proxy variability, and (iii) non-degenerate additive noise. These assumptions replace the previous statement that the conditional distributions differ "for many i" and make explicit the conditions required for the rank arguments used in the proof. The revised assumptions are presented in Section 2 and are referenced directly throughout the updated proof. This covers the assumptions involving joint injectivity of the proxy generation mechanism $g = (g_1, ..., g_k)$. Partitionable proxy variability further requires that the proxies can be partitioned into multiple conditionally independent subsets, each of which remains informative about the latent confounder. This assumption guarantees that the conditional probability matrices associated with the proxy groups achieve the ranks required by the Kruskal-based identifiability argument. As a result, the revised proof can explicitly trace how proxy variability translates into the rank conditions needed for latent confounder recovery.
>
> **Revised statement of the proxy-number requirement.**  We appreciate the reviewer's observation regarding the choice of $r = d + 1$. We have revised the presentation to clarify that the resulting condition $k >= 2d + 4$ is intended as a sufficient condition rather than a necessary or tight bound. We now explicitly state the result in terms of a latent discretization level $r >= d + 1$, yielding the sufficient condition $k >= 2r + 2$. We further explain that the choice $r = d + 1$ is motivated by preserving the intrinsic dimensionality of the latent manifold in the discretized argument, but we do not claim that this is the smallest sufficient value. These discussions have been revised accordingly in Section 4.1.
>
> **Revised proof of Theorem 1.**  The proof in Appendix A has been substantially rewritten to incorporate the new assumptions and to clarify the role of each step.
>
> **Replacing the invariance theorem with a substantive finite-sample estimation analysis.** We thank the reviewer for this important observation. We agree that the original Theorem 2 established only an invariance property under an invertible transformation of the latent confounder. In response, we have revised the manuscript in two ways. First, we have removed Theorem 2 as a standalone theorem and replaced it with a dedicated discussion subsection, Section 4.2. The revised text now explicitly presents the result as an invariance property rather than as a theorem. Second, we agree that the practically relevant question concerns estimation rather than invariance alone. To support this analysis, we introduce a bi-Lipschitz regularity assumption on the transformation $h$, ensuring that distances in the latent space are preserved up to constant factors under the identifiability equivalence class. Furthermore, we have added a new subsection, Section 5.5, that studies the estimation properties of the proposed framework. In particular, we now provide an explicit bound on the ITE estimation error that links inaccuracies in latent confounder recovery to downstream errors in treatment effect estimation. Combined with standard regularity assumptions on the outcome model, this allows us to establish that errors in estimating the latent representation induce controlled errors in the estimated ITE.

---

> > ### Author Response · Authors · 2026-06-26
> >
> > **Clarifying GP-induced coherence and validating the latent prior through ablations.** We thank the reviewer for this careful observation. We agree that the original presentation did not sufficiently distinguish between the role of the GP prior during training and its role during inference. In response, we have adopted the first option suggested by the reviewer and substantially revised Sections 5.1 and 5.2, together with the introduction in Section 1, to clarify the active role of the GP component within our framework. Specifically, the revised manuscript presents the proposed IntervalGP-VAE within a unified Bayesian framework, where the GP serves two distinct purposes: (i) providing a point-wise interval-based regularization term during training through the ELBO objective, and (ii) enabling coherent posterior interpolation and uncertainty propagation at inference time through the GP posterior. To support inference-time posterior prediction, we have introduced an explicit smoothness assumption on the posterior recovery map from proxies to latent confounders. This regularity condition justifies the use of kernel-based GP posterior interpolation during inference, since nearby proxy observations are expected to induce nearby posterior latent representations, enabling stable posterior smoothing between neighbouring observations. In addition, we have added a new latent-prior regularization ablation study in Section 6.1.1 to make the active ingredient empirically clearer. The revised study compares the default per-sample IntervalGP regularizer with a standard isotropic Gaussian prior, an input-dependent Gaussian prior, joint mini-batch GP regularizers, and sparse inducing-point GP approximations. All variants use the same architecture, training protocol, hyperparameters, and data-generating settings, differing only in the latent prior regularization imposed on $q(u | z)$. The results show that the per-sample IntervalGP regularizer provides the most favourable overall balance between point-estimation accuracy, interval sharpness, empirical coverage, calibration, and latent recovery. The input-dependent Gaussian prior performs substantially worse, indicating that the observed gains cannot be explained merely by input-dependent variance scaling. Joint and sparse GP regularizers can achieve competitive point-estimation accuracy, but in this setting they tend to produce overly narrow intervals and weaker coverage. These findings justify the use of the per-sample IntervalGP regularizer as an efficient default choice, while clarifying that full joint GP regularization remains a possible extension when additional computational cost and calibration trade-offs are acceptable.
> >
> > **Calibration and comparative claims.**  We agree that the original wording, especially the claim that IntervalGP-VAE "outperforms existing methods", was too strong. The revised manuscript no longer presents the method as universally outperforming all alternatives. Instead, we now state more carefully that IntervalGP-VAE achieves accurate ITE estimation, reliable latent recovery, and well-calibrated ITE intervals on identifiable synthetic datasets, while providing competitive PEHE and ATE estimation, sharper intervals, and lower computational cost on the semi-synthetic IHDP benchmark. This revision is important because the two experimental regimes have different meanings. The synthetic datasets are deliberately constructed to satisfy the proxy-identifiability assumptions, including conditional independence, sufficient variability, and proxy richness. In this favourable regime, calibrated interval estimation is expected and empirically supported. By contrast, IHDP is not designed as a proxy-identification benchmark. The retained continuous covariates may be weakly informative, redundant, or statistically dependent beyond the latent confounder, and therefore may violate the assumptions required by Theorem 1. We now explicitly describe this limitation and no longer interpret IHDP as evidence of universal calibration. Instead, we present it as a realistic stress test under possible proxy-assumption violations.

---

> > > ### Author Response · Authors · 2026-06-26
> > >
> > > **Expanded baseline comparison.** We agree that a single comparison to TEDVAE is insufficient for a paper centred on uncertainty quantification. In the revised manuscript, we have expanded the baseline comparison to include both point-estimation and interval-producing methods. Specifically, we added:
> > >
> > > * **CEVAE**, as a classical VAE-based latent-confounder baseline;
> > >
> > > * **DeepPCL-like proxy learning**, as a proxy-based causal representation baseline;
> > >
> > > * **BART/BCF-style ensemble**, as a tree-based uncertainty-aware baseline;
> > >
> > > * **Conformalized TARNet**, as a conformal CATE-style interval-producing baseline;
> > >
> > > * **Conformalized TEDVAE**, as a strong oracle uncertainty baseline based on a closely related latent-variable method.
> > >
> > > The revised baseline results show that, on the identifiable synthetic settings, IntervalGP-VAE achieves the lowest average PEHE and ATE error among the compared methods, while maintaining well-calibrated ITE intervals. Conformalized TEDVAE also provides reliable interval coverage, but at a substantially higher computational cost. The BART/BCF-style baseline produces comparatively narrow intervals but suffers from severe under-coverage, whereas Conformalized TARNet attains very conservative coverage by producing substantially wider intervals. Overall, these results provide a more balanced comparison between point-estimation methods and interval-producing methods.
> > >
> > > **Experiments for $d \ge 2$ and the proxy-count condition.**  We agree with the reviewer that experiments with only a scalar latent confounder do not sufficiently stress the proxy-count implication of Theorem 1. The main quantitative content of the theorem is that the required number of proxies scales with the latent dimension $d$. In response, we have added a new scaling analysis of proxy count and hidden-confounder recovery in Section 6.1.5. Specifically, we now consider latent confounders $U \in R^d$ with $d \in \{(1,2,3,4,5)\}$, and vary the number of proxies below, at, and above the theoretical threshold $k = 2d + 4$. For each pair $(d,k)$, we repeat the experiment over five random seeds. The proxy mechanism is constructed so that, when $k >= 2d + 4$, the generated proxies can be partitioned into three blocks satisfying $$
> > > |L_1| = d + 1,\quad |L_2| = d + 1,\quad |L_3| \geq 2.
> > > $$ When $k < 2d + 4$, the same proxy-generation rule is used, but the block-size condition is intentionally violated. This allows us to directly compare sub-threshold and at/above-threshold regimes under a controlled data-generating mechanism.
> > >
> > > The results show that the effect of the proxy-count condition becomes more visible as the latent dimension increases. For $d = 1$, the task is relatively easy even below the theoretical threshold, suggesting that a small number of informative nonlinear proxies may already be sufficient for finite-sample latent recovery. In contrast, for $d >= 2$, moving from the sub-threshold regime to the at/above-threshold regime consistently improves PEHE, ATE error, empirical coverage, and smoothed latent correlation. This supports the intuition that additional proxies become increasingly important when the latent confounder is multidimensional, since they provide information about different latent directions. At the same time, the empirical transition around $k = 2d + 4$ is gradual rather than abrupt. We therefore interpret the proxy-count condition in Theorem 1 as a conservative sufficient condition for structural proxy richness, rather than as a necessary condition or a deterministic finite-sample guarantee. Settings below this threshold are not necessarily unidentifiable; they simply fall outside the regime for which Theorem 1 provides a formal identifiability guarantee. The experiment therefore directly stresses the scaling implication of the theory while also clarifying the distinction between structural identifiability and finite-sample empirical performance.

---

> > > > ### Author Response · Authors · 2026-06-26
> > > >
> > > > **More realistic characterization of IHDP.** We agree that the IHDP result needs to be characterized more carefully. In the revised manuscript, we explicitly distinguish between identifiable synthetic settings and the semi-synthetic IHDP benchmark. The original manuscript did not sufficiently emphasize that IHDP may violate the proxy-identifiability assumptions. The revised discussion now states that the lower IHDP coverage should not be viewed as a failure of the interval mechanism alone, but as evidence that the proxy-richness and variability assumptions may not hold strongly in this benchmark.
> > > >
> > > > To bridge the gap between the identifiable synthetic regime and the IHDP regime, we added a controlled proxy-quality and assumption-violation analysis. In this experiment, we gradually weaken the proxy mechanism by increasing proxy noise, reducing the number of informative proxies, and introducing redundancy or dependence among proxies. The goal is not to show that IntervalGP-VAE remains calibrated under arbitrary violations, but to demonstrate how ITE estimation and interval calibration degrade as the assumptions required for proxy-based identifiability are weakened. This new experiment directly addresses the reviewer's request for an intermediate benchmark between favourable synthetic settings and the more challenging IHDP setting.
> > > >
> > > > **Clarification of the reported test-time ITE interval.** In the revised manuscript, we clarify the distinction between the latent GP posterior interval and the final ITE interval. The latent GP posterior first produces an interval-valued posterior over the recovered latent confounder. This latent interval is then propagated through the causal outcome model. The final reported ITE interval is obtained from the ITE-space posterior, which incorporates uncertainty from latent posterior sampling, latent GP smoothing, and full GP posterior propagation in the ITE space.
> > > >
> > > > We also added an ablation study on ITE interval uncertainty sources to clarify the role of each component. The point-only variant has degenerate intervals and therefore no meaningful coverage. The Monte Carlo latent sampling variant captures encoder posterior uncertainty and gives conservative but reliable intervals. The latent-GP-only variant produces overly narrow intervals and under-covers. The full GP-ITE posterior variant achieves the best balance between PEHE, interval width, empirical coverage, and interval score. This supports the use of the full GP-ITE posterior mechanism as the reported test-time ITE interval estimator.
> > > >
> > > > **Kernel ablation.** We agree that the original statement about compatibility with kernels beyond RBF required empirical support. In the revised manuscript, we added a kernel ablation study comparing RBF, Matern-1/2, Matern-3/2, Matern-5/2, rational quadratic, and linear-plus-RBF kernels. The results show that IntervalGP-VAE remains stable across all tested kernels: PEHE remains low, empirical coverage remains consistently above the nominal 90% level, and latent recovery remains strong. The RBF kernel remains a strong default, while rational quadratic and Matern variants also perform competitively. This substantiates the claim that the framework is not tied to a single kernel choice.
> > > >
> > > > **Scalability and sparse inducing-point GP approximation.** We agree with the reviewer that the scalability claim should be supported by experiments at larger sample sizes. In the revised manuscript, we have added a scalability analysis using synthetic datasets with sample sizes $n \in \{(1000, 5000, 10000)\}$. We also compare the current per-sample IntervalGP prior with sparse inducing-point IntervalGP variants using $M_{\mathrm{ind}} \in \{(32,64,128)\}$ inducing points.
> > > >
> > > > The results show that the proposed implementation scales efficiently to $n = 10000$. The per-sample IntervalGP prior remains the strongest default, achieving the best accuracy-runtime trade-off at larger sample sizes. The sparse inducing-point variants are stable and competitive, with performance generally improving as the number of inducing points increases, but they incur additional kernel-computation costs in the current implementation. All variants maintain high empirical coverage at larger sample sizes, although the intervals become conservative as  $n$ increases. The width comparison does not show a consistent advantage for sparse inducing-point regularization across all sample sizes. Overall, the results support sparse inducing-point IntervalGP regularization as a feasible scalability extension, while confirming that the per-sample IntervalGP prior remains an efficient and effective default in the present implementation.

---

> > > > > ### Author Response · Authors · 2026-06-26
> > > > >
> > > > > **Positioning relative to proximal causal inference and clarifying $Z_Y$.** We thank the reviewer for this insightful comment. We agree that the original manuscript did not sufficiently distinguish our approach from the proximal causal inference framework, and that the role of $Z_Y$ required further clarification. In response, we have substantially revised Section 3.1 to position our work more explicitly within the latent-variable causal inference literature. Specifically, we now clarify that our framework belongs to the latent-confounder recovery tradition, where multiple proxies are generated from a common latent confounder and the objective is to recover the latent confounder itself, up to an equivalence class, in order to support downstream counterfactual estimation. We contrast this setting with proximal causal inference, which uses two structurally distinct sets of proxies and identifies causal effects through bridge functions under bounded-completeness assumptions, without requiring recovery of the latent confounder. In addition, we have expanded Section 2.1 to provide a more precise description of the role of $Z_Y$. We further clarify why $Z_Y$ is held fixed across counterfactual worlds in the counterfactual framework.
> > > > >
> > > > > **Title revision.** In response to the reviewer's concern that the original title overstated the scope of the contribution, we have revised the title to:
> > > > >
> > > > > *IntervalGP-VAE: Uncertainty-Aware Individual Treatment Effect Estimation via Identifiable Proxy-Based Latent Confounder Recovery*
> > > > >
> > > > > This revised title emphasizes the three central components of the paper: uncertainty-aware ITE estimation, proxy-based latent confounder recovery, and identifiability.
> > > > >
> > > > > **Figure 2 and visual calibration.** We agree that using the raw sample index on the horizontal axis limits the informativeness of the calibration plot. In the revised manuscript, we have updated the interval plots for both the recovered hidden confounder and the estimated ITE by sorting individuals according to their true $U$ values and true ITE values (see Figure 4).
> > > > >
> > > > > **Training time and computational cost.** In the revised experiments, we report the runtime of all baseline methods, including TEDVAE, for both the synthetic experiments, see Table 4, and the semi-synthetic experiments, see Section 6.2.3, using the same hardware setting. We also clarify at the beginning of Section 6 that all experiments were rerun on upgraded hardware with an improved configuration.
> > > > >
> > > > > **Code availability.** In response to the reviewer and to support TMLR's reproducibility expectations, we have added an anonymous code link:
> > > > > https://github.com/anonymous-for-tmlr/IntervalGP-VAE.
> > > > > This link will be included in the final manuscript to facilitate reproducibility of the experiments.
> > > > >
> > > > > **Broader impact statement.** We agree that a short broader impact discussion is appropriate, especially
> > > > > because the motivating applications include healthcare, economics, and social
> > > > > sciences, where treatment-effect estimates may inform consequential decisions.
> > > > > We have added a Broader Impact Statement after the Conclusion. The added
> > > > > discussion describes both the potential benefits and limitations of the proposed
> > > > > method. In particular, we emphasize that uncertainty-aware ITE estimation can
> > > > > help avoid overconfident decision-making when latent confounders are recovered
> > > > > from noisy proxies. At the same time, we clarify that IntervalGP-VAE should not
> > > > > be used as an automated decision-making tool without careful validation of the
> > > > > proxy assumptions, domain-specific assessment, sensitivity analysis, and human
> > > > > oversight. We also note that poor-quality, biased, redundant, or dependent proxy
> > > > > variables may lead to unreliable latent recovery and potentially misleading
> > > > > treatment-effect intervals.
> > > > >
> > > > > **Notation for potential and estimated outcomes.** We have revised the notation to improve consistency. In the original manuscript, Table 1 listed $\hat Y(t)$, while the main text also used $Y_j(t)$ for potential outcomes. We now distinguish these quantities explicitly: $Y_j(t)$ denotes the structural potential outcome for individual $j$ under treatment value $t$, whereas $\hat Y_j(t)$ denotes the model-estimated counterfactual outcome. Correspondingly, the ITE is defined structurally as
> > > > > $
> > > > >     \tau_j = Y_j(1)-Y_j(0),
> > > > > $
> > > > > while the estimated ITE is denoted by
> > > > > $
> > > > >     \hat\tau_j = \hat Y_j(1)-\hat Y_j(0).
> > > > > $
> > > > > We have harmonized the notation in Table 1 and throughout the manuscript accordingly.
> > > > >
> > > > > **GP posterior equations and test-time noise variance.** In the revised manuscript, we have added this discussion under ``Role of kernel hyperparameters during training'' in Section~5.1. Our setting differs from standard GP regression because uncertainty is already represented through the encoder posterior variance $(\sigma_u^i)^2$, which induces the interval-valued latent representation. Consequently, the encoder-derived latent intervals absorb the role typically played by GP observation-noise variance.

---

> > > > > > ### Author Response · Authors · 2026-06-26
> > > > > >
> > > > > > **Training strategy.** We agree that the original description of the ``staged strategy'' was unclear. We have revised Section 5.3 to explicitly describe the stages used in implementation. The training procedure consists of: (i) joint training of the VAE encoder/decoder and outcome head using the proxy reconstruction, outcome prediction, KL, and IntervalGP regularization terms; (ii) an outcome-head refinement stage, in which the latent representation is fixed or weakly updated while improving counterfactual prediction; and (iii) an optional VAE refinement stage used to stabilize the latent posterior intervals. We now describe these stages explicitly in Section 5.3.3 and ensure that the description is consistent with Algorithm 1 and the implementation.
> > > > > >
> > > > > > **Interval level $\alpha$.** We use nominal 90\% ITE intervals, corresponding to $1-\alpha=0.90$ and hence $\alpha=0.10$. For the two-sided central interval, the Gaussian quantile is $z_{1-\alpha/2} = z_{0.95} \approx 1.645$. We have clarified this notation in Algorithm~1 and specified the interval level in the experimental settings.

---

> > > > > > > ### Comment · Reviewer_LLQ5 · 2026-06-26
> > > > > > > **Thank you so much for revision**
> > > > > > >
> > > > > > > Thank you so much for substantially updating and reworking the manuscript, and I also appreciate a very detailed response. I do not have many further concerns. :-)

---

> > > > > > > > ### Author Response · Authors · 2026-06-27
> > > > > > > > **Many thanks.**
> > > > > > > >
> > > > > > > > Dear Reviewer,
> > > > > > > >
> > > > > > > > Thank you very much for your kind and encouraging feedback. We are glad to hear that the revisions have addressed your concerns. Again, we sincerely appreciate your high-quality suggestions, which have significantly improved the quality of our manuscript.
> > > > > > > >
> > > > > > > > Many thanks.
> > > > > > > >
> > > > > > > > Kind
> > > > > > > > Authors

---

### Review · Reviewer_B7U7 · 2026-05-13

**Summary Of Contributions:**

This paper studies individual treatment effect estimation in the presence of unobserved confounding. The authors propose IntervalGP-VAE, a latent variable model that uses observed proxy variables to infer the distribution of an unobserved confounder. The method combines a VAE based latent representation with an interval valued Gaussian Process component, with the goal of quantifying uncertainty in latent confounder recovery and propagating this uncertainty to interval estimates of individual treatment effects.

A key strength of the paper is that the problem setting is important and practically relevant. Estimating treatment effects under hidden confounding is challenging, and the idea of recovering a distribution over latent confounders from proxies is interesting. The paper also attempts to connect proxy based identification, latent variable modeling, and uncertainty quantification, which is a promising direction.

However, I found several parts of the formulation insufficiently clear. First, it is not fully clear whether the learned outcome function is consistent with the structural outcome model and the counterfactual interpretation used in the paper. Since the final ITE estimates depend directly on this learned outcome function, this point is important for the causal interpretation of the method. Second, although the method aims to recover the unobserved confounder using a VAE, the paper does not sufficiently discuss whether the inferred latent variable actually corresponds to the true confounder, beyond the stated identifiability assumptions. More empirical or theoretical discussion on the quality of latent confounder recovery would strengthen the paper. Third, the Bayesian structure of the model is not organized clearly enough. In particular, the precise probabilistic role of the GP prior, the VAE posterior, and the interval representation is difficult to follow. This makes it hard to assess whether the proposed model truly provides coherent Bayesian uncertainty quantification.

**Additional Comments:**

I would also like to encourage the authors to present the proposed method as a complete Bayesian model. In particular, it would be helpful to explicitly state what the model parameters are, what prior distributions are assigned to them, what the likelihood is, and what posterior distribution is being approximated or optimized.

For example, the paper should clearly specify the roles of the VAE parameters, the latent confounder $U$, the additive noise variables, the GP kernel hyperparameters, and the interval valued latent representation. It should also clarify whether the GP component defines a prior over latent functions, a prior over finite dimensional latent vectors, or only a regularization term in the training objective.

A clearer Bayesian formulation would make it easier to understand the proposed inference procedure. In particular, the paper should explain how the posterior over $U$ is defined, how the variational posterior $q_\phi(u\mid z)$ relates to the GP prior, and how the test time GP posterior is connected to the VAE encoder. Without this clarification, it is difficult to assess whether the proposed method provides coherent Bayesian uncertainty quantification or whether the GP term should be interpreted mainly as a heuristic regularizer.

**Audience:**

Yes

**Audience Explanation:**

The problem of estimating individual treatment effects under unobserved confounding is important, and the idea of using proxy variables to infer a distribution over latent confounders is interesting. Although I have concerns about the current formulation and evidence, the overall direction of combining proxy based causal inference with uncertainty aware latent modeling is relevant and potentially useful.

**Broader Impact Concerns:**

This work is mainly methodological and theoretical, and I do not see major broader impact concerns. The proposed direction could have positive impact by improving uncertainty quantification in individual treatment effect estimation, especially in domains where unobserved confounding is common.

If further developed and validated, this type of uncertainty aware causal inference method may be useful for supporting more reliable decision making in areas such as healthcare, economics, and public policy. The paper also contributes to the broader discussion on how latent variable models and Bayesian uncertainty modeling can be used in causal inference.

Overall, I do not identify any specific negative societal impacts beyond the general caution that causal effect estimates should be interpreted under the assumptions of the model.

**Claims And Evidence:**

No

**Claims Explanation:**

The main claims of the paper are only partially supported by the current evidence. While the problem is important and the proposed direction is interesting, several key parts of the method are not sufficiently justified, including the learned outcome function, the recovery of the latent confounder, and the use of the GP component.

First, the paper relies on a learned outcome function $f_\psi(u,t,z_Y)$ to estimate counterfactual outcomes and ITEs. However, it is not clear whether this learned outcome function is a statistically consistent or unbiased estimator of the true structural outcome function $f(T,U,Z_Y)$. Since the final ITE estimate is computed by evaluating the learned function under $t=0$ and $t=1$, the causal validity of the method depends critically on whether $f_\psi$ can recover the correct counterfactual outcome mechanism. The paper does not provide enough justification for this point.

Second, the method infers a latent variable $U$ using a VAE, but it is not sufficiently demonstrated that the inferred latent representation actually recovers the true unobserved confounder, even up to the equivalence class claimed in the identifiability analysis. The identifiability result is stated at the population or structural level, whereas the implemented model uses a finite sample VAE objective. The connection between the theoretical identifiability of $U$ and the empirical recovery of $U$ by the VAE is not clearly established.

Relatedly, the paper does not sufficiently justify why it is valid to plug the VAE inferred latent variable $\hat U$, rather than the true latent confounder $U$, into the learned outcome function. If $\hat U$ does not preserve all confounding information relevant to both treatment assignment and outcome generation, then the estimated potential outcomes may still be biased. This issue is central to the proposed method, but the paper does not provide enough theoretical or empirical evidence that the inferred latent variable is adequate for causal effect estimation.

Another concern is the treatment of the GP kernel. The kernel hyperparameters, such as the lengthscale and variance, appear to be fixed constants in the experiments rather than learned from the data. This is problematic because the GP component is supposed to encode smoothness and similarity structure in the proxy space. If these hyperparameters are fixed without sufficient justification or sensitivity analysis, it is unclear whether the resulting uncertainty estimates are well calibrated. Moreover, in the training objective, the interval GP prior appears to depend only on the marginal variance $k(Z_i,Z_i)$. For an RBF kernel, $k(Z_i,Z_i)=\sigma_f^2$, so the lengthscale does not affect this term. This raises the question of how the claimed GP induced latent space coherence is actually learned during training.

There is also a mismatch between training and inference. During training, the GP prior term seems to use only pointwise marginal probabilities and does not use the cross sample covariance structure $k(Z_i,Z_j)$. In contrast, the inference procedure uses the full GP posterior involving the kernel matrix and cross covariances. Thus, the GP covariance structure is not used to regularize the latent representation during training, but it is used for prediction at test time. This makes the role of the GP inconsistent between training and inference.

The notation and formulation of Proposition 1 also require clarification. The paper writes

$$
\hat{U} = (\hat{U}_1,\dots,\hat{U}_n) \sim \mathcal{GP}(0,K(Z_i,Z_j)).
$$

This notation is imprecise. When $d=1$, $\hat U$ is a finite dimensional random vector, so it should be described as a finite dimensional Gaussian marginal induced by a GP, for example $\hat{\mathbf U}\sim \mathcal N(0,K_Z)$, rather than as directly following a GP. When $d>1$, the latent confounder $U\in\mathbb R^d$ is vector valued, whereas the paper appears to use a standard scalar output GP prior. The paper does not specify whether different latent dimensions are modeled independently, through a multi output GP, or through some structured covariance across latent dimensions.

Finally, the Bayesian model framework is not organized clearly enough. It is difficult to understand the precise probabilistic relationship among the VAE encoder, the decoder, the GP prior, the interval representation, and the test time GP posterior. Because the paper presents the method as uncertainty aware and Bayesian, a clearer generative model and posterior inference framework are necessary. Without such clarification, it is hard to assess whether the proposed method provides coherent Bayesian uncertainty quantification.

**Requested Changes:**

1. Please provide a clearer and more explicit Bayesian model formulation. In particular, the paper should specify the full generative model, the variational posterior, the role of the GP prior, the interval representation, and the test time posterior prediction procedure in a unified probabilistic framework. As currently written, it is difficult to understand the precise probabilistic relationship among the VAE encoder, the decoder, the GP prior, the interval valued latent representation, and the GP posterior used during inference.

2. Please clarify the statistical and causal validity of the learned outcome function $f_\psi(u,t,z_Y)$. Since the ITE estimate is obtained by evaluating this learned function under $t=0$ and $t=1$, the paper should discuss whether $f_\psi$ is expected to be a consistent or unbiased estimator of the true structural outcome function $f(T,U,Z_Y)$. Without this clarification, it is hard to assess whether the estimated counterfactual outcomes have the intended causal interpretation.

3. Please provide stronger justification that the VAE inferred latent variable $\hat U$ actually recovers the true latent confounder $U$, at least up to the equivalence class described in the identifiability analysis. The theoretical identifiability result is stated at the structural or population level, but the implemented method uses a finite sample VAE training objective. The paper should clarify how the theoretical identifiability of $U$ translates into empirical recovery of $U$ by the proposed VAE model.

4. Please clarify why it is valid to plug the VAE inferred latent variable $\hat U$ into the learned outcome function in place of the true latent confounder $U$. If $\hat U$ does not preserve all confounding information relevant to both treatment assignment and outcome generation, the estimated potential outcomes may remain biased. This is a central step in the method and requires more explicit theoretical or empirical support.

5. Please clarify the role of the GP kernel hyperparameters. The lengthscale and variance appear to be fixed constants in the experiments rather than learned from the data. If they are fixed, the paper should explain how these values are selected and provide sensitivity analyses. If they are learned, the objective through which they are learned should be stated explicitly. This is particularly important because, under the stated training prior, the interval GP term appears to depend only on $k(Z_i,Z_i)$. For an RBF kernel, $k(Z_i,Z_i)=\sigma_f^2$, so the lengthscale does not affect this term. Therefore, it is unclear how the GP smoothness structure is learned during training.

6. Please address the mismatch between training and inference in the use of the GP covariance structure. During training, the GP prior term seems to use only pointwise marginal probabilities and does not use the cross sample covariance $k(Z_i,Z_j)$. In contrast, the inference procedure uses the full GP posterior involving the kernel matrix and cross covariances. The paper should explain why this is a coherent inference procedure, or revise the training objective so that the GP covariance structure is also used during training.

7. Please revise the notation and formulation of Proposition 1. The paper writes
$$
\hat{U} = (\hat{U}_1,\dots,\hat{U}_n) \sim \mathcal{GP}(0,K(Z_i,Z_j)).
$$
This notation is imprecise. When $d=1$, $\hat U$ is a finite dimensional random vector, so it should be described as a finite dimensional Gaussian marginal induced by a GP, for example $\hat{\mathbf U}\sim \mathcal N(0,K_Z)$, rather than as directly following a GP. The paper should distinguish clearly between a GP as a prior over functions and the finite dimensional Gaussian distribution induced by evaluating that function at observed inputs.

8. Please clarify the model when $d>1$. Since the latent confounder satisfies $U\in\mathbb R^d$, the latent variable is vector valued when $d>1$. However, the paper appears to use a standard scalar output GP prior. The authors should specify whether different latent dimensions are modeled independently, through a multi output GP, or through some structured covariance across latent dimensions. Without this discussion, the proposed GP formulation is under specified for multidimensional latent confounders.

9. Please clarify the role and selection of $Z_Y$. The paper assumes that $Z_Y\subseteq Z$ directly affects the outcome, but it does not explain whether $Z_Y$ is assumed to be given by domain knowledge, selected by the researcher, or learned from data. It would also be helpful to explicitly clarify whether the dimension of $Z_Y$ is always assumed to satisfy $k'\leq k$, and how this subset is represented in the implementation. Since $Z_Y$ directly enters the outcome function, this choice can affect both prediction and causal interpretation.

---

> ### Author Response · Authors · 2026-05-27
> **Acknowledgement of Reviews and Upcoming Revision**
>
> Dear Reviewers,
>
> We sincerely appreciate your high-quality reviews, which are very helpful to us. We are currently working intensively on additional experiments and theoretical clarifications to address your concerns. We will post our revised manuscript and detailed responses shortly to begin the discussion.
>
> Thank you again for your patience and valuable feedback.
>
> Best regards,
> The Authors

---

> > ### Author Response · Authors · 2026-06-22
> >
> > Dear Reviewers,
> >
> > We have added many new experiments and made substantial revisions to the manuscript. We are now at the final revision stage and will upload the revised version this week.
> >
> > Many thanks for your patience.
> >
> > Best regards, The Authors

---

> ### Author Response · Authors · 2026-06-26
>
> Many thanks and the comments are very helpful in improving the paper. We have addressed all the issues - please see more details below.
>
> **Linking structural identifiability, finite-sample latent recovery, and ITE validity.**  We thank the reviewer for these important comments. In response, we have added a new subsection, Section 5.5, entitled *From Structural Identifiability to Finite-Sample Estimation*, which explicitly addresses the relationship between latent confounder identifiability, finite-sample latent recovery, outcome estimation, and ITE estimation.
>
> The consistency of the estimated potential outcomes $f_\psi(t,\hat{U},Z_Y)$ depends on accurate recovery of the latent confounder representation as well as sufficient approximation capacity of the outcome model. Under standard assumptions, including compact support of the latent space and Lipschitz continuity of the structural outcome function, we show that if the outcome model is trained consistently and the latent representation converges to the identifiable equivalence class, then the learned outcome head converges to the corresponding structural outcome mechanism. We further establish that errors in latent recovery induce bounded errors in the predicted potential outcomes. In particular, we derive an explicit bound on the ITE estimation error, demonstrating how inaccuracies in latent recovery and outcome model estimation jointly contribute to the final counterfactual error. This analysis clarifies that the estimated ITE converges to the true ITE as both the latent approximation error and the outcome estimation error decrease with increasing sample size.
>
> **Empirical support for latent recovery and ITE validity.**  To provide an intuitive illustration, the revised manuscript includes an example based on Case 12 in the synthetic experiments, demonstrating empirical latent recovery and ITE estimation; see Figure 4. Beyond reporting PEHE and coverage rates across all experiments, we also report latent recovery error and latent correlation in the scalability analysis; see Table 7 and Figures 8 and 9. These additional results provide empirical support for the validity of the proposed method.
>
> **Presenting IntervalGP-VAE within a unified Bayesian framework.** In response, we have substantially revised Sections 5.1 and 5.2, including Proposition 1, to present the proposed IntervalGP-VAE within a unified Bayesian framework and to clarify the modelling assumptions underlying the GP component. In particular, the revised manuscript now explicitly distinguishes between the generative model, the variational posterior approximation, the interval-based GP regularization term used during training, and the GP posterior prediction procedure used at inference time. For $U \in \mathbb{R}^d$, the GP prior is defined over a vector-valued latent recovery function $F(Z) = (F_1(Z), \ldots, F_d(Z))$, with each component assigned an independent scalar-output GP prior.
>
> **Distinguishing training-time GP regularization from inference-time GP prediction.** We explicitly state that the GP serves two distinct purposes. First, during training, it provides point-wise interval-based regularization through the ELBO objective. Second, during inference, it enables uncertainty-aware interpolation and posterior prediction through the full GP posterior. Consequently, the cross-sample covariance structure is used only at inference time when predicting latent confounders for unseen proxy observations.
>
> To further clarify the role of the GP during training, we explicitly distinguish this formulation from standard GP regression. Because the latent confounder is unobserved and represented through interval-valued encoder posteriors, the model is not trained using the standard GP marginal likelihood objective. Instead, the training objective depends only on the marginal GP distribution evaluated at each observed proxy value. As a result, the GP regularization term is intentionally point-wise and does not incorporate cross-sample covariance terms during training.
>
> Section 5.1 now explains that this design choice arises naturally from the interval-valued latent representation and the ELBO-based training objective.

---

> > ### Author Response · Authors · 2026-06-26
> >
> > **Clarifying the role and selection of GP kernel hyperparameters.** We have also expanded the discussion of kernel hyperparameters. Since the point-wise training objective does not depend on cross-sample covariance terms, kernel lengthscale parameters cannot be learned through marginal-likelihood optimization as in standard GP regression. Furthermore, uncertainty in the latent representation is already captured by the encoder posterior variance $\left(\sigma_u^i\right)^2$, which determines the width of the latent intervals. Therefore, the current implementation employs fixed kernel hyperparameters and a fixed GP observation-noise term. The revised manuscript, Section 5.1, discusses this modelling choice explicitly and clarifies its implications.
> >
> > To address sensitivity to the kernel choice, we have added a kernel ablation study in Section 6.1.3. In this study, the encoder, decoder, outcome head, latent dimension, training schedule, batch size, learning rate, and data-generating mechanisms are kept fixed, and only the latent GP kernel is changed. The results in Table 7 show that IntervalGP-VAE remains stable across RBF, Matérn, rational quadratic, and linear-plus-RBF kernels, with consistently low PEHE, high empirical coverage, and strong latent-confounder recovery correlation. These results support the use of the RBF kernel as a stable default and indicate that the proposed uncertainty estimates are not dependent on a finely tuned kernel specification.
> >
> > To support TMLR's reproducibility expectations, we have added an anonymous code link: https://github.com/anonymous-for-tmlr/IntervalGP-VAE. This link will be included in the final manuscript to facilitate reproducibility of the experiments.
> >
> > **Justifying inference-time GP interpolation through a smoothness assumption.** To support inference-time GP prediction, we also introduce an explicit smoothness assumption on the posterior recovery map from proxies to latent confounders in Section 5.1. This assumption is required because, unlike standard GP regression, the proposed training objective does not learn smoothness through a joint GP marginal likelihood involving cross-sample covariance terms. Instead, the GP covariance structure is introduced only at inference time to perform posterior interpolation for unseen proxy observations. Consequently, some regularity of the latent recovery function is needed to justify the use of GP-based interpolation and uncertainty propagation beyond the observed samples.
> >
> > Finally, we have revised Proposition 1, in Section 5.2, to improve the notation and clarify the presentation of multidimensional latent confounders.
> >
> > **Clarifying the selection, dimension, and implementation of $Z_{Y}$.** We have expanded Section 2.1 to provide a more precise description of the role of $Z_{Y}$. We further clarify why we hold $Z_{Y}$ fixed across counterfactual worlds in the counterfactual framework. $Z_{Y}$ is treated as a pre-specified subset, chosen based on domain knowledge or the experimental design. Its dimension is denoted by $k'$ and satisfies $0 \leq k' \leq k$ by definition.

---

> > > ### Comment · Reviewer_B7U7 · 2026-06-29
> > >
> > > We thank the authors for the substantial revision. The added Section 5.5, the unified
> > > Bayesian formulation in Section 5.1, and the expanded empirical analyses (latent recovery
> > > in Section 6.1.5, the ablations in Sections 6.1.1–6.1.4) meaningfully improve the
> > > clarity of the paper. The notation in Proposition 1, the treatment of $d>1$, and the
> > > role of $Z_Y$ are now much clearer. We have a few remaining points where additional
> > > clarification would help us fully assess the method.
> > >
> > >
> > > ## 1. Bayesian formulation and the posterior over the GP
> > >
> > > Section 5.1 places a GP prior over the latent recovery function,
> > > $F_r \sim \mathcal{GP}(0, k(\cdot,\cdot))$ for $r=1,\dots,d$, with $U^i = F(Z^i)$
> > > (Eqs. 10–13). Our main question is the following: once a GP prior is placed on $F$, a
> > > fully Bayesian treatment would compute the posterior over $F$ given the observed data, i.e. the posterior of $F$ given
> > > $\{ Z^i \}$, $\{ T^i \}$ and $\{ Y^i \}$ over $i = 1, \dots, n$ (equivalently
> > > $p(U \mid Z, Y)$), by combining this prior with the likelihood terms
> > > $p_\theta(Z^i \mid U^i)$ and $p_\psi(Y^i \mid T^i, U^i, Z^i_Y)$. Could the authors
> > > clarify why this posterior is not computed, and where in the framework the GP prior is
> > > actually combined with the likelihood?
> > >
> > > As written, the connection is not transparent to us for two reasons. During training,
> > > the KL term $\mathrm{KL}(q_\phi \,\|\, p_{GP})$ in Eq. 16 is replaced by the
> > > interval-containment term in Eq. 17, which depends only on the marginal
> > > $\mathcal{N}(0, k(Z^i, Z^i))$, so the off-diagonal covariance $k(Z^i, Z^j)$ of the
> > > prior $\mathcal{N}(0, K_Z)$ (Eqs. 12–13) never enters the objective. At inference, the
> > > GP posterior conditions on the encoder outputs $\mu_{u,r}$ rather than on the outcomes
> > > $Y$.
> > >
> > > It would help us greatly if the authors could clarify whether the framework is indeed
> > > intended to approximate a posterior over $F$, and if so, to state the Bayesian
> > > components explicitly: what the model parameters are, what constitutes the observed
> > > data, and the precise forms of the prior, the likelihood, and the posterior being
> > > approximated. Making these elements explicit would clarify how the GP prior, the
> > > encoder posterior $q_\phi$, and the inference-time GP posterior fit together within a
> > > single probabilistic model.
> > >
> > > ## 2. Whether the recovered $\hat{U}$ supports latent ignorability
> > >
> > > The ITE identification relies on the latent ignorability assumption
> > > $(Y(0), Y(1)) \perp T \mid U$ from Section 2, which is stated for the *true*
> > > confounder $U$. In practice, however, the ITE is computed from the recovered
> > > representation $\hat{U}$, i.e.
> > > $\hat\tau = f_\psi(1, \hat{U}, Z_Y) - f_\psi(0, \hat{U}, Z_Y)$, and in general
> > > $\hat{U} \neq U$.
> > >
> > > In this case, we were wondering how one can be sure that ignorability still holds for
> > > the recovered representation, that is, $(Y(0), Y(1)) \perp T \mid \hat{U}$,
> > > and whether there is any practical way to check or verify it. A brief discussion on
> > > this point would help us better understand the validity of the plug-in ITE.
> > >
> > > ## 3. Making the bound in Section 5.5 more explicit
> > >
> > > We had a couple of questions about the terms on the right-hand side of the ITE error
> > > bound in Eqs. 21, 27 and 31.
> > >
> > > First, we were not entirely sure what $\epsilon_U(N)$ refers to. Is it a quantity that
> > > can actually be computed, or is it assumed to be some function that decreases with $N$?
> > > It would help us a lot if its explicit form as a function of $N$ could be given (or, if
> > > that is not available, the assumptions under which it is taken to decrease).
> > >
> > > Second, we were wondering the same about the three error terms appearing in the bound:
> > > the generalisation error $\mathcal{E}_g$, the approximation error $\mathcal{E}_a$ and
> > > the optimisation error $\mathcal{E}_o$. Could the authors clarify whether these can be
> > > made explicit as functions of $N$ as well? Knowing how each term behaves as $N$ grows would make the bound much
> > > easier to interpret.
> > >
> > > We thank the authors again for the careful revision and hope these clarifications will
> > > further strengthen the paper.

---

> > > > ### Author Response · Authors · 2026-07-02
> > > >
> > > > **Bayesian formulation and posterior over the GP.** We sincerely thank the reviewer for the encouraging assessment of our revision and for recognising the improvements made in the previous round. We are very grateful for this further insightful comment, which identifies an important point regarding the probabilistic interpretation of the proposed IntervalGP framework. We agree that the presentation could still more clearly explain the motivation behind the IntervalGP design and the respective roles of the GP prior, the variational posterior, and the inference-time GP posterior.
> > > >
> > > > The key point is that the proposed IntervalGP is intentionally designed as a Gaussian-process formulation for *interval-valued latent inference*, rather than as a standard GP regression model. This design follows directly from the latent-variable causal formulation introduced in Section~2. Unlike standard GP regression, where both the inputs and the corresponding function values are directly observed and learning proceeds by maximizing the GP marginal log-likelihood over the joint covariance matrix of all observations, our problem consists of two coupled generative processes: the structural proxy model $Z=g(U)+\epsilon$ and the structural outcome model $Y=f(T,U,Z_Y).$ Consequently, the learning objective is substantially different: rather than regressing directly observed function values, the objective is to recover the latent confounder from the observed proxies and subsequently estimate the individualized outcome using the recovered latent representation.
> > > >
> > > > This distinction naturally motivates our modelling choices. The IntervalGP is introduced to model the latent recovery function $F:Z\rightarrow U$, which is the inverse of the proxy-generation mechanism, rather than the causal outcome function itself. Accordingly, the IntervalGP naturally takes the observed proxy variables $Z$ as its inputs. The outcome variable $Y$ is incorporated separately through the causal likelihood during variational learning of the latent representation, instead of directly conditioning the latent recovery function. Consequently, the framework approximates the latent posterior $p(U\mid Z)$ through the variational posterior $q_{\phi}(U\mid Z)$, rather than constructing a GP posterior of the form $p(U\mid Z,Y)$. This preserves the interpretation of the IntervalGP as modelling proxy-based latent recovery while allowing the outcome information to influence the learned learned representation through the causal decoder.
> > > >
> > > > A second important distinction is that the latent recovery function is never directly observed. In standard GP regression, the function values are available as point-valued observations and the GP marginal likelihood can therefore be optimized directly. In our setting, however, the encoder produces interval-valued posterior summaries of the latent variables rather than exact function values. Consequently, the IntervalGP prior is reformulated in terms of interval-containment probabilities, which evaluate the probability that the latent function value lies within the learned encoder interval. Since these intervals are defined independently for each latent variable, the resulting IntervalGP regularisation is naturally pointwise during training and therefore does not require the full cross-sample covariance matrix in the optimisation objective. This is a deliberate modelling choice arising from interval-valued latent observations.
> > > >
> > > > The full GP covariance is instead reserved for inference. After the latent representation has been learned, the inference-time IntervalGP posterior exploits the covariance between neighbouring proxy observations to interpolate the latent recovery function for previously unseen samples. This interpolation is justified by the smoothness assumption on the posterior recovery map $m(z)=\mathbb{E}[U\mid Z=z]$, which we introduced in the revision. Thus, the proposed framework deliberately separates latent representation learning from posterior interpolation: interval-valued local GP regularisation is used during training, while the full GP posterior is used during inference for uncertainty-aware latent recovery.
> > > > It is precisely this separation that enables scalable mini-batch optimisation during training while still allowing the complete GP covariance structure to contribute to uncertainty-aware posterior interpolation at inference.

---

> > > > > ### Author Response · Authors · 2026-07-02
> > > > >
> > > > > To make this interpretation explicit, we have further revised Section~5.1 in two ways.
> > > > >
> > > > > * First, we have added a new introductory subsection entitled *Design Motivation of IntervalGP*, which explains the rationale behind the proposed IntervalGP formulation, why it differs from standard GP regression, and why interval-valued latent observations naturally lead to pointwise GP regularisation during training and full GP posterior interpolation during inference.
> > > > >
> > > > > * Second, we have added a new *Summary of the Probabilistic Framework* paragraph, which explicitly lists the observed data, latent variables, prior, likelihood, variational posterior being approximated, and trainable model parameters.
> > > > >
> > > > > The newly added text highlighted in **magenta**. We hope these additions make the modelling assumptions and probabilistic interpretation of the proposed IntervalGP framework substantially clearer.
> > > > >
> > > > > **Whether the recovered $\hat{U}$ supports latent ignorability.**  We thank the reviewer for raising this important question regarding the role of the recovered latent $U$ in preserving the latent ignorability assumption. We agree that distinguishing the theoretical result from the practical finite-sample estimator improves the presentation, and we have revised the manuscript accordingly.
> > > > >
> > > > > First, we would like to clarify that the latent ignorability assumption,
> > > > > $$
> > > > > (Y(0),Y(1)) \perp T \mid U,
> > > > > $$
> > > > > is a standard causal identification assumption and is not involved directly in our theoretical identifiability analysis in Theorem 1. The purpose of Theorem 1 is to establish *latent recovery*: under the proxy model and the proposed structural assumptions (conditional independence, variability, and bounded completeness), it provides sufficient conditions under which the latent confounder is identifiable from the observed proxies, up to a smooth invertible transformation. This does not involve the treatment or outcome variables and therefore does not rely on the latent ignorability assumption.
> > > > >
> > > > > Second, at the theoretical level, Theorem~1 establishes that the recovered latent representation satisfies $\hat U = h(U)$, for some smooth invertible mapping $h$. Since invertible reparameterisations preserve the information contained in the latent confounder, conditioning on $\hat U$ is theoretically equivalent to conditioning on $U$. Namely, if latent ignorability holds for the true confounder,
> > > > > $$
> > > > > (Y(0),Y(1)) \perp T \mid U,
> > > > > $$
> > > > > then it also holds for the recovered representation,
> > > > > $$
> > > > > (Y(0),Y(1)) \perp T \mid \hat U.
> > > > > $$
> > > > >
> > > > > In practice, the learned latent representation is estimated from finite data and therefore only approximates the population solution. Rather than assuming that latent ignorability is exactly preserved under estimation error, our analysis in Section~5.5 explicitly studies this practical setting by deriving a bound on the ITE estimation error as a function of the latent recovery error. This provides a quantitative connection between imperfect latent recovery and downstream causal estimation.
> > > > >
> > > > > To make this distinction clearer, we have added a new paragraph call **Relation to latent ignorability** at the end of Section~5.5 to explicitly clarify this relationship and discuss the latent recovery problem from finite-samples. The newly added text highlighted in **magenta**.

---

> > > > > > ### Author Response · Authors · 2026-07-02
> > > > > >
> > > > > > Notably, while exact latent ignorability cannot be directly verified from purely observational data because the true $U$ and potential outcomes are unobserved, our manuscript already contain results to assess whether the learned representation $\hat U$ provides a plausible surrogate for the unobserved confounder in synthetic settings. There is an empirical evaluation of latent recovery quality using both direct and indirect diagnostics. In synthetic settings, where the true latent confounder $U$ is observed, we directly evaluate recovery accuracy by first aligning $\hat U$ with $U$, since the latent representation is identifiable only up to an invertible reparameterisation. This alignment accounts for common linear indeterminacies such as rotation, scaling, sign changes, and permutation. We then report aligned latent RMSE and average absolute latent correlation for both the raw encoder posterior mean and the GP-smoothed latent representation in Section~6.1.5. These metrics quantify, respectively, the distance between the recovered and true latent variables and the degree to which the learned representation preserves the latent confounding structure. For real and semi-synthetic datasets, where the true $U$ is unavailable, direct latent-recovery metrics cannot be computed. We therefore rely on indirect evidence, particularly uncertainty calibration, measured by empirical ITE interval coverage and interval width. These quantities assess whether the uncertainty induced by imperfect latent recovery is appropriately propagated to downstream treatment-effect estimates. Thus, while these diagnostics cannot prove latent ignorability, they provide empirical evidence on whether $\hat U$ behaves as a reasonable confounding-sufficient representation for plug-in ITE estimation.
> > > > > >
> > > > > > **Making the Finite-Sample ITE Error Bound More Explicit.** We thank the reviewer for this helpful comment. We agree that the interpretation of the finite-sample error terms should be made more explicit. We have further revised the manuscript to expand Section~5.5 (*From Structural Identifiability to Finite-Sample Estimation*) to strengthen the assumptions underlying the finite-sample analysis and clarify the behaviour of each error term.
> > > > > >
> > > > > > Specifically, we have updated the subsection *Regularity assumptions for finite-sample estimation*. Besides the previously introduced Bi-Lipschitz Regularity of the identifiable mapping $h$, we have strengthened the Compact Support assumption to cover both the latent and observed variables, and introduced a new *Bounded Empirical Objective* assumption for the proposed IntervalGP-VAE training objective. These additional assumptions provide the standard regularity conditions required to justify the uniform-convergence arguments based on Rademacher complexity used in the finite-sample analysis.
> > > > > >
> > > > > > We also explicitly clarify that only the statistical generalisation error decreases directly with $N$, whereas the approximation and optimisation errors are determined by the chosen hypothesis class and optimisation procedure, respectively.
> > > > > >
> > > > > > Specifically, for the outcome head, we decompose $$\mathcal E_{\mathrm{out}}(N)=\mathcal E_{\mathrm{gen}}(N)+\mathcal E_{\mathrm{approx}}+\mathcal E_{\mathrm{opt}}.$$ Under standard bounded-loss assumptions, $$\mathcal E_{\mathrm{gen}}(N)=O\left(\mathfrak R_N(\mathcal F_\Psi)\right)=O(N^{-1/2}),$$ where $\mathfrak R_N(\mathcal F_\Psi)$ is the Rademacher complexity of the outcome-head hypothesis class.
> > > > > >
> > > > > > The approximation error $$\mathcal E_{\mathrm{approx}}=\inf_{f_\psi\in\mathcal F_\Psi}\Vert f_\psi-\tilde f\Vert_\infty$$ is not primarily a function of $N$ for a fixed architecture; rather, it reflects the expressive capacity of the chosen neural-network hypothesis class and decreases as the hypothesis class becomes more expressive.
> > > > > >
> > > > > > The optimisation error $$\mathcal E\_{\mathrm{opt}}=\hat{\mathcal R}\_N(f\_{\psi})-\min\_{f\in\mathcal F\_{\Psi}}\hat{\mathcal R}\_N(f)$$ depends on the optimisation algorithm and the number of training iterations rather than directly on $N$, and decreases as optimisation converges.
> > > > > >
> > > > > > Similarly, the latent approximation error can be decomposed as $$\epsilon\_U(N) = \mathcal E\_{\mathrm{gen}}^{U}(N) + \mathcal E\_{\mathrm{approx}}^{U} + \mathcal E\_{\mathrm{opt}}^{U},$$ where $$\mathcal E\_{\mathrm{gen}}^{U}(N) = O(N^{-1/2})$$ under the same regularity assumptions, while $\mathcal E\_{\mathrm{approx}}^{U}$ and $\mathcal E\_{\mathrm{opt}}^{U}$ represent the latent-model approximation and optimisation errors, respectively.
> > > > > >
> > > > > > Consequently, the ITE estimation bound may be interpreted as $$\mathbb E[ |\hat\tau(\hat u,Z\_y)-\tau(u,Z\_y)| ] \le O(N^{-1/2}) + 2\mathcal E\_{\mathrm{approx}} + 2\mathcal E\_{\mathrm{opt}} + 2L\_fL\_{h^{-1}} \left[ O(N^{-1/2}) + \mathcal E\_{\mathrm{approx}}^{U} + \mathcal E\_{\mathrm{opt}}^{U} \right].$$ All of the above clarifications have been incorporated into the revised Section~5.5, with new text highlighted in **magenta**.

---

### Author Response · Authors · 2026-06-26
**Summary of Major Revisions**

We sincerely thank the Associate Editor and all reviewers for their detailed and constructive comments. Their feedback has significantly improved both the theoretical foundations and the presentation of the paper. In response, we have undertaken a substantial revision of the manuscript. The major changes are summarized below.

* **Strengthened theoretical identifiability analysis.**
  We have revised the theoretical assumptions and proofs underlying the identifiability results. In particular, we introduce an explicit Bounded Completeness assumption to justify the transition from the discrete to the continuous latent setting, reformulate the proxy variability assumptions to include joint injectivity, partitionable proxy variability, and non-degenerate additive noise, and substantially revise the proof of Theorem 1. We also clarify that the proxy-number requirement provides a sufficient condition rather than a tight or necessary bound.

* **Improved treatment of latent confounder recovery and finite-sample estimation.**
  To bridge the gap between structural identifiability and practical learning, we add a new subsection, Section 5.5, entitled *From Structural Identifiability to Finite-Sample Estimation*. This section introduces finite-sample consistency assumptions, analyzes latent confounder approximation, derives bounds for outcome and ITE estimation, and explains why the learned latent representation can serve as a suitable surrogate for the unobserved confounder.

* **Revised Bayesian formulation of the proposed IntervalGP-VAE.**
  Section 5 has been substantially reorganized to present the proposed framework within a unified Bayesian formulation. The revised presentation explicitly distinguishes the structural generative model, variational posterior, interval-valued latent representation, GP prior, training objective, and inference-time GP posterior prediction, thereby clarifying the probabilistic relationships among all model components.

* **Clarified the role of the Gaussian Process component.**
  The discussion of the GP model in Section 5 has been significantly expanded to clarify the distinction between training and inference, the role of kernel hyperparameters, the interpretation of the interval GP prior, the treatment of multidimensional latent confounders, and the assumptions supporting inference-time GP interpolation. We also revise Proposition 1 accordingly and clarify the notation used throughout the GP formulation.

* **Improved positioning with respect to related causal inference frameworks.**
  We revise the related work and problem formulation to more clearly position our method within the latent-variable causal inference literature, while explicitly distinguishing it from proximal causal inference in Section 3. We also clarify the role of the outcome-related proxy subset $Z_Y$ within the structural causal model and counterfactual estimation procedure.

---

> ### Author Response · Authors · 2026-06-26
> **Summary of Major Revisions**
>
> * **Substantially expanded experimental evaluation.**
>   To further validate the proposed framework and address the reviewers' suggestions, we have added several new empirical studies covering latent confounder recovery, uncertainty quantification, robustness, and scalability in Section 6. To support TMLR's reproducibility expectations, we have added an anonymous code link:  https://github.com/anonymous-for-tmlr/IntervalGP-VAE. This link will be included in the final manuscript to facilitate reproducibility of the experiments. The new experimental analyses include the following:
>
>   * **Expanded empirical evaluation with additional baselines.**
>     We have substantially expanded the experimental evaluation beyond the original comparison with TEDVAE. In the revised manuscript, we compare IntervalGP-VAE against multiple representative baselines, including latent-variable, proxy-based, conformalized, and ensemble-style treatment-effect estimators. We also report runtime for all compared methods under the same hardware setting, for both the synthetic and semi-synthetic experiments, to provide a fairer assessment of accuracy, uncertainty calibration, and computational cost.
>
>   * **Latent recovery and ITE estimation analysis.**
>     To make the latent recovery mechanism more transparent, we add an illustrative case study based on Case 12 in the synthetic experiments. The revised figures visualize both recovered latent confounder intervals and ITE intervals after sorting individuals by their true latent values and true treatment effects. In addition to PEHE and empirical coverage, we now report latent recovery error and latent correlation in the experimental analysis to provide direct empirical evidence that the inferred latent representation recovers the underlying confounding structure in identifiable synthetic settings.
>
>   * **Kernel and uncertainty ablation studies.**
>     We have added a kernel ablation study to examine whether the proposed method is sensitive to the choice of latent GP kernel. The revised experiments compare RBF, Matérn, rational quadratic, and linear-plus-RBF kernels while keeping all other model components fixed. We also add an ablation study on ITE interval uncertainty sources, comparing point-only estimation, encoder Monte Carlo latent sampling, latent-GP-smoothed Monte Carlo intervals, and the full GP-ITE posterior. These experiments clarify how different uncertainty components contribute to interval coverage, interval width, and interval score.
>
>   * **Scalability analysis.**
>     To support the scalability claims, we add experiments on larger synthetic datasets with $n \in \{(1000, 5000, 10000)\}$. We compare the current per-sample IntervalGP prior with sparse inducing-point IntervalGP variants using $ M_{\mathrm{ind}} \in \{(32,64,128)\}$ inducing points. The results show that the per-sample IntervalGP prior remains an efficient and accurate default at larger sample sizes, while sparse inducing-point regularization provides a feasible extension for more structured GP regularization.
>
>   * **Proxy-quality and assumption-violation analysis.**
>     To better explain the behavior observed on the IHDP benchmark, we add controlled experiments studying how proxy noise, proxy informativeness, redundancy, and conditional-independence violations affect latent recovery and ITE uncertainty calibration. These results show that weaker or dependent proxies can degrade latent recovery and lead to under-coverage, providing a synthetic-to-real explanation for the lower empirical coverage observed on IHDP under partial violation of the proxy-identifiability assumptions.
>
> * **Improved presentation throughout the manuscript.**
>   We have revised numerous sections to improve clarity, motivation, and consistency of terminology. We have also expanded discussions where requested by the reviewers, corrected notation, and incorporated additional explanations to better connect the theoretical developments with the practical implementation. Several theoretical results have been reorganized for improved clarity. The original Theorem 2 has been reformulated as a discussion subsection describing invariance under latent transformations, while the substantive estimation analysis has been moved to the new finite-sample estimation section. In addition, the notation table, Table 1, has been revised to improve readability and maintain consistency throughout the manuscript.
>
> For clarity, modifications in the revised manuscript are shown in red. Below, we respond to each reviewer individually.